# Threat reduction must be coupled with targeted recovery programmes to conserve global bird diversity

Kerry Stewart [1] ✉, Chris Venditti [1], Carlos P. Carmona [2], Joanna Baker [1], Chris Clements [3], Joseph A. Tobias [4] & Manuela González-Suárez [1]

Ambitious international commitments have been made to preserve biodiversity, with the goal of preventing extinctions and maintaining ecosystem resilience, yet the efficacy of large-scale protection for preventing near-term extinctions remains unclear. Here, we used a trait-based approach to show that global actions—such as the immediate abatement of all threats across at least half of species ranges for ~10,000 bird species—will only prevent half of the projected species extinctions and functional diversity loss attributable to current and future threats in the next 100 years. Nonetheless, targeted recovery programmes prioritizing the protection of the 100 most functionally unique threatened birds could avoid 68% of projected functional diversity loss. Actions targeting 'habitat loss and degradation' will prevent the greatest number of species extinctions and proportion of functional diversity loss relative to other drivers of extinction, whereas control of 'hunting and collection' and 'disturbance and accidental mortality' would save fewer species but disproportionately boost functional richness. These findings show that conservation of avian diversity requires action partitioned across all drivers of decline and highlight the importance of understanding and mitigating the ecological impacts of species extinctions that are predicted to occur even under optimistic levels of conservation action.

Biodiversity is declining at an unprecedented rate, with implications for ecosystem functioning and the delivery of ecosystem services[1,2]. Human activity has led to widespread decline in the extent and structural condition of ecosystems and changes in community trait composition[3]. High functional diversity—the diversity of traits that describe an organism's ecological niche—has been associated with greater ecosystem functioning[4,5], more reliable ecosystem service delivery[6] and greater ecosystem resilience[7,8]. Therefore, changes in community composition could undermine the persistence of natural communities. Owing to the potential importance of functional diversity in supporting ecosystem

function and resilience[9], identifying effective measures for conserving functional diversity alongside species richness is paramount[10].

Ambitious policies and substantial conservation resources have been dedicated to halting and reversing biodiversity loss by reducing the impact of threats[11]. Programmes designed to alleviate threats at a large scale, a strategy referred to as threat abatement[12], are essential for the long-term persistence of species; however, it remains unclear to what extent they can avert imminent extinctions and functional diversity loss. Previous studies have rarely extended past isolated analyses of single threats and their impacts on species richness or

[1]Ecology and Evolutionary Biology, School of Biological Sciences, University of Reading, Reading, UK. [2]Institute of Ecology and Earth Sciences, University of Tartu, Tartu, Estonia. [3]School of Biological Sciences, University of Bristol, Bristol, UK. [4]Department of Life Sciences, Imperial College London, London, UK. ✉e-mail: kerrysmith189@gmail.com

broader syntheses of the coverage of conservation targets[13,14]. The main alternative to threat abatement strategies is direct management interventions such as breeding programmes and translocations. These measures can be effective[15,16], particularly for rare species or those that are vulnerable to human pressures[17]. However, targeted recovery programmes, including ex-situ conservation and in-situ measures to boost species survival and success, are often prohibitively expensive[18], limiting their application as a global conservation strategy[17]. Therefore, conserving bird diversity can probably only be achieved with a combination of large-scale protection through threat abatement coupled with targeted species recovery programmes[15,19]. However, the extent to which abatement can reduce the need for intensive management to boost species population and reproductive success remains unclear.

Here, we used a trait-based approach to evaluate how much biodiversity and associated ecological function could be protected in the near term, defined as the next 100 years, under different global conservation strategies. We assessed the probable success of strategies focusing on the abatement of current and future drivers of extinction and estimated whether shortfalls in efficacy can be countered through targeted species recovery programmes. We used a phylogenetic generalized linear mixed model (PGLMM) to predict species extinction risk based on threats listed by the International Union for the Conservation of Nature (IUCN), accounting for non-independence geographically and across the avian tree of life. We quantified the importance of conserving unique species, which provide a disproportionate contribution to the global diversity of form and function in birds.

## Results and Discussion

### Projected species extinctions and functional diversity loss

We projected expected bird extinctions for the next 100 years based on IUCN Red List threat data[20]. We fitted a PGLMM implemented in a Bayesian Markov chain Monte Carlo framework that predicted species assignment to Red List category with 86.8% accuracy (Supplementary Analyses), using data on threat scope and severity and including random effects describing the spatial and phylogenetic relationships among species. We then stochastically projected species extinctions based on expected probabilities of extinction for each Red List category (see Methods).

In the baseline extinction scenario, we assumed that human activity and natural threats would continue to impact bird populations as currently listed. In this scenario, we predicted that 5.2 ± 0.2% (mean ± s.d.) of the 9,873 extant birds studied would go extinct in the next 100 years (517 ± 19 species) (Fig. 1); more than three times the recorded number of bird extinctions since 1500. This figure falls within the range of previously predicted bird extinctions, ranging from 226–589 species extinctions in the next 500 years[21] to 669–738 species extinctions in the next 100 years[22]. Extinctions on this scale are expected to fundamentally alter the global bird assemblage, potentially reducing functional diversity[23,24].

To quantify projected change in functional diversity in the world's avifauna (n = 9,873), we used published data on 11 continuous morphological traits that collectively capture bird ecological niches through their well-established association with diet, dispersal and habitat[25–27]. These traits were summarized using the first three axes produced by phylogenetic principal component analysis (pPCA), which explained 87.2% of variance in the dataset (Extended Data Fig. 1), providing an overview of global avian functional diversity (Extended Data Fig. 2; see Methods). We estimated functional diversity using probabilistic hypervolumes[28], which can be applied to multidimensional data and have been shown to be less sensitive to extreme trait values than other methods, such as convex hulls[29]. We quantified the volume of trait space occupied by the current global avian assemblage (n = 9,873) as well as under future extinction scenarios (Extended Data Fig. 3). Under the baseline extinction scenario, functional diversity was projected to decrease by 3.2 ± 0.4% in the next 100 years relative to present-day functional diversity. This is probably a conservative estimate that only

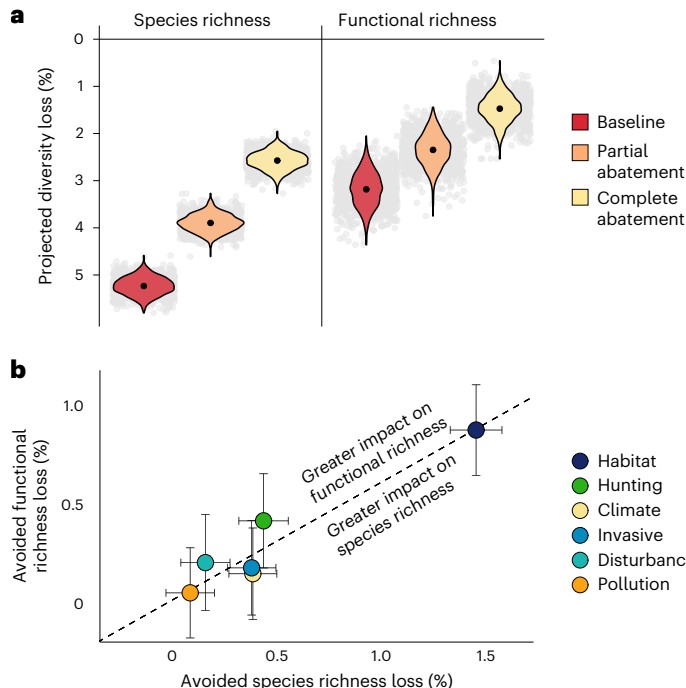

**Fig. 1 | Projected loss of avian diversity in the next 100 years. a**, Loss in species richness and functional richness under three scenarios: baseline extinction, partial abatement of all drivers of extinction and complete abatement of all drivers of extinction. Black points show mean loss across 1,000 iterations for each scenario, with variation in those points shown by their distribution (violin plots) and the individual values (grey dots). **b**, Diversity loss avoided under driver-specific complete abatement of six major drivers of extinction (circles represent the mean; error bars, 0.5 s.d.). Note that in some iterations, loss avoided could be negative, as more diversity was lost with driver-specific abatement than under the baseline scenario. The dotted diagonal line shows mean functional richness loss per species richness loss under complete abatement of all threats. Drivers above this line show greater avoidance of functional richness loss per species richness loss avoided relative to the mean across all drivers of extinction. Hunting, hunting and collection; climate, climate change and severe weather; invasive, invasive species, genes and disease; disturbance, disturbance and accidental mortality. Analyses based on 9,873 species (of which 2,087 species currently listed as Near Threatened or in threatened categories were modelled and could have reduced extinction risk in the abatement scenarios). A total of 1,000 iterations were run for each extinction scenario.

reflects loss in three-dimensional functional space and ignores internal erosion of the space[23,30]. Projected functional diversity loss varied between 2.4 ± 0.9% and 3.8 ± 0.4% when measured with two dimensions and four dimensions, respectively (Supplementary Fig. 7).

### Large-scale protection from the drivers of extinction

Threat abatement could prevent species extinctions and reduce functional diversity loss. However, it is unclear to what extent imminent biodiversity loss can be avoided, and what scale of action is required to prevent species extinctions and functional diversity loss altogether. Using our PGLMM model, we predicted how extinction risk would change under three management scenarios that reflect varying levels of threat abatement (see Methods and Fig. 2). Complete abatement involved removal of all direct drivers of extinction across the entirety of all species ranges; partial abatement involved removal of all direct drivers of extinction across at least half of all species ranges (threat spatial scopes downgraded to 'Minority < 50%'); and minimal abatement involved the removal of all direct drivers of extinction across at least 10% of all species ranges (threat spatial scopes downgraded to 'Majority 50–90%').

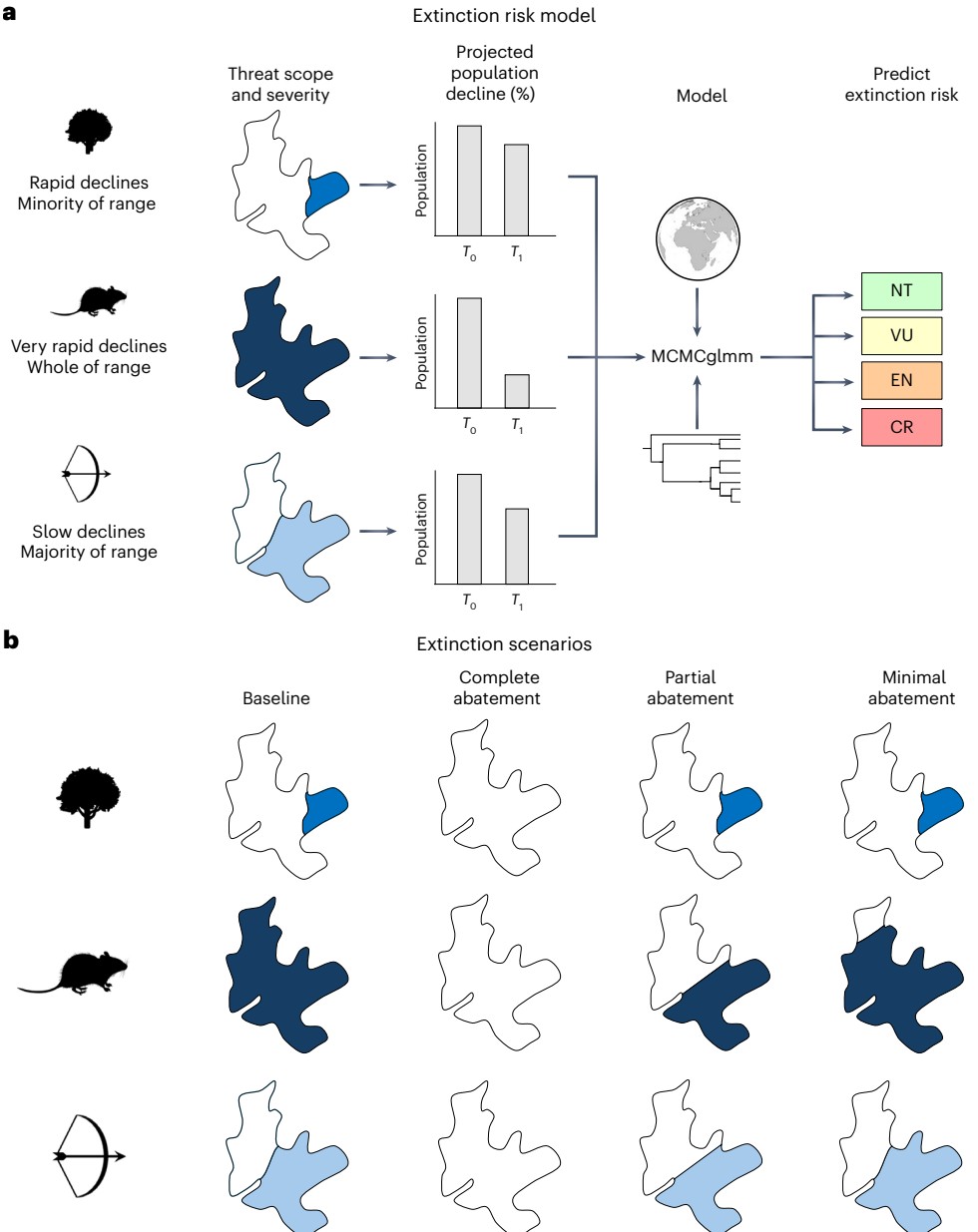

**Fig. 2 | The extinction risk model and extinction scenarios. a**, IUCN Red List data[20] on threat scope and severity were used to assign projected population decline over a 10-year period or three generations, according to previous publications[83,43]. Data on projected population decline for all species and all threats were used in an MCMCglmm to predict IUCN extinction risk category, using phylogenetic and spatial variables as random effects. NT, Near Threatened; VU, Vulnerable; EN, Endangered; CR, Critically Endangered. **b**, Four extinction scenarios were used: baseline, in which current, future and likely-to-return threats remained as listed by the IUCN[20]; complete abatement, in which threats were removed across the entirety of the species range; partial abatement, in which threats were removed from at least 50% of the species range; and minimal abatement, in which threats were removed from at least 10% of the species range. *Rattus fuscipes* (Rachel T Mason, CC0 1.0) and *Quercus robur* silhouettes from Phylopic. Globe silhouette from ClipSafari (Sev, CC0 1.0).

Under the complete abatement scenario, half of the biodiversity loss predicted under the baseline scenario could be prevented (Fig. 1 and Extended Data Table 1). However, an average loss of 2.6 ± 0.2% of species richness (254 ± 19 species) and 1.5 ± 0.3% of functional richness remained. Given that our model did not include Least Concern species (for which threat data are scarce), it could not predict assignment to the Least Concern category. However, in reality, threat abatement could result in full recovery to Least Concern. To evaluate the impact of this on projected diversity loss, we tested the effect of assuming a low extinction probability of 0.0001 for the Near Threatened category, equal to the expected for the Least Concern category ($1 \times 10^{-6}$ extinctions per species per year). We obtained similar estimates (241 ± 18 species

extinctions; 1.4 ± 0.3% functional diversity loss; Supplementary Analyses), showing that lack of assignment of species to the Least Concern category under abatement scenarios did not notably affect our results.

Some extinctions were not preventable even with complete abatement; therefore, they were not attributable to current and future drivers of extinction. These extinctions could reflect particularly vulnerable species that have high extinction risk despite being affected by few threats, as well as species that were severely affected by past threats that can no longer be managed or abated. The model captured variation in species vulnerability to extinction that was not described by threats through spatial and phylogenetic random effects. The relevance of spatial and phylogenetic variables was supported by the fact

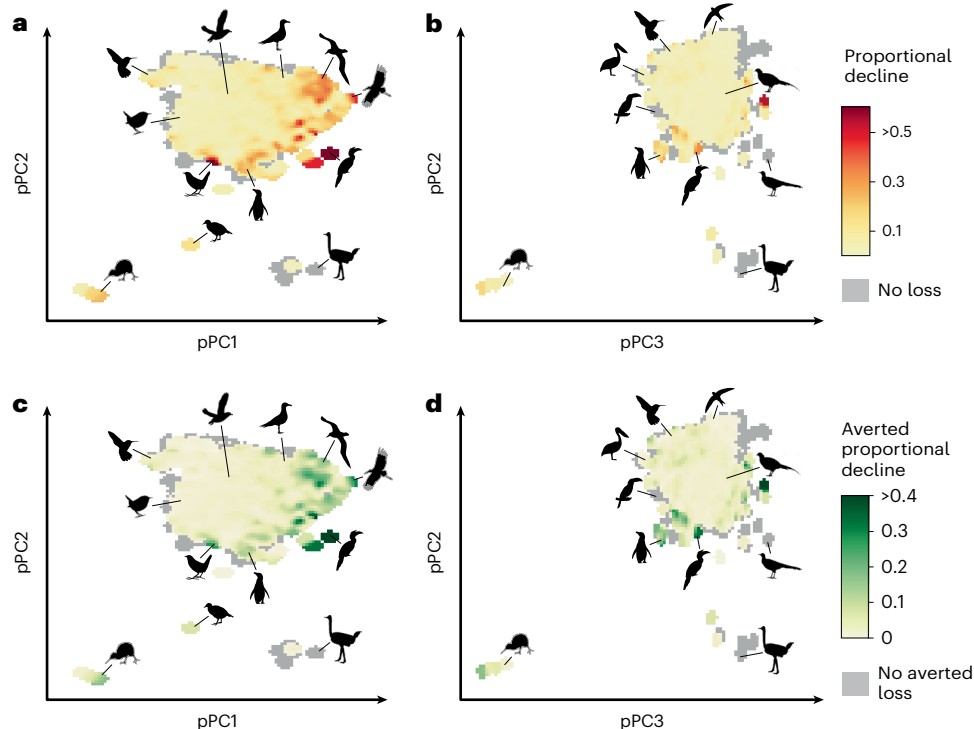

**Fig. 3 | Change in occupation of morphospace under extinction and conservation.** pPC1 is a descriptor of body size, pPC2 is associated with wing morphology and pPC3 is associated with beak and tail morphology (for trait loadings, see Supplementary Table 3). **a,b,** Predicted proportional decline in functional trait space occupation in the next 100 years under the baseline extinction scenario with respect to pPC1 and pPC2 (**a**) and pPC3 and pPC2 (**b**). **c,d,** Averted proportional decline under the complete abatement scenario for pPC1 and pPC2 (**c**) and pPC2 and pPC3 (**d**). In all panels, grey colour shows areas where no functional diversity loss was projected or where no functional diversity loss was avoided under complete abatement (fewer than five pixels in all panels). Analyses based on 9,873 species (of which 2,087 species currently listed as Near Threatened or in threatened categories were modelled and could have reduced extinction risk in the abatement scenarios). A total of 1,000 iterations were run for each extinction scenario. All silhouettes are from Phylopic. In **a** and **c** (left to right): *Apteryx* (Ferran Sayol, CC0 1.0), *Mellisuga helenae* (Steven Traver,

CC0 1.0), *Troglodytes hiemalis* (Andy Wilson, CC0 1.0), *Pteroptochos castaneus* (Ferran Sayol, CCO 1.0), *Atlantisia rogersi* (there was no silhouette of *A. rogersi* so a silhouette of *Gallirallus australis* was used instead; T. Michael Keesey and HuttyMcphoo, CC BY-SA 3.0), *Pelecanoides urinatrix* (Louis Ranjard, CC BY 3.0), *Spheniscus humboldti* (Juan Carlos Jerí, CC0 1.0), *Larus* (Ferran Sayol, CC0 1.0), Diomedeidae (Ferran Sayol, CC0 1.0), *Struthio camelus* (Darren Naish and T. Michael Keesey, CC BY 3.0), *Buceros* (Ferran Sayol, CC0 1.0) and *Leptoptilos javanicus* (T. Michael Keesey and Vaibhavcho, CC BY-SA 3.0). In **b** and **d** (left to right): *Apteryx* (Ferran Sayol, CC0 1.0), *Pelecanus* (Ferran Sayol, CC0 1.0), Ramphastidae (Federico Degrange, CC0 1.0), *S. humboldti* (Juan Carlos Jerí, CC0 1.0), *M. helenae* (Steven Traver, CC0 1.0), *Buceros* (Ferran Sayol, CC0 1.0), *Apus apus* (Ferran Sayol, CC0 1.0), *Phasianus colchicus* (Mattia Menchetti, CC0 1.0), *Menura* (T. Michael Keesey, CC0 1.0) and *S. camelus* (Darren Naish and T. Michael Keesey, CC BY 3.0).

that background extinction rates and extinctions owing to stochastic events varied by taxonomic group[31] and across space[32].

Deviations from expected extinction risk captured by spatial and phylogenetic variables could reflect fast or slow life history[33], variation in overlap with areas of high human influence[34] and isolation and connectivity[35]. These factors are expected to be important for explaining variation in both extinction risk and species recovery[36–38] and often exhibit high degrees of spatial or phylogenetic correlation[39–41]. For example, island endemics are particularly sensitive to extinction because of their small and isolated ranges[35], which may be captured through spatial random effects, and evolution of traits associated with increased extinction risk, such as flightlessness, which may be captured by phylogenetic random effects[35]. The Cebu flowerpecker (*Dicaeum quadricolor*) is a Critically Endangered species that was predicted to be at risk of extinction even under complete abatement. Like many island species, it has a very small remaining population (60–70 individuals)[42], and our analysis suggests that it is likely to go extinct without complementary measures such as habitat restoration or ex-situ conservation.

Our finding that even large-scale and ambitious actions leading to the removal of all present, future and likely-to-return threats will fail to prevent almost half of projected species extinctions challenges some of the key assumptions of global metrics used to track conservation progress. For example, the Species Threat Abatement and Restoration

(STAR) metric[43] is based on IUCN Red List data on threat scope and severity but assumes that complete threat abatement will allow the vast majority of species to be downgraded to Least Concern, an assumption that our findings did not support. Although those authors[43] acknowledge that some species may require restoration to be downgraded to Least Concern, our results suggested that many species will require conservation measures in addition to threat abatement. Even when species are not affected by current or future threats, they may still be threatened with extinction. Although we did not explicitly test the reasons for ongoing declines, they could occur because of continued population decline (particularly in populations which are no longer self-sustaining), high vulnerability to stochastic events because of small population or range size, or reduced fitness as a result of severe population decline in the past[44].

Partial abatement was somewhat effective at reducing avian diversity loss, preventing about one-quarter of projected losses (26 ± 4% of projected species extinctions and 26 ± 13% of projected functional diversity loss; Fig. 1 and Extended Data Table 1). Species that are experiencing severe declines but are affected by few threats showed the greatest reduction in extinction risk under partial abatement. The green-faced parrot finch (*Erythrura viridifacies*) and the Saint Vincent parrot (*Amazona guildingii*) responded particularly well to partial abatement, with a reduction in extinction risk that was almost as

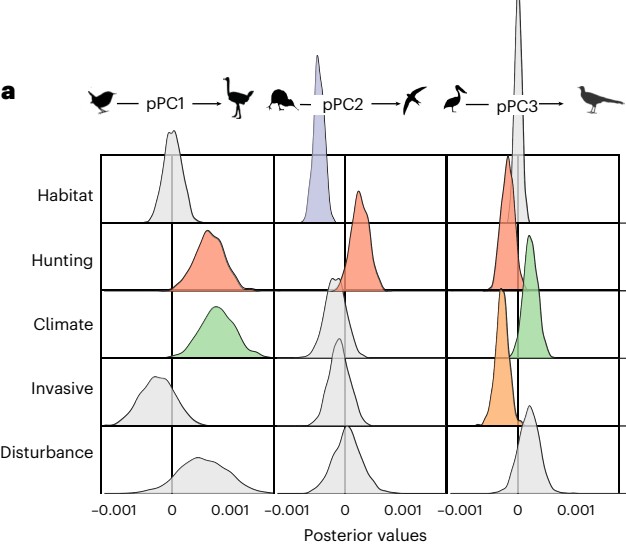

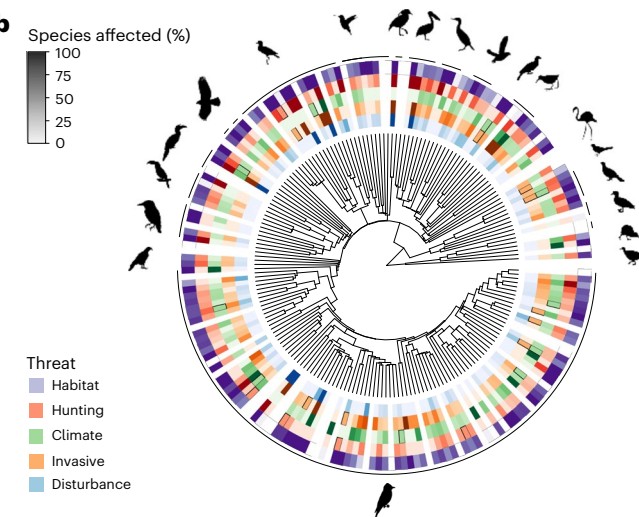

**Fig. 4 | Drivers of extinction vary across morphospace and the avian tree of life. a**, Posterior values from a multi-response MCMCglmm showing the relationships between pPC values and the frequency (from 1,000 iterations across 9,873 species) in which extinction was avoided under driver-specific complete abatement scenarios. pPC1 is a descriptor of body size, pPC2 is associated with wing morphology and pPC3 is associated with beak and tail morphology (Supplementary Table 3). Least Concern species were not included in the extinction risk model, as improvements under driver-specific complete abatement could not occur by definition. **b**, Distribution of drivers of extinction with respect to phylogeny, shown by family (9,873 species across 194 families, of which threat information was included for 2,087 Near Threatened and threatened species), with the intensity of colour reflecting the proportion of species in a family affected by each driver (families including only Least Concern or Data Deficient species are shaded white). All silhouettes are from Phylopic. In **a** (left to right): *T. hiemalis* (Andy Wilson, CC0 1.0), *S. camelus* (Darren Naish and T. Michael Keesey, CC BY 3.0), *Apteryx* (Ferran Sayol, CC0 1.0), *A. apus* (Ferran Sayol, CC0 1.0), *Pelecanus* (Ferran Sayol, CC0 1.0), *Menura* (T. Michael Keesey, CC0 1.0). In **b** (left to right): Falconiformes (Kai Caspar, CC0 1.0), Coraciiformes (Estelle Bourdon, CC0 1.0), Piciformes (Federico Degrange, CC0 1.0), Bucerotiformes (Ferran Sayol, CC0 1.0), Charadriiformes (Auckland Museum, CC BY 3.0), Apodiformes (Andy Wilson, CC0 1.0), Passeriformes (Andy Wilson, CC0 1.0), Eurypygiformes (Ferran Sayol, CC0 1.0), Pelecaniformes (Ferran Sayol, CC0 1.0), Suliformes (Juan Carlos Jerí, CC0 1.0), Procellariiformes (Louis Ranjard, CC BY 3.0), Musophagiformes (Ferran Sayol, CC0 1.0), Gruiformes (Ferran Sayol, CC0 1.0), Phoenicopteriformes (T. Michael Keesey, PDM 1.0), Mesitornithiformes (Ferran Sayol, CC0 1.0), Galliformes (Elisabeth Östman, PDM 1.0), Anseriformes (Rebecca Groom, CC BY 3.0), Apterygiformes (Ferran Sayol, CC0 1.0) and Tinamiformes (Darren Naish and T. Michael Keesey, CC BY 3.0).

great as the reduction in extinction risk under complete abatement. Although partial abatement prevented some losses, there was still a 3.9% decrease in species richness (385 ± 18 species) and a 2.3 ± 0.3% decrease in functional richness. Of the diversity loss that was attributable to the drivers of extinction (diversity loss under complete abatement), approximately half was prevented through partial abatement (50 ± 7% of species extinctions and 49 ± 26% of functional diversity loss). The minimal abatement scenario prevented only a small proportion of biodiversity loss (Extended Data Table 1). Using different traits to quantify functional diversity did not affect our conclusions (Supplementary Analyses), as we obtained similar results when we used pPCs constructed from three-dimensional scans of beak morphology[45] and when the first pPC (largely describing variation in body size) was removed.

Protecting species from the drivers of extinction does not provide a comprehensive solution to biodiversity loss in the near future without additional measures such as targeted species recovery programmes, habitat restoration and prioritization of protection in important areas[46]. This finding is consistent with previous studies assessing biodiversity impacts of future conservation and mitigation scenarios. One study[47] found that although it was possible to bend the curve of biodiversity loss with an integrated strategy, protected area management and expansion to avert the impact of habitat loss and degradation were insufficient to avoid more than 50% of projected biodiversity loss on average in biodiversity-rich regions. Similarly, another study[48] predicted that in a 2015–2050 scenario of strong land use and climate change mitigation globally, rates of biodiversity loss would decrease but biodiversity would continue to decline. Threat reduction is an essential component of tackling the biodiversity crisis and is necessary to ensure that species with healthy, stable populations do not go into decline in the future[15]. However, it is not enough. The abatement scenarios explored here represented significant management efforts with optimistic assumptions about their impact and uptake. We assumed that the drivers of extinction, and the species declines caused by these drivers, could be halted immediately and that all drivers of extinction could be alleviated, including climate change, which arguably may be difficult to mitigate with site-based protection. Even in these ambitious and optimistic scenarios, we predicted that over half of the projected species extinctions and loss of functional diversity in the next 100 years would occur anyway.

Projected loss of functional diversity was not evenly distributed across functional space (Figs. 3 and 4). Areas of trait space with high pPC1 values (generally larger birds) were predicted to show the greatest proportional losses under the baseline extinction scenario. Complete abatement was predicted to reduce loss across functional space (Fig. 3 and Extended Data Fig. 4) but was less effective in a region of high pPC1 and pPC2 values, predominantly occupied by large aquatic predators (Extended Data Fig. 5).

## Impact of six major drivers of extinction

Species' traits shape their vulnerability to human activity, but different areas of trait space are affected by different threats[49,50]. As such, abatement of drivers of extinction could have differential outcomes for functional diversity. To test this concept, we focused on six drivers of extinction (Supplementary Dataset 1) and quantified the 'maximum avoidable contribution' from each driver, describing the species and functional diversity loss avoided when the impact of current and future threats within each driver of extinction were completely removed, relative to diversity loss in the baseline scenario (see Methods). 'Habitat loss and degradation' had the highest maximum avoidable contribution, as driver-specific complete abatement was projected to avoid 1.4 ± 0.2% species richness loss (141 ± 24 species extinctions) and 0.9 ± 0.5% functional diversity loss (Fig. 1 and Extended Data Table 2). Driver-specific complete abatement of 'hunting and collection' was projected to avoid 0.4 ± 0.5% functional diversity loss (Fig. 1), almost half that of 'habitat

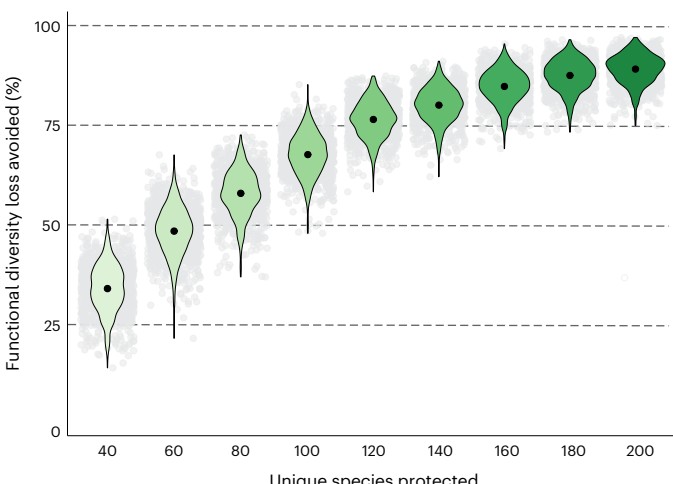

**Fig. 5 | Preventing extinction of unique threatened species reduced projected functional diversity loss.** Functional diversity loss avoided (as a percentage of projected functional diversity loss under the baseline scenario) from 1,000 iterations. Black points show mean loss avoided; violin plots show the distribution; grey points show individual values of loss avoided under each iteration. The number of unique threatened species that were prevented from going extinct ('protected') varied between 40 and 200 unique threatened species at intervals of 20 species.

loss and degradation' despite requiring action for 591 species rather than 1,658 species. Other drivers of extinction had smaller maximum avoidable contributions (Fig. 1). The relative magnitude of maximum avoidable contributions among drivers was comparable when simulating driver-specific partial abatement and driver-specific minimal abatement rather than driver-specific complete abatement.

Although assessments of individual drivers on avian functional diversity exist[13,14,51], assessments of multiple drivers simultaneously are rare and are important for quantifying the relative impact of different drivers of extinction on avian functional diversity. We found that driver-specific complete abatement of 'hunting and collection' and 'disturbance and accidental mortality' was projected to result in disproportionately high avoidance of functional diversity loss for the number of species extinctions avoided (Extended Data Fig. 6). As abatement of different drivers of extinction had different value for the preservation of functional diversity, we argue that it is necessary to consider functional diversity in conservation planning and prioritization.

To assess which species traits were vulnerable to drivers of extinction, we used a mixed-effects multi-response regression model of reduction in extinction risk under driver-specific complete abatement scenarios against values of three pPCs (see Methods). Reduction in extinction risk was quantified as the number of iterations in which species extinction was avoided in driver-specific complete abatement scenarios relative to the baseline scenario. A significant positive relationship was detected when abatement of a given driver of extinction reduced extinction risk in species with high values of a given pPC, and a significant negative relationship occurred when abatement of a given driver of extinction reduced extinction risk in species with low values of a given pPC (Fig. 4).

We found that the abatement of different drivers of extinction would benefit distinct morphologies. Birds with large body size (pPC1) were more likely to experience a reduction in extinction risk when abating 'hunting and collection' or 'climate change and severe weather' (Fig. 4; $P_{MCMC} < 0.01$ for both). Although extinction risk bias towards species with large body size is widely reported[52,53], we found that this was not the case for all threats, as there was no evidence of bias with respect to pPC1 for other drivers of extinction[49].

The bias in extinction avoidance with respect to wing morphology (pPC2) was variable across drivers of extinction. Birds with broader wings (those with low pPC2 values) were more likely to experience a reduction in extinction risk under abatement of 'habitat loss and degradation' ($P_{MCMC} < 0.01$). By contrast, birds with slender wings (high pPC2 values) were more likely to experience a reduction in extinction risk under abatement of 'hunting and collection' ($P_{MCMC} = 0.08$; values under 0.1 are treated as significant to give a one-tailed significance test of overlap with zero rather than the default two-tailed test). Our finding that extinction avoidance was more likely for species with broad wings (low values of pPC2) when abating habitat loss and degradation is consistent with recent studies that show that birds with a low hand-wing index (described by pPC2) are more sensitive to fragmentation[54] and deforestation[55].

The bias in extinction avoidance with respect to tail and beak morphology (pPC3) was also variable across drivers of extinction. Species with long tails and short beaks (low pPC3 values) were more likely to experience a reduction in extinction risk following abatement of both 'hunting and collection' and 'invasive species, and disease' ($P_{MCMC} = 0.96$ and $P_{MCMC} < 0.05$, respectively (hunting and collection was not significant when insignificant variables were removed)), whereas species with short tails and long beaks (high pPC3 values) were more likely to experience a reduction in extinction risk following abatement of 'climate change and severe weather' ($P_{MCMC} = 0.08$). Reduced extinction risk following abatement of climate change and severe weather was associated with traits involved in thermoregulation[56–59]. Birds with large body size (pPC1) but also large beaks (pPC3) were more likely to avoid extinction when climate change and severe weather was abated. As a bird's beak also influences its trophic niche[60], failing to mitigate species decline caused by climate change could have knock-on effects for trophic interactions. Variable extinction avoidance across functional trait space suggests that prioritizing threat abatement based on the magnitude of projected biodiversity loss alone is inappropriate. Reducing the impact of multiple drivers of extinction is necessary to ensure that diverse functional morphologies are conserved.

## The potential of targeted species recovery programmes

Even with ambitious action, large-scale threat abatement will not prevent all species extinctions and functional richness loss in the next 100 years. As such, targeted species recovery programmes will be needed, which we defined as in-situ and ex-situ measures to boost species survival and reproductive success that do not involve threat reduction. Here, we explored one possible approach, quantifying the benefits of preventing a small number of species extinctions targeted to reduce the loss of global functional richness. Using a metric of functional uniqueness that describes the probability of functional richness loss as a result of species extinction (Extended Data Fig. 7), we identified the most unique threatened species among the 9,873 bird species studied. Preventing the extinction of the most unique species (we tested scenarios protecting between 40 and 200 species; Fig. 5) was effective at reducing projected functional diversity loss. We found that preventing the extinction of the top 100 most unique threatened species avoided 68 ± 5% of projected functional diversity loss under the baseline scenario compared to the 26 ± 13% avoided by partial abatement of all threats for all species. By preventing the extinction of 100 species (1% of species), 2.2 ± 0.32% of functional diversity could be conserved if the most functionally unique threatened species were prioritized. This approach would require the avoidance of 37 ± 25 projected extinctions in the next 100 years. A previous publication[17] reported that 21–32 bird species have been saved from extinction by conservation efforts since 1993, suggesting that this could be an achievable goal (although ten extinctions occurred despite management).

The most functionally unique birds spanned taxonomic and ecological groups, from the Sulu hornbill (*Anthracoceros montani*) of the

southernmost Philippine islands[61] to the Ascension frigatebird (*Fregata aquila*) that patrols the Atlantic Ocean. Some were wide-ranging, like the southern royal albatross (*Diomedea epomophora*), and others are thought to only survive in one location, like Stresemann's bristle-front (*Merulaxis stresemanni*). Unique species included scavengers, such as the Andean condor (*Vultur gryphus*), nectarivores, such as the yellow-bellied sunbird-asity (*Neodrepanis hypoxantha*), vertivores, such as the Madagascar serpent-eagle (*Eutriorchis astur*) and frugivores, such as the bare-necked umbrellabird (*Cephalopterus glabricollis*). A full list of the top 200 most unique threatened birds is given in Supplementary Dataset 2.

Previous studies have found that functionally unique species are more likely to be threatened with extinction than less functionally unique species[30,62]. We provide evidence that conservation strategies for birds should prioritize functionally unique species, as has been proposed for other taxonomic groups[63,64]. In addition to their inherent value, functionally unique species are more likely to be used by humans for food, material and medicine; therefore, preventing the extinction of functionally unique species could be important for the delivery of ecosystem services[65]. Effective targeted recovery programmes that explicitly consider species uniqueness hold great potential for conserving global functional diversity as a complementary strategy to threat abatement.

## Conclusions

Both large-scale protection from the drivers of extinction and targeted species recovery programmes will be needed to prevent avian extinctions and functional diversity loss in the next 100 years. Although not effective at preventing all biodiversity loss, threat abatement is essential for ensuring that species that currently have healthy, stable populations do not fall into decline[15]. Nevertheless, our findings suggest that conservation policy should not focus solely on large-scale protection from the drivers of extinction, given that even in ambitious scenarios, only half of the projected species extinctions and functional diversity loss attributable to these drivers of extinction could be avoided.

Reducing the impact of different drivers of extinction protected distinct areas of functional trait space. Abatement of 'habitat loss and degradation' made the greatest overall contribution to avoided species extinctions and functional diversity loss, but management of 'hunting and collection' and 'disturbance and accidental mortality' prevented greater functional diversity loss proportional to the number of species projected to become extinct. Given that different areas of functional trait space were impacted by different drivers, consideration and abatement of all drivers of extinction is necessary to conserve functional diversity.

When completely or partially abating the drivers of extinction, functional richness loss was correlated with species extinctions, so reducing species extinctions is projected to reduce functional diversity loss. However, targeted species recovery programmes that focus on functionally unique species hold great potential for the conservation of functional diversity, while requiring conservation of relatively few species. By conserving the top 100 most unique threatened species, it may be possible to prevent more than two-thirds of the projected functional diversity loss through avoiding ~37 species extinctions. Although prioritization of recovery programmes offers great potential for protecting functional diversity, the ethical questions about prioritizing some species over others and the risks of overlooking ecosystem functions and services provided by other species, whether known or unknown, must be considered. If human activity continues to affect biodiversity as it is today, we project that in the next 100 years, we will lose more than three times the number of bird species as have been lost since 1500. It is, therefore, urgent that we decide which dimensions of biodiversity we wish to protect, consolidate their measurement and include them in every stage of conservation planning, monitoring and impact assessment.

## Methods

### Data collection

We used data on species morphological and geographical traits, threats and phylogenetic relationships to conduct this study. Trait data for 11,003 extant bird species[66] were obtained from AVONET[26]. Under BirdLife taxonomy, only 8% of species in AVONET have imputed data for one or more traits, and <5% of species have imputed data for more than one trait. For all study species, data on threats were obtained in June 2022 from the IUCN Red List[20] using the function *rl_threats* in the package *rredlist*[67] in R[68]. Bird species are reassessed every 4 years, causing a slight possible delay between species decline or recovery and reported change in extinction risk category[69]. Taxonomic discrepancies between AVONET and IUCN (n = 141 species) were reconciled using the function *rl_synonym*. One-to-one matches were found for all species; therefore, these taxonomic differences did not impact the results.

A maximum clade credibility phylogenetic tree was constructed from the first 1,000 trees in a previous publication[70] based on the Hackett backbone[70,71]. The authors[70] included 9,993 species in their analysis; we refer to the species nomenclature and taxonomic treatments adopted in this study as 'BirdTree taxonomy'. To enable analysis of functional diversity loss while accounting for phylogenetic covariance between species, differences between the BirdLife[66] and BirdTree[70] taxonomies were reconciled using the crosswalk provided with AVONET[26] (Supplementary Analyses). This gave 9,879 selected synonym matches between BirdLife and BirdTree (89.9% of BirdLife synonyms and 98.9% of BirdTree synonyms). Repeating analyses with all BirdLife synonyms and non-pPCA had a small impact on the percentage of projected species extinctions and functional diversity loss but did not affect our conclusions (Supplementary Fig. 6). Five species treated as Extinct in the Wild and one species listed as Extinct by IUCN[20] but not listed as extinct in the AVONET crosswalk (*Zosterops conspicillatus*) were removed from the analysis, giving a total of 9,873 species.

### Estimating functional diversity

Functional diversity quantifies the diversity of functional traits within an assemblage, defined as the measurable characteristics of an organism that influence its ecological niche[72,73]. We used 11 continuous morphological traits extracted from AVONET[26], including body mass and linear measurements of beak, wing, tail and tarsus. These traits collectively capture bird ecological niches through their association with diet, dispersal and habitat[25–27]. Using continuous morphological traits enables more fine-grained discrimination between species sharing the same ecological groups, thus providing more in-depth information about ecological variation between species than categorical traits[60]. As life history traits are more useful for explaining variation in species response to human activity rather than the ecological impacts of decline[26], they were not included in functional diversity estimations. Trait data were $log_{10}$ transformed and scaled to unit variance.

We used pPCA to reduce dimensionality. PCA produces axes that are mathematically uncorrelated but may be phylogenetically correlated if species trait data are non-independent owing to shared evolutionary history[74,75]. pPCA accounts for phylogenetic correlation between axes by removing phylogenetic covariance and calculating major axes of non-phylogenetic residual variation[75]. pPCA was carried out using the *phyl.pca* function in *phytools*[76] based on covariance and using lambda to obtain the correlation structure, which was optimized using restricted maximum likelihood. The first three pPCs described over 80% of the variance in the dataset (87.2%). Adding more pPCs described comparatively less variation (see scree plots, Extended Data Fig. 1). We therefore used the first three pPCs to summarize variation in the dataset. Using the first two or four pPCs instead, or using alternative ordination methods, did not affect our conclusions (see Supplementary Tables 4 and 5 and Supplementary Figs. 7 and 9).

We calculated global functional richness for the whole assemblage (9,873 species) using trait probability densities[28]. Firstly, a multivariate

Gaussian probability distribution was fitted for each species (Extended Data Fig. 3), in which means were provided by pPC values derived from functional trait data[26] and standard deviations were estimated using a bandwidth selector (*Hpi.diag* function from package *ks*[77]). Next, we took the sum of species probability distributions to obtain the community trait probability density (Extended Data Fig. 3). This was implemented through the *TPDsMean* and *TPDc* functions in package *TPD*[78], with 50 divisions for each pPC. We calculated functional richness using the *REND* function in package *TPD*.

### Modelling extinction risk

To predict how threat reduction affected projected avian diversity loss, we constructed a model of species extinction risk (IUCN Red List category[20]) with threats as explanatory variables and accounting for spatial and phylogenetic covariance (referred to as 'the extinction risk model'). The extinction risk model allowed us to quantify the independent contribution of each threat to extinction risk, while considering that many species were affected by multiple threats (1,978 species out of 2,104 Near Threatened and threatened species with threats listed) and comparatively few were affected by only one (126 out of 2,104 Near Threatened and threatened species with threats listed). Overlooking non-independence between threats can result in misleading findings about the relationship between threats and extinction risk as well as patterns of bias in the impacts of these drivers across species assemblages[79]. Species vulnerability to extinction, and threat prevalence (Fig. 4 and Supplementary Fig. 1), may be affected by where species live or their evolutionary history, so species data are not independent of one another[80]. We included phylogeny and spatial variables as random effects to account for non-independence among species owing to non-random baseline extinction rates[80], as well as other factors influencing extinction risk and recovery, such as small population size[32], that are not caused by population decline resulting from listed past, ongoing or future threats.

The extinction risk model was fitted using a Markov chain Monte Carlo multivariate generalized linear mixed model (MCMCglmm), predicting extinction risk for species that were listed by the IUCN Red List as Near Threatened, Vulnerable, Endangered and Critically Endangered. Threats have been described for 99% of species in these categories. MCMCglmms were fitted using the R package MCMCglmm[81]. We used 39 pseudo-continuous fixed effects, describing the expected percentage population decline over a 10-year period or three generations (from threat scope and severity data; see details below and Extended Data Table 3) caused by each threat under the second-level classification described by the IUCN[82] (for example, '1.1 Housing & Urban Areas' and '1.2 Commercial and Industrial Areas'). Threats that affected ten or fewer species were grouped with other threats (Supplementary Dataset 1).

Threats were assigned an expected population decline (over a 10-year period or three generations) based on their scope (percentage of species range affected by a threat) and severity, following previous publications[43,83] (Extended Data Table 3). If multiple threats were listed for the same threat category under the second-order classification listed by the IUCN[82], the maximum expected population decline was used. For example, *Acrocephalus familiaris* is experiencing slow, continuous declines owing to the invasive species *Schistocerca nitens* across most of its range, but it is also experiencing rapid declines caused by *Oryctolagus cuniculus* across the whole of its range. For '8.1 Invasive non-native/alien species/diseases', *A. familiaris* was assigned an expected population decline of 24%, associated with rapid declines across the whole of its range (Extended Data Table 3). We took the maximum expected population decline for second-order threats where multiple third-order threats were listed, as not all second-order threats had information on third-order threats, and without further information, it was difficult to estimate the expected population decline from multiple third-order threats. Only 16% of species–threat combinations

had more than one third-order threat listed, and when running the extinction risk model using the sum of expected population decline rather than the maximum, we found that this had minimal impacts on projected species extinctions. Threats expected to cause no decline or negligible declines across a majority or minority of a species' range had an expected population decline of zero and were effectively discarded.

Across all species, 11.58% of threat data were missing scope or severity values. Missing scope and severity data were imputed with missForest imputation (implemented through the R package *missForest*[84]) from threat type, scope, severity and timing, and incorporating phylogenetic data through eigenvectors[85]. Removing a similar proportion of values from complete data on threat timing, scope and severity to test imputation accuracy gave a mean accuracy of 82.5% (Supplementary Analyses). A total of 48 rows were also missing data on timing (needed for creating extinction scenarios; see 'Extinction scenarios' section of the Methods), and these data were imputed in the same way as scope and severity.

Phylogeny, minimum latitude, maximum latitude and centroid longitude were included in the model as random effects (Supplementary Analyses and Supplementary Fig. 1), where centroid longitude describes the longitude of the midpoint of species ranges. Spatial variables were obtained from AVONET and had been calculated from species' breeding and resident ranges, including areas where the species was coded as extant and either native or reintroduced[26]. Centroid latitude was not informative in explaining variation in extinction risk[86] (Supplementary Fig. 1) and therefore was not included as a random effect. In total, 17 species with incomplete spatial information were excluded from the extinction risk model. Species with missing spatial information were either Possibly Extinct, had no known breeding or resident range, or their range data had been redacted to protect them from trafficking risk[26]. A total of 22 species had no threat data listed and were not included in the extinction risk model. The final model was fitted for the remaining threatened and Near Threatened species ($n = 2,087$).

For fixed effects, Cauchy-scaled Gelman priors were used (with an expected value of zero), as is recommended for ordinal regressions[87]. For phylogenetic random effects, we used a chi-squared prior (expected covariance of 1, degree of belief of 1,000, mean vector of 1 and covariance matrix of 1), as this best approximates a uniform distribution, giving an uninformative prior[81,88,89]. For spatial random effects, we used parameter-expanded priors (expected covariance of 1, degree of belief of 1, mean vector of 0 and covariance matrix of 625), as they are often less informative than the default inverse-Wishart prior[90]. As it is not possible to estimate the residual variance with an ordinal response variable (extinction risk), the residual variance was fixed to 1 following previous work[90]. The model was insensitive to alternative prior specification (Supplementary Figs. 10–14). MCMC chains were run for 103,000 iterations, with a burn-in of 3,000 iterations and sampling every 100 iterations. Model convergence was assessed by parameter traces produced through the *plot* function in package MCMCglmm[81]. Non-significant fixed effects were removed iteratively, removing the least significant fixed effect, rerunning the model and repeating until only significant fixed effects remained. Significance was assessed using the pseudo *P* value ($P_{MCMC}$), estimated by MCMCglmm[81]. The pseudo *P* value is calculated as the probability that the posterior is greater or less than zero, whichever is smaller, multiplied by two[91]. A significance threshold of 0.1 was used, giving a one-tailed significance test that the posterior distribution overlaps with zero, rather than the default two-tailed test. The final model structure was:

Extinction risk category ~ X1.2 + X1.3 + X2.1 + X2.2 + X2.3 + X4.2
+X5.1 + X5.3 + X5.4 + X6.3 + X7.1 + X7.2 + X8.1 + X8.1 + X8.2
+X9.3 + X10.1 + X11.1 + X11.4 + X12.1 + random (phylogeny
+minimum latitude + maximum latitude + centroid longitude)

where X1.2 is the expected percentage population decline (Extended Data Table 3) owing to IUCN second-order threat '1.2 Commercial and Industrial Areas' and so on. X10.1 grouped threat impacts from X10.1, X10.2 and X10.3, as each threat affected fewer than ten species (see Supplementary Dataset 1 for threat codes, threat descriptions and model parameter estimates). We expected most threats to have a positive posterior mean, indicating that species affected by these threats had higher extinction risk; however, '2.2 Wood and pulp plantations' and '10 Geological events' (including X10.1, X10.2 and X10.3) had a small, negative posterior mean. Our model estimates the independent contribution of each threat to extinction risk; therefore, although it may appear that wood and pulp plantations and geological events are contributing to decline when combined with other threats, our model suggests that, in general, species threatened by wood and pulp plantations and geological events have slightly lower extinction risk.

Model accuracy was assessed as the proportion of species for which the category listed by the IUCN[20] matched the category that most frequently (across iterations) had the highest probability.

### Projected diversity loss

We used the extinction risk model to predict the probability that species belonged to each Red List category; then, using the expected probability of extinction for each Red List category, simulated extinctions that are likely to occur in the next 100 years. We used an approach explicitly incorporating uncertainty in model estimates and stochasticity in realized extinctions given extinction probabilities.

The extinction risk model returned 1,000 posterior estimates (1,000 iterations) of the probability that each species belonged to each extinction risk category (Near Threatened, Vulnerable, Endangered and Critically Endangered). Posterior estimates were extracted from the model using the function *predict2* from the postMCMCglmm package[92]. All Least Concern species and 39 Near Threatened or threatened species that were missing spatial or threat data (species not included in the extinction risk model) were assigned a probability of 1 of belonging to their Red List category as currently listed by the IUCN. Species classified as Data Deficient ($n = 41$) were conservatively assigned to the Least Concern category. For many classes, Data Deficient species are likely to be at higher risk of extinction than data-sufficient species[93]; however, this is not the case for birds[94]. Given that Data Deficient species make up a very small percentage of total species (0.4%), uncertainty over their extinction risk was expected to have a negligible impact on projected diversity loss.

An overall probability of extinction was then calculated for each species (equation 1):

$$ex_{p,s} = \sum_c \left( ex100_c \times cat_{p,c,s} \right) \tag{1}$$

where $ex_{p,s}$ is the probability of extinction in the next 100 years for species $s$ according to the posterior estimation $p$; $ex100_c$ is the assigned probability of extinction in the next 100 years of a species in IUCN extinction risk category $c$; and $cat_{p,c,s}$ is the probability that species $s$ belonged to IUCN extinction risk category $c$ according to posterior estimation $p$ or, for species not included in the extinction risk model, a probability of 1 for their Red List category as currently listed by the IUCN and a probability of 0 for all other Red List categories. Values of $ex100_c$ were based on previous work[24,29,95] and set to 0.999 for Critically Endangered, 0.667 for Endangered, 0.1 for Vulnerable, 0.01 for Near Threatened and 0.0001 for Least Concern.

Estimates of $ex_{p,s}$ were converted to a binary outcome of extinct or extant using the R function *sample*, in which the probability of being assigned extinct was $ex_{p,s}$. For each scenario, we report the mean and standard deviation in the number of extinctions across 1,000 iterations as a percentage of the total number of species included in the study (9,873 species). Functional diversity loss was estimated by removing

species projected to go extinct, calculating functional diversity across species predicted to be extant and comparing to the functional diversity of the full assemblage (9,873 species).

### Threat reduction scenarios

We estimated projected loss in species and functional diversity in the next 100 years under a baseline extinction scenario and three threat reduction scenarios: complete abatement, partial abatement and minimal abatement. Under the baseline scenario, we used the extinction risk model to predict the probability that species belonged to Red List categories, assuming that the impact of all threats remained as currently listed by IUCN[20], following the method for predicting extinctions outlined above. Under the complete abatement scenario, predictions were obtained after setting the expected population decline to zero for all threats with a timing of 'Ongoing', 'Past, Likely to Return' and 'Future'. The expected population decline of threats with a timing of 'Past, Unlikely to Return' was retained as they cannot be prevented but could still contribute to extinction risk through extinction lags (although predictions were similar if expected population decline for these threats was set to zero; Supplementary Table 6). Under the partial abatement scenario, threat impacts were altered to simulate removal of threats across at least 50% of species ranges by reassigning expected population declines for threats with a scope of 'Whole (>90%)' or 'Majority (50–90%)' to the decline expected for a scope of 'Minority (<50%)' (Extended Data Table 3). Under the minimal abatement scenario, threat impacts were altered to simulate the removal of threats across at least 10% of species ranges by reassigning expected population declines of threats with a scope of 'Whole (>90%)' to expected decline for a scope of 'Majority (50–90%)'. Least Concern species were not included in the extinction risk model, given that this is the lowest risk category and, by definition, could not show a reduction in extinction risk under abatement scenarios.

We used Cohen's D to quantify the effect size of the difference in means of diversity loss under each threat reduction scenario, divided by their pooled standard error. We do not report P values, as the sample size (number of iterations) could be increased easily, reducing standard error and giving significance even with very small differences in means, leading to type 1 errors.

### Vulnerable bird morphologies and hotspots of conservation potential

To determine the bird morphologies with the greatest extinction risk, we plotted the loss in density of trait space occupation under the baseline scenario using trait probability densities. We constructed a community probability distribution by taking the sum of species probability distributions, in which each species probability distribution was given a weight between 0 and 1,000, and describing the number of iterations in which they did not go extinct under the baseline scenario. The resulting community probability distribution was rescaled to show absolute rather than relative change in the density of occupation of trait space. For each cell, we compared the density of occupation of the whole assemblage to the density of occupation in iterations of the baseline extinction scenario. The loss in density was calculated as a percentage of the density of trait space occupation in the full assemblage for which all species had a value of 1,000, indicating no extinction (Extended Data Fig. 4). Areas of trait space with high loss in density (approaching 100%) were predicted to have a high risk of extinction.

To determine the bird morphologies with the greatest reduction in extinction risk under complete abatement of all threats (protection in Extended Data Fig. 4), we plotted the density of trait space occupation under the complete abatement scenario using trait probability densities, whereby all species were given a weight according to the number of iterations in which extinction was avoided in the complete abatement relative to the baseline scenarios. We also plotted the loss in density of trait space occupation that was not avoided, as a proportion

of density loss under complete abatement. For plotting loss in density under the baseline scenario, loss in density averted under the complete abatement scenario and loss in density not averted under the complete abatement scenario, pPCs were divided into 100 bins to provide high plot resolution. To facilitate visualization, this process was carried out in two dimensions (two pPCs at a time).

### Impact of six major drivers of extinction

We aimed to assess the independent contribution of six major drivers of extinction to projected avian diversity loss. Threats in the extinction risk model were grouped into six 'drivers of extinction': 'habitat loss and degradation', 'hunting and collection', 'climate change and severe weather', 'disturbance and accidental mortality', 'invasive species and disease' and 'pollution' (see Supplementary Dataset 1). Geological events and threats described as 'other' were grouped into an 'Other' category. Although these other threats were included in the model of extinction risk and their impact was accounted for when assessing the impact of all drivers of extinction together, we did not assess their impact individually.

We projected species and functional diversity loss under driver-specific complete abatement scenarios in which the impact of threats in a given driver of extinction with a timing of 'Ongoing', 'Past, Likely to Return' and 'Future' were removed by setting their expected population decline to zero. The 'maximum avoidable contribution' was calculated as the difference in predicted species and functional diversity loss between the baseline scenario and the driver-specific complete abatement scenario.

We used Cohen's $D$ to quantify the effect size of the difference in means of diversity loss under the baseline and driver-specific complete abatement scenarios, divided by their pooled standard error. As before, $P$ values were not reported.

To determine the severity of functional richness loss under a given driver of extinction in relation to the number of species projected to become extinct, we used a linear mixed-effects model to describe the functional richness loss avoided, using the number of species extinctions avoided and the driver of extinction (categorical) as explanatory variables. Model iteration was used as a random effect to account for non-independence in calculated differences in functional diversity loss and species extinctions from baseline to threat reduction scenarios. The dredge function from the *MuMIn* package[96] was used to identify the best model from all combinations of explanatory variables and an interaction between species extinctions and driver of extinction. The best model included the number of species extinctions avoided and the driver of extinction but not the interaction between species extinctions avoided and the driver of extinction.

### Biases in extinction avoidance

We aimed to find whether abatement of different drivers of extinction could avoid extinction in different regions of trait space. We constructed a multi-response MCMCglmm with the three pPC values for each species as response variables (including all 9,873 species studied) and the number of iterations in which extinction was avoided under driver-specific complete abatement for each driver of extinction as explanatory variables, accounting for phylogenetic covariance. Pollution was not included, as a driver-specific complete abatement of pollution had a negligible impact on functional richness loss. The residual structure was allowed to vary for each response variable. Random effect priors were provided as a diagonal matrix, with an expected covariance between response variables of zero and expected variance within response variables of 1. The degree of belief parameter for the random effect prior was 2 (ref. 97). The expected value of fixed effects and the theta-scale parameter were zero, with a covariance matrix in which the expected covariance between fixed effects was zero and the expected variance within fixed effects was $1 \times 10^{10}$. Posterior distributions were plotted, and a significance threshold of 5% overlap with zero was used ($P_{MCMC} = 0.1$).

### The potential of targeted species recovery programmes

To identify the potential value of preventing the extinction of the most functionally unique threatened species, we calculated functional uniqueness for all species (Extended Data Fig. 7). For each species, we calculated the proportion of the density of the community probability distribution that was occupied by the species probability distribution for each grid cell in which the species probability density was greater than zero. We calculated the mean proportion for each species across grid cells in which the species probability distribution was greater than zero, giving greater weight to cells in which the species distribution had greater probability (higher density). The maximum uniqueness possible was 1, indicating that a particular species' probability distribution had no overlap with the probability distribution of any other species included in the analysis. Uniqueness tended towards a lower limit of zero, indicating that the species probability distribution had high overlap with many other species probability distributions, and many other species occupied the same area of trait space.

We then identified the most unique and threatened (listed as Vulnerable, Endangered or Critically Endangered) species as potential targets for action. We compared functional richness loss in the baseline scenario with functional richness loss when extinctions of the most unique threatened species were prevented. We explored the consequences of avoiding extinction for the top 40–200 most unique threatened species (in intervals of 20 species). A total of 1,000 posterior estimates were obtained for each extinction scenario.

### Reporting summary

Further information on research design is available in the Nature Portfolio Reporting Summary linked to this article.

### Data availability

AVONET data on morphological, ecological and geographical traits for all birds[26] is available for use under a Creative Commons licence (CC BY 4.0): https://doi.org/10.6084/m9.figshare.16586228. v7. Data on IUCN extinction risk categories and threats affecting each species are available from the IUCN Red List[20] and can be accessed through the package rredlist[67]. Information on the terms of use of IUCN Red List data can be found at https://www.iucnredlist.org/terms/terms-of-use. Supplementary datasets are provided at https://doi.org/10.6084/m9.figshare.26067970 under a Creative Commons license (CC BY 4.0).

### Code availability

The code used for figures and analyses is provided at https://doi.org/10.6084/m9.figshare.26067970 under a Creative Commons license (CC BY 4.0).

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

## Acknowledgements

K.S. acknowledges PhD studentship funding from the SCENARIO NERC Doctoral Training Partnership grant NE/S007261/1. C.C. was funded by the Natural Environment Research Council (grants NE/Z000130/1 and NE/T006579/1). C.P.C. was funded by the Estonian Research Council (grants PRG2142 and MOBERC100) and the European Union (ERC, PLECTRUM, 101126117).

## Author contributions

K.S., C.V., C.P.C., J.B., C.C., J.A.T. and M.G.S. conceived and designed the analyses. K.S., C.V., J.B. and M.G.S. conducted analyses. C.P.C. provided analysis tools. J.A.T. provided data. K.S., C.V., C.P.C., J.B., C.C., J.A.T. and M.G.S. wrote and edited the manuscript.

## Competing interests

The authors declare no competing interests.

## Additional information

**Extended data** is available for this paper at https://doi.org/10.1038/s41559-025-02746-z.

**Correspondence and requests for materials** should be addressed to Kerry Stewart.

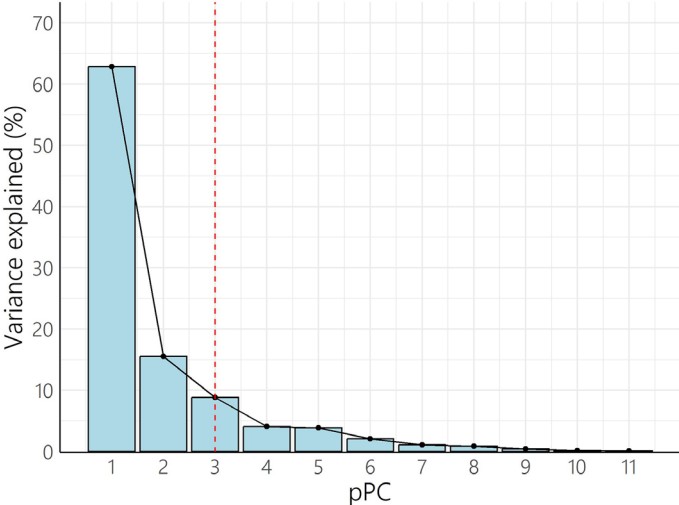

**Extended Data Fig. 1 | Variance explained by phylogenetic principal components (pPC).** Red dotted line indicates elbow after which adding additional phylogenetic principal components would explain little additional variance ($n$=9873 species).

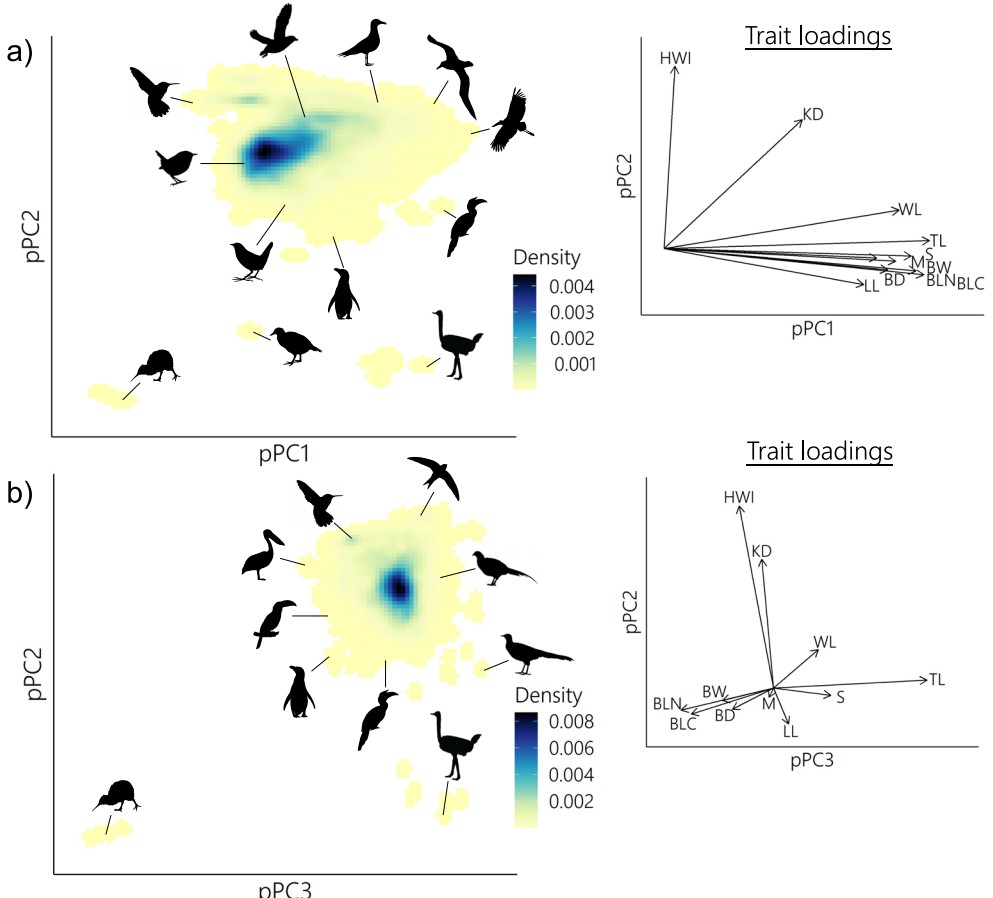

**Extended Data Fig. 2 | Occupation of morphospace in extant birds.** Shown on 2-dimensional plane with respect to **a**, pPC1 and pPC2, and **b**, pPC2 and pPC3 with trait loadings (also see Supplementary Table 3). BD = beak depth, BLC = beak length (culmen), BLN = beak length (nares), BW = beak width, HWI = hand-wing index, KD = Kipp's distance, LL = tarsus length, M = body mass, S = first secondary length, TL = tail length, WL = wing length (*n*=9873 species). All silhouettes from Phylopic. Panel a left to right: *Apteryx* (Ferran Sayol, CC0 1.0), *Mellisuga helenae* (Steven Traver, CC0 1.0), *Troglodytes hiemalis* (Andy Wilson, CC0 1.0), *Pteroptochos castaneus* (Ferran Sayol, CCO 1.0), *Pelecanoides urinatrix* (Louis Ranjard, CC BY 3.0), *Gallirallus australis* (there was no silhouette of *Atlantsia rogersi* so a silhouette of *Gallirallus australis* was used instead, T. Michael Keesey,

CC BY-SA 3.0), *Spheniscus humboldti* (Juan Carlos Jerí, CC0 1.0), *Larus* (Ferran Sayol, CC0 1.0), *Diomedeidae* (Ferran Sayol, CC0 1.0), *Struthio camelus* (Darren Naish and T. Michael Keesey, CC BY 3.0), *Buceros* (Ferran Sayol, CC0 1.0), *Leptoptilos javanicus* (T. Michael Keesey, CC BY-SA 3.0). Panel b left to right: *Apteryx* (Ferran Sayol, CC0 1.0), *Pelecanus* (Ferran Sayol, CC0 1.0), *Ramphastidae* (FJDegrange, CC0 1.0), *Spheniscus humboldti* (Juan Carlos Jerí, CC0 1.0), *Mellisuga helenae* (Steven Traver, CC0 1.0), *Buceros* (Ferran Sayol, CC0 1.0), *Apus apus* (Ferran Sayol, CC0 1.0), *Struthio camelus* (Darren Naish and T. Michael Keesey, CC BY 3.0), *Phasianus colchicus* (Mattia Menchetti, CC0 1.0), *Menura* (T. Michael Keesey, CC0 1.0).

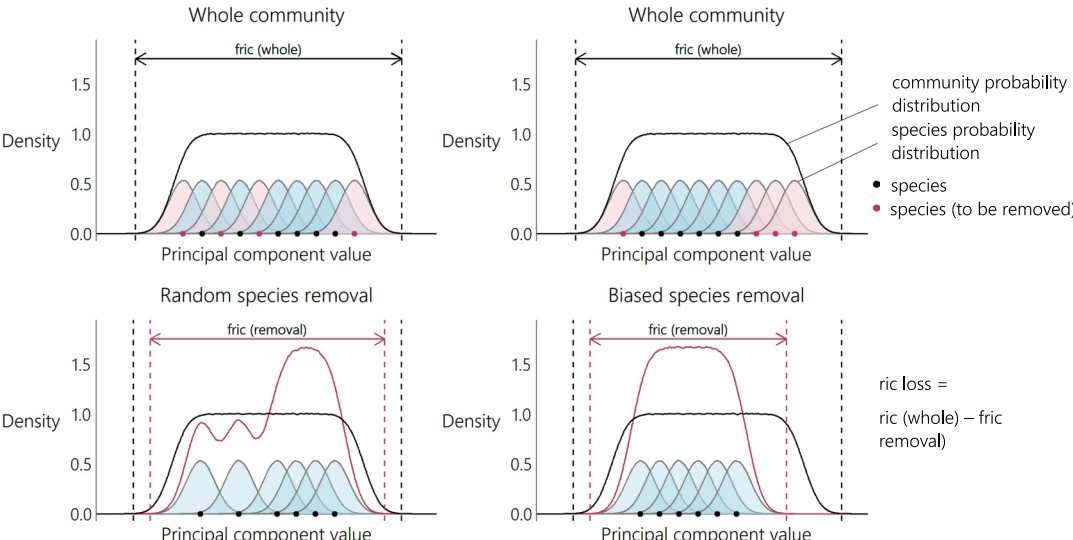

**Extended Data Fig. 3 | Estimating functional richness loss.** When species removal is biased with respect to species traits (principal component values) functional richness loss is greater. Shown in one dimension for simplicity, functional richness was calculated in three-dimensional trait space composed of the first three phylogenetic principal components.

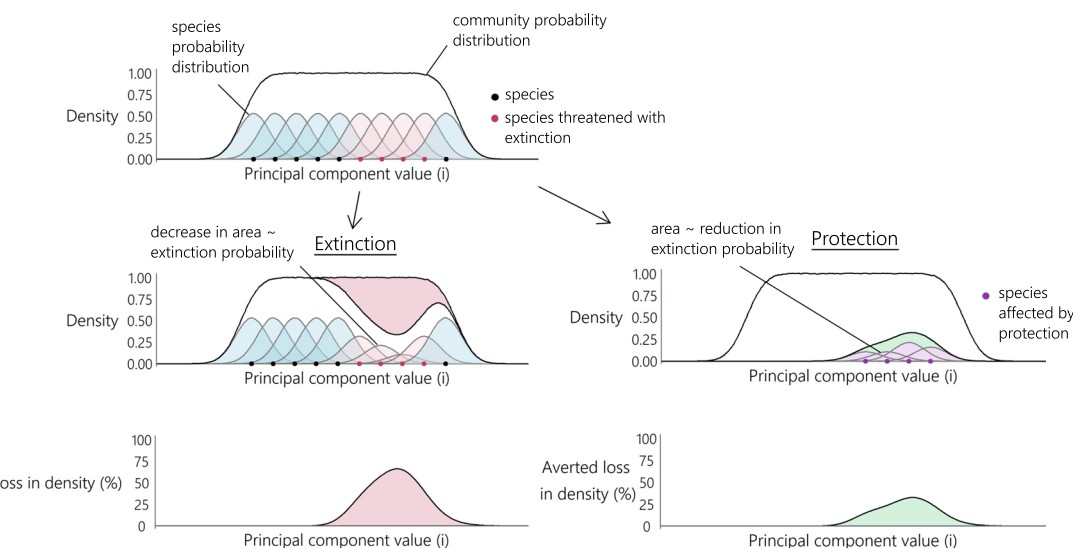

**Extended Data Fig. 4 | Estimating change in density in morphospace under extinction and conservation.** Loss in density of morphospace was calculated using the baseline extinction scenario, and averted loss in density of morphospace was calculated using the complete abatement scenario.

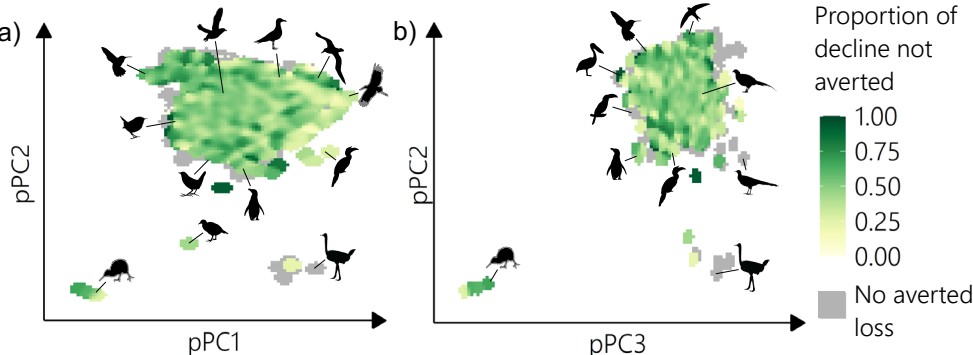

**Extended Data Fig. 5 | Proportion of decline in density of morphospace occupation that was not averted by complete abatement.** Plotted for **a**) pPC1 and pPC2, and **b**) pPC2 and pPC3. Grey shows areas where no functional diversity loss was projected, or where no functional diversity loss was avoided under complete abatement (only three pixels in panel a and two pixels in panel b). Analyses based on 9873 species (of which 2087 species currently listed as Near Threatened or in threatened categories were modelled and could have reduced extinction risk in the abatement scenarios). 1000 iterations were run for each extinction scenario. All silhouettes from Phylopic. Panel a left to right: *Apteryx* (Ferran Sayol, CC0 1.0), *Mellisuga helenae* (Steven Traver, CC0 1.0), *Troglodytes hiemalis* (Andy Wilson, CC0 1.0), *Pteroptochos castaneus* (Ferran Sayol, CC0 1.0), *Atlantisia rogersi* (there was no silhouette of *Atlantsia rogersi* so a silhouette of *Gallirallus australis* was used instead, T. Michael Keesey and HuttyMcphoo, CC BY-SA 3.0), *Pelecanoides urinatrix* (Louis Ranjard, CC BY 3.0), *Spheniscus humboldti* (Juan Carlos Jerí, CC0 1.0), *Larus* (Ferran Sayol, CC0 1.0), *Diomedeidae* (Ferran Sayol, CC0 1.0), *Struthio camelus* (Darren Naish and T. Michael Keesey, CC BY 3.0), *Buceros* (Ferran Sayol, CC0 1.0), *Leptoptilos javanicus* (T. Michael Keesey and Vaibhavcho, CC BY-SA 3.0). Panel b left to right: *Apteryx* (Ferran Sayol, CC0 1.0), *Pelecanus* (Ferran Sayol, CC0 1.0), *Ramphastidae* (Federico Degrange, CC0 1.0), *Spheniscus humboldti* (Juan Carlos Jerí, CC0 1.0), *Mellisuga helenae* (Steven Traver, CC0 1.0), *Buceros* (Ferran Sayol, CC0 1.0), *Apus apus* (Ferran Sayol, CC0 1.0), *Phasianus colchicus* (Mattia Menchetti, CC0 1.0), *Menura* (T. Michael Keesey, CC0 1.0), *Struthio camelus* (Darren Naish and T. Michael Keesey, CC BY 3.0).

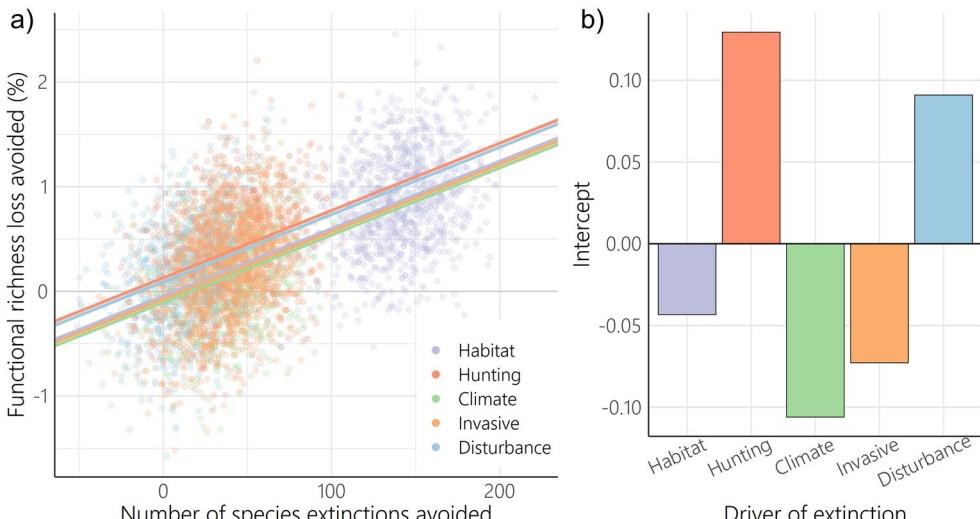

**Extended Data Fig. 6 | Abatement of hunting and collection, and disturbance and accidental mortality provides disproportionate benefits for functional richness. a**, Number of species extinctions avoided under driver-specific complete abatement against functional richness loss avoided (% of functional richness of full assemblage) as described by a linear mixed effects model including number of species extinctions avoided and driver of extinction as fixed effects, and iteration number as a random effect. **b**, Intercepts of linear mixed effect model of number of species extinctions avoided against functional richness loss for each driver of extinction showing the proportional impact of each direct driver of extinction given the number of species extinctions. Habitat = habitat loss and degradation, Hunting = hunting and collection, Climate= climate change and severe weather, Invasive = invasive species and disease, Disturbance = disturbance and accidental mortality. Pollution was not included as it made a negligible contribution to functional richness loss (see Extended Data Table 3) (*n* = 5000, 1000 iterations for each extinction scenario).

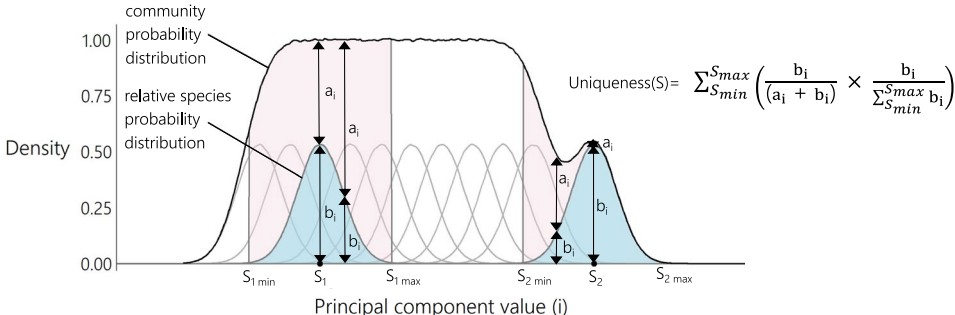

**Extended Data Fig. 7 | Functional uniqueness calculation.** Functional uniqueness describes the proportion of the community probability distribution that was composed of the species probability distribution. Proportions were summed across cells in which the species probability distribution was greater than 0, with a weight proportional to the probability of species occurrence in that cell, indicated by the height of the species probability distribution.

**Extended Data Table 1 | Predicted effect (Cohen's D) of management scenarios on diversity loss**

|  | Metric | Baseline | Complete abatement | Partial abatement |
|---|---|---|---|---|
| Complete abatement | SP | 13.92*** |  |  |
|  | FR | 4.93*** |  |  |
| Partial abatement | SP | 7.15*** | 7.00*** |  |
|  | FR | 2.25*** | 2.64*** |  |
| Minimal abatement | SP | 0.63** | 13.35*** | 6.54*** |
|  | FR | 0.02 | 4.84*** | 2.20*** |

1000 iterations for each extinction scenario, SP = number of species extinctions and FR= percentage functional richness loss (* = small to medium impact [0.2 < Cohens D < 0.5], ** = medium to large impact [0.5 < Cohens D < 0.8], *** = large impact [Cohens D > 0.8][98]).

**Extended Data Table 2 | Predicted effect (Cohen's D) of driver-specific complete abatement**

| Driver of extinction | Species richness | Functional richness |
|---|---|---|
| Habitat loss and degradation | 7.37*** | 2.29*** |
| Hunting and collection | 2.23*** | 1.04*** |
| Climate change and severe weather | 1.98*** | 0.35* |
| Invasive species and disease | 1.96*** | 0.42* |
| Disturbance and accidental mortality | 0.83*** | 0.49* |
| Pollution | 0.44* | 0.10 |

1000 iterations for each extinction scenario (* = small to medium impact [0.2 < Cohens D < 0.5], ** = medium to large impact [0.5 < Cohens D < 0.8], *** = large impact [Cohens D > 0.8][98]).

**Extended Data Table 3 | Expected population decline over a 10-year period or three generations (%) from scope and severity scores (from Garnett et al.[83] and Mair et al.[43])**

| | | Severity | | | | | |
|---|---|---|---|---|---|---|---|
| | | Very rapid declines | Rapid declines | Slow, significant declines | Negligible declines | No decline | Causing/could cause fluctuations |
| **Scope** | Whole (>90%) | 63 | 24 | 10 | 1 | 0 | 10 |
| | Majority (50-90%) | 52 | 18 | 9 | 0 | 0 | 9 |
| | Minority (<50%) | 24 | 7 | 5 | 0 | 0 | 5 |

Scope describes the percentage of a species range covered by a threat.

# Reporting Summary

## Statistics

For all statistical analyses, confirm that the following items are present in the figure legend, table legend, main text, or Methods section.

| n/a | Confirmed | |
|---|---|---|
| ☐ | ☒ | The exact sample size (*n*) for each experimental group/condition, given as a discrete number and unit of measurement |
| ☐ | ☒ | A statement on whether measurements were taken from distinct samples or whether the same sample was measured repeatedly |
| ☐ | ☒ | The statistical test(s) used AND whether they are one- or two-sided *Only common tests should be described solely by name; describe more complex techniques in the Methods section.* |
| ☐ | ☒ | A description of all covariates tested |
| ☐ | ☒ | A description of any assumptions or corrections, such as tests of normality and adjustment for multiple comparisons |
| ☐ | ☒ | A full description of the statistical parameters including central tendency (e.g. means) or other basic estimates (e.g. regression coefficient) AND variation (e.g. standard deviation) or associated estimates of uncertainty (e.g. confidence intervals) |
| ☐ | ☒ | For null hypothesis testing, the test statistic (e.g. *F*, *t*, *r*) with confidence intervals, effect sizes, degrees of freedom and *P* value noted *Give P values as exact values whenever suitable.* |
| ☐ | ☒ | For Bayesian analysis, information on the choice of priors and Markov chain Monte Carlo settings |
| ☒ | ☐ | For hierarchical and complex designs, identification of the appropriate level for tests and full reporting of outcomes |
| ☐ | ☒ | Estimates of effect sizes (e.g. Cohen's *d*, Pearson's *r*), indicating how they were calculated |

*Our web collection on statistics for biologists contains articles on many of the points above.*

## Software and code

Policy information about availability of computer code

| Data collection | No software was used to collect data for this study. Data was downloaded from publicly available sources cited in the manuscript and detailed in the data availability statement. |
|---|---|
| Data analysis | R version 4.2.2 was used to conduct analyses. R packages used in the analyses (with their versions) are cited in the manuscript. Code used to run analyses and produce figures is provided in the FigShare repository: https://doi.org/10.6084/m9.figshare.26067970, as detailed in the Code Availability statement in the manuscript. |

For manuscripts utilizing custom algorithms or software that are central to the research but not yet described in published literature, software must be made available to editors and reviewers. We strongly encourage code deposition in a community repository (e.g. GitHub). See the Nature Portfolio guidelines for submitting code & software for further information.

## Data

Policy information about availability of data

All manuscripts must include a data availability statement. This statement should provide the following information, where applicable:
- Accession codes, unique identifiers, or web links for publicly available datasets
- A description of any restrictions on data availability
- For clinical datasets or third party data, please ensure that the statement adheres to our policy

AVONET data on morphological, ecological and geographical traits for all birds is available for use under the creative commons licence (CC BY 4.0): https://

# Research involving human participants, their data, or biological material

Policy information about studies with human participants or human data. See also policy information about sex, gender (identity/presentation), and sexual orientation and race, ethnicity and racism.

| | |
|---|---|
| Reporting on sex and gender | n/a |
| Reporting on race, ethnicity, or other socially relevant groupings | n/a |
| Population characteristics | n/a |
| Recruitment | n/a |
| Ethics oversight | n/a |

Note that full information on the approval of the study protocol must also be provided in the manuscript.

# Field-specific reporting

Please select the one below that is the best fit for your research. If you are not sure, read the appropriate sections before making your selection.

☐ Life sciences    ☐ Behavioural & social sciences    ☒ Ecological, evolutionary & environmental sciences

For a reference copy of the document with all sections, see nature.com/documents/nr-reporting-summary-flat.pdf

# Ecological, evolutionary & environmental sciences study design

All studies must disclose on these points even when the disclosure is negative.

| | |
|---|---|
| Study description | Our study uses existing datasets to compare outcomes between biodiversity metrics (functional richness and species richness) under different conservations scenarios for extant birds. Extinction risk was modelled using a phylogenetic generalized linear mixed model using expected population decline from threats (as listed by the IUCN [2022]) as fixed effects, and phylogeny and spatial variables as random effects. The extinction risk model was used to apply a range of extinction scenarios, describing varying degrees of threat reduction. 9873 extant birds were included representing 89% of the extant avian assemblage. |
| Research sample | The sample-size was determined by the number of species for which there was phylogenetic, spatial and morphological data, accounting for mismatches between taxonomies so as to avoid repeating data between synonyms. Our sample includes 89% of extant birds globally. Phylogenetic data were obtained from Jetz et al. (2012), and morphological data and spatial data were obtained from AVONET (Tobias et al., 2021). Data on threats affecting bird species were obtained from the IUCN Red List (IUCN, 2022). |
| Sampling strategy | All birds for which we had phylogenetic, spatial and morphological data were included. We accounted for mismatches between taxonomies so as to avoid repeating data between synonyms. This is described in detail in the Methods and Supplementary Information provided with the manuscript. |
| Data collection | We used publicly available datasets, detailed in the Methods and Data Availability statement of the manuscript. |
| Timing and spatial scale | Global. Present and project 100 years into the future. |
| Data exclusions | A matching procedure was used to translate between BirdLife (threat and morphological data, used by Tobias et al. [2021] and IUCN [2022]) and BirdTree (used by Jetz et al., 2012) taxonomies. This was needed to enable analysis of functional diversity loss whilst accounting for phylogenetic covariance between species. Of the 11 003 extant birds listed in the BirdLife taxonomy in 2023, 9873 synonyms were included. The impact of including all synonyms has been assessed and is detailed in the Supplementary Information. |
| Reproducibility | We have made our code available via a Figshare repository (https://doi.org/10.6084/m9.figshare.26067970). We have stated packages (including their versions) used, and included information on arguments used, and priors used for Markov-Chain Monte Carlo models. In the code we provide seed values where appropriate so that the analyses can be reproduced exactly. |
| Randomization | This was not an experimental study so no samples, organisms or participants were used. |
| Blinding | This was not an experimental study so no blinding was used. |

Did the study involve field work?    ☐ Yes    ☒ No

# Reporting for specific materials, systems and methods

We require information from authors about some types of materials, experimental systems and methods used in many studies. Here, indicate whether each material, system or method listed is relevant to your study. If you are not sure if a list item applies to your research, read the appropriate section before selecting a response.

## Materials & experimental systems

| n/a | Involved in the study |
|-----|----------------------|
| ☒ ☐ | Antibodies |
| ☒ ☐ | Eukaryotic cell lines |
| ☒ ☐ | Palaeontology and archaeology |
| ☒ ☐ | Animals and other organisms |
| ☒ ☐ | Clinical data |
| ☒ ☐ | Dual use research of concern |
| ☒ ☐ | Plants |

## Methods

| n/a | Involved in the study |
|-----|----------------------|
| ☒ ☐ | ChIP-seq |
| ☒ ☐ | Flow cytometry |
| ☒ ☐ | MRI-based neuroimaging |

## Plants

Seed stocks
> n/a

Novel plant genotypes
> n/a

Authentication
> n/a

