## [Peer Review File · Nature Ecology & Evolution]

Threat reduction must be coupled with targeted recovery programmes to conserve global bird diversity

Corresponding Author: Ms Kerry Stewart

Version 0:

Decision Letter:

21st August 2024

Dear Dr Stewart,

Your Article, "Threat reduction must be coupled with targeted recovery programmes to conserve global bird diversity" has now been seen by four reviewers. You will see from their comments copied below that while they find your work of considerable potential interest, they have raised quite substantial concerns that must be addressed. In light of these comments, we cannot accept the manuscript for publication, but would be very interested in considering a revised version that addresses these serious concerns.

We hope you will find the reviewers' comments useful as you decide how to proceed. If you wish to submit a substantially revised manuscript, please bear in mind that we will be reluctant to approach the reviewers again in the absence of major revisions.

If you choose to revise your manuscript taking into account all reviewer and editor comments, please highlight all changes in the manuscript text file [OPTIONAL: in Microsoft Word format].

* Include a "Response to reviewers" document detailing, point-by-point, how you addressed each referee comment. If no action was taken to address a point, you must provide a compelling argument. This response will be sent back to the referees along with the revised manuscript.

* If you have not done so already we suggest that you begin to revise your manuscript so that it conforms to our Article format instructions at <http://www.nature.com/natecolevol/info/final-submission>. Refer also to any guidelines provided in this letter.

Link Redacted

If you wish to submit a suitably revised manuscript we would hope to receive it within 6 months. If you cannot send it within this time, please let us know. We will be happy to consider your revision so long as nothing similar has been accepted for publication at Nature Ecology & Evolution or published elsewhere.

Nature Ecology & Evolution is committed to improving transparency in authorship. As part of our efforts in this direction, we

are now requesting that all authors identified as 'corresponding author' on published papers create and link their Open Researcher and Contributor Identifier (ORCID) with their account on the Manuscript Tracking System (MTS), prior to acceptance. This applies to primary research papers only. ORCID helps the scientific community achieve unambiguous attribution of all scholarly contributions. You can create and link your ORCID from the home page of the MTS by clicking on 'Modify my Springer Nature account'. For more information please visit www.springernature.com/orcid.

Thank you for the opportunity to review your work.

[redacted]

Reviewer expertise:

Reviewer #1: phylogenetic diversity, extinction risk, birds

Reviewer #2: threat abatement scenarios

Reviewer #3: extinction risk, species recovery

Reviewer #4: biodiversity distribution and management, birds

Reviewers' comments:

Reviewer #1 (Remarks to the Author):

This is a thought-provoking and interesting piece of research that highlights the different impacts of threats and potential threat abatement scenarios on the conservation of two facets of biodiversity: species richness and 'functional diversity'. The work is very well composed, with clear English and visualisations, and there is a huge number of results and outputs to consume (perhaps bordering too much for a single paper!). There are, however, some fundamental flaws in this manuscript--related to the omission of key trait data and the incorrect estimation of Red List categories--that I believe are currently prohibitive to its publication if left unaddressed, which I outline below.

Functional diversity

My first issue centres on the use of the term 'functional diversity' (FD) in this manuscript. While the analyses using the morphospace are interesting and appropriate, it feels like a stretch to call the morphospace 'FD' whilst it lacks any information on other key 'functional' traits, particularly related to fecundity and longevity. This is a pronounced omission given that the work focuses on short term extinction scenarios—scenarios that include criteria related to generation length—where one would expect traits such as fecundity, age at maturity, and lifespan to play an important role in the fate of species following threat abatement.

For example, when looking at the extinction risk to the FD of birds using traits related to fecundity and longevity, Carmona et al. (2021, *Science Advances*: <https://doi.org/10.1126/sciadv.abf2675>) found that "extinction risk is not randomly distributed but localized in certain areas of the functional space occupied by species with large size, slow pace of life, or low fecundity", and "species with later fledging ages, longer incubation times, and larger sizes having up to six times higher threat risk than smaller species with faster breeding times". However, searching through the manuscript there is zero mention of 'fecundity', 'reproduction', or 'longevity' despite their well-established relevance to extinction risk. I think it is important to ideally include additional analyses including these data for birds to show the effects on the results, or at a minimum address their absence in the text, by exchanging the term 'FD' for 'morphospace' or including discussion around why these traits were not deemed important to characterise conservation-relevant functional indices for birds under extinction scenarios.

Threat data and extinction scenarios

The threat work is very thorough and involved, but there are some issues that require at a minimum clarification and at a maximum rethinking entirely. First, it feels intuitively circular to use threat data that was used to inform the assessment of a species on the Red List to then predict its Red List category – species with high threat severity with broad scope will be experiencing the greatest impacts and be assessed in the highest categories. It reminds me of previous work using range size to predict Red List categories – despite Red List categories typically being designated based on range size in the first place. This concern is exacerbated by the fact that the Red List are in the process of revising this system for measuring impact due in part to it being subjectively applied. There are spatially explicit data, derived from Red List threat data, available to estimate the pressure of different drivers across the range of a species that could help parameterise or validate the predictions made here (Harfoot et al. 2021, *Nature Ecology & Evolution*: <https://www.nature.com/articles/s41559-021-01542-9>).

A potentially bigger issue is what appears to be a fundamental misunderstanding of the IUCN Red List criteria in the formulation of extinction scenarios, though perhaps I have just misunderstood the details in the methods. As I understand it from reading the manuscript, the 'complete abatement' scenario sets all future population declines to zero, and the models

are used to predict the Red List category using this new data? However, given that this means species % population declines are 0 for 10 years or three generations (as you state on lines 434-435), and by definition the threats driving the declines are removed, that after this 0% decline for 10 years / 3 generations, all species would qualify as Least Concern unless they qualified for a threatened category under criteria B1ac, B2ac, or D1 (see the IUCN Red List criteria here: <https://www.iucnredlist.org/resources/categories-and-criteria>). Further, by applying the expected population declines from Garnett et al., only two situations represent where, following partial abatement, a species would be still eligible to be assessed as VU, EN, or CR on the Red List based on % population declines alone: scope of 'Majority' or 'whole' with severity of 'very rapid declines' (see supplementary table 2 in Mair et al. 2021).

I think this has the potential to explain the authors' surprising findings that extinction risk does not decrease as much as expected following the abatement of threat: they are artificially inflating extinction risk—as derived from Red List categories—in their models by incorrectly assigning species to more severe Red List categories.

Indeed, on lines 155-161, the authors surmise that their findings challenge “some of the key assumptions of global conservation strategies”, referencing the STAR metric and its potentially incorrect assumption that threat abatement would facilitate species being downlisted to Least Concern. However, this assumption is correct in terms of the application of the Red List criteria, as explained above. Species with no declines for 10 years / 3 generations and no current or plausible future threats would indeed be assessed as LC in the vast majority of cases, and I would expect to see a greater decline in extinction risk with the criteria correctly applied. In fact, the authors' suggestion that abatement alone is not enough, but restoration is also needed, is acknowledged by Mair et al. 2021 in the paper outlining the STAR metric, where they say: “Restoration may be particularly important for some species, including those assessed under Red List sub-criteria D/D1 (with a very small population), or Bac (with a small range with severe fragmentation, plus extreme fluctuations). For species uniquely assessed under these criteria (2.8% of those included in this study), threat abatement alone is unlikely to eliminate extinction risk, and so might need to be complemented by restoration in order to achieve Least Concern status (see Supplementary Discussion). Moreover, depending on habitat loss and threat type, restoration of habitat may be beneficial for a larger proportion of threatened species.”

Minor comments:

Line 431: when a group is described as 'comprehensively' assessed on the Red List, that means that >80% of the species in the group have been assessed on the Red List, not that the threat data is comprehensively described for each species. A group of species could be comprehensively assessed but have poor threat data (e.g. turtles).

Line 686: Top 100 unique threatened species seems arbitrary and its unclear why NT species were omitted despite being included in the extinction scenarios.

Figure 1 and supplementary figures don't adequately display underlying data and this will need to be addressed (e.g. box plots with points rather than bar charts, no raw points or measures of uncertainty are shown in figure 1).

Reviewer #2 (Remarks to the Author):

Review:

This is a fabulous paper that investigates the effectiveness of different conservation strategies in preserving both species richness and functional diversity among birds over the next 100 years. I have a few major concerns and a few minor concerns listed below.

Major:

- The methodology assumes that all threats can be reduced by 100%, 50%, or 10%. This is a significant assumption that may not be realistic in many cases. For example, it would be nearly impossible to eliminate livestock farming and ranching entirely across the entire range of the Red Goshawk, which covers 1,600,000 km². Similarly, it's unlikely that invasive cats could be reduced by 10% across all Australasian bittern habitats. Threats like disease and climate change also present challenges that are difficult to address fully. It would be useful to consider the feasibility of these reductions in the modelling, taking into account the extent of the area affected and the practical challenges involved in managing each threat.
- The research assumes that all threats can be halted immediately. Again, this is just not feasible for most threats. Is there a way to step wise the threat reduction over several years to capture what is more realistic?
- Line 488: more detail is required on the missForest imputation. Imputation of missing scope, severity, and timing data may introduce biases if the imputation model doesn't fully capture the variability of the original data, but hard to tell if this is a problem with so little detail.
- The use of a Markov Chain Monte Carlo multivariate generalized linear mixed model with a fixed residual variance of 166 might oversimplify the model, particularly for ordinal response variables like extinction risk. This simplification could lead to inaccurate estimates of extinction risk and functional diversity loss. Perhaps try to evaluate different model structures or priors to ensure robustness, and consider using alternative methods to estimate residual variance for ordinal data.
- Disentangling the independent effects of multiple threats is complex, and the assumption that threats are not randomly distributed with respect to phylogeny and spatial variables may oversimplify real-world conditions. This could lead to inaccurate assessments of threat impacts and biased extinction projections. Maybe explore alternative modeling approaches that account for potential interactions between threats and spatial or phylogenetic variables more explicitly.
- The methodology uses phylogenetic principal component analysis (pPCA) to reduce dimensionality and assess functional trait space, but the choice of three pPCs might not fully capture all relevant variation. The selected number of pPCs could

affect the accuracy of functional diversity estimates and the identification of critical traits. I suggest performing robustness checks with different numbers of pPCs and validate the results using alternative dimensionality reduction techniques.

Minor:

- Title: the title confusing. Usually, I think of targeted recovery programs to include threat reduction. Maybe; 'Threat reduction must be coupled with captive breeding programs to conserve global bird diversity'
- Line 61: requires references as this is not the case in most literature I have read
- Line 65: the term 'targeted recovery programmes' is confusing, are you talking about captive breeding (ex-situ actions), or recovery plans which capture all that is needed (invasive species management, appropriate fire regimes, etc)?
- Line 70: Unclear as to what 'abatement can reduce the need for direct action remains unclear.' Usually we talk about threat abatement (i.e., reducing invasive cat populations) as direct action, but that doesn't seem to be the case here?
- Maybe add some caveats in the manuscript about some species have already gone extinct but have not yet been updated in the database (e.g., *Neochmia ruficauda ruficauda*).
- The study reconciles taxonomic discrepancies between AVONET and IUCN using a crosswalk but does not mention the extent to which taxonomic changes or updates might impact the results.

Reviewer #3 (Remarks to the Author):

This manuscript models the relationship between threats to birds and their extinction risk, and then predicts extinctions under different scenarios of threat abatement. The authors use species traits to consider how functional diversity will be affected by predicted extinctions under those threat abatement scenarios. They conclude that threat abatement alone is not sufficient to prevent extinctions and the loss of functional diversity among birds.

The study is interesting and timely, and conclusions support previous work. However, it is quite dense and the methods are difficult to get to grips with. My overarching comment is conceptual, and I have two major queries on the methods, which require clarification.

Main comment on the concept:

I am not convinced that the model structure forms an ecological basis for the conclusion that species would continue to go extinct even under complete threat abatement. If I understand correctly, then under the complete threat abatement scenario, all threats affecting the species are set to zero, and hence the drivers of predicted extinction risk are the random model effects, which are phylogeny and longitude/latitude. I don't believe that from an ecological perspective, these variables are sufficient to predict the ability of species to recover, or not, if their threats were abated. The ability of a species to recover without targeted intervention is likely to be reliant on the history of impacts on the species (resulting in e.g. very small populations or genetic bottlenecks) as well as environmental factors such as the availability of habitat, see e.g. the justifications provided by Bolam et al. 2022. Phylogeny and location have been shown to explain variation in extinction risk, but to what extent do they explain recovery? I think this is a model of extinction risk, not of recovery, hence while I think that the model reaches reasonable conclusions, I'm not sure that it does so in an ecologically sound way.

Taking the text at L142-154, I don't find this paragraph provides a convincing discussion of the model behaviour, and I'm struggling to understand the ecological explanations here. In what way, i.e. through which specific variables, does the model capture species that are vulnerable despite having few threats? The example of the Grenada dove seems to suggest that actually the modelling framework itself is not particularly appropriate for this kind of case. The manuscript states that the dove is predicted to remain at risk of extinction even under complete threat abatement, because of it has a small population in a tiny area affected by multiple threats. However, if there was complete threat abatement, then at a minimum, the dove would no longer be affected by those threats, and potentially more likely, the threats would no longer be operating in the area. So what does multiple threats have to do with it? I have inferred that the model predicts a high extinction risk for this species because of the phylogenetic and geographical random effects in the model – is that correct? If so, this really runs counter to the exercise, as the abatement of threats to the species would be place-specific, meaning Grenada would no longer be a high-threat environment for the species. The potential limitation, then, to the recovery of the species would be its small population size and the small area of Grenada combined with limited natural habitat – these would be the key attributes to identifying that the species requires more than threat abatement to recover. I don't think that these are captured in the model. Hence, I don't find this interpretation convincing because it is not an accurate description of how the model is behaving.

I would be open to persuasion on this point, but I do not currently see evidence in the manuscript for the suitability of this model to predict species response to threat abatement.

Main comments on methods 1:

The manuscript does not include the final model used to predict species extinction risk based on threats as fixed effects and phylogeny/geography as random effects. It would help the reader understand the modelling, and hence how conclusions are reached, if the structure of the final model was presented. This is a fairly major omission in terms of the transparency of the methods.

Main comments on methods 2:

It is unclear to me how Least Concern species have been treated. As far as I understand, the model of extinction risk against threats does not include LC species, yet the model predicting numbers of extinctions does seem to include LC species, despite the fact that their threat profile cannot be varied among scenarios. I feel like this exaggerates the sample size, given LC species do not vary among scenarios. If the authors wish to argue against this, then I would suggest that there needs to be a much more explicit statement on the fact that LC species remain LC in all scenarios and are not considered to be impacted by threats. Related to this, at L358-361, I don't understand what is meant by not including threats affecting LC species – does this not mean rather that LC species as a whole are not included?

Specific comments:

Fig 1a. I don't understand why circles are used here – are the extinctions in complete abatement a perfect subset of the extinctions in partial abatement, i.e. the same species? Why is the scale on the left-hand side non-linear?

L137-141. Re-running the analysis using LC weightings for NT species doesn't particularly make sense to me, as I don't think we can expect NT species to have comparable threat profiles to LC species.

L155-161. Firstly, which global conservation strategies make this assumption? STAR is not a conservation strategy and, for example, the Kunming-Montreal Global Biodiversity Framework includes Target 4, which was developed specifically to address the fact that some species will need targeted conservation. Secondly, I think there needs to be an explanation as to exactly which model parameters are predicting that there will still be species extinctions when all threats are alleviated. Taking the example of the STAR metric – Mair et al. 2021 state that their major assumption does not hold for species listed under specific Red List criteria. Whereas it is not at all clear here what is driving those predicted extinctions in the model.

Fig 2: can the pPC axes be described? It's not clear what functional space is represented here, and so it's not possible to understand what is lost or not. The use of silhouettes makes for an attractive figure, but they don't provide enough information to understand the patterns.

Fig 3: Similarly, the reliance on silhouettes means I'm not sure what variation is related to the drivers of extinction. Part b is a nice figure but it doesn't allow me to see the data supporting the conclusion that drivers of extinction are unevenly distributed across the avian tree of life.

L247-280. This section could be re-written to be more accessible. The text describes model results in relation to pPC axes, and then includes the relevant morphology description in brackets, which makes it hard work for the reader. L273-275 presents results the other way around and is much easier to understand.

L474-481. I'm not convinced by the treatment of multiple threats under the same level 2 category - this is effectively ignoring multiple threats. The threat classification system is hierarchical, which means that each of the most detailed levels documents a different threat, even if those different threats are e.g. three different invasive species.

L503-511. The purpose of grouping threats into broad categories of drivers is not explained here, and hence this paragraph doesn't make sense to the reader. I think in general that some careful revision of methods structure to make the flow more logical could help the reader understand what has been done.

L534-537. How can a DD species have a probability of extinction? It is not explained in the text what probability of extinction is actually assigned to DD species.

L554-556. This is a confusing assumption, surely under this logic, in the threat abatement scenarios every threat that has been abated should become a past threat, and so should be maintained rather than given a zero?

Reviewer #4 (Remarks to the Author):

Key results

The stand-out finding of this study is that even a scenario of worldwide large-scale threat abatement is predicted only to mitigate 50% of the biodiversity loss (both taxonomic and functional) expected from ongoing, and projected to continue, threat levels. This is notably at odds with an assumed outcome that complete threat abatement would allow the great majority of species to be downgraded to Least Concern. Importantly, relative levels of mitigation between taxonomic and functional diversity loss are found to vary according to the specific threat driver being abated. Finally, the findings of this study point to the need for a combination of large-scale threat abatement and targeted species recovery programmes to conserve maximum levels of biodiversity.

Validity

The data interpretation and robustness of the conclusions are generally very good. The focus on the statistic of avoidance of 68% of projected functional diversity loss by targeted protection of 100 most functionally unique threatened birds is a little crude. In line with the use of several abatement scenarios, I would prefer to understand the sensitivity of these statistics to several different multiples of the number of functionally unique species targeted. Also, emphasis on the 68% figure raises a couple of points. On the one hand 68% of the 3.2% of projected FD decline under baseline conditions, is equivalent to circa

2% of global avian FD loss avoided. While 2% sounds less significant, it would be illustrative to explicitly state that this means targeting of circa 1% of avian species saves around 2% of avian FD, if that 1% is the most functionally unique threatened species.

Significance

This is a significant step forward in understanding the influence of threat drivers on taxonomic and functional diversity. This study provides further clear evidence of the critical need for both large-scale threat abatement as well as targeted species interventions and will be of wide interest to conservation ecologists, conservation managers and policy makers. This further underscores the radical steps and commitments needed to reverse biodiversity declines. The finding that `hunting and collection` and `disturbance and accidental mortality` disproportionately impacts functional diversity, hence ecosystem functioning and resilience, is an important message of this work. While halting wholesale habitat loss is a top priority, the steady erosion of ecosystem functional elements via various forms of disturbance and exploitation is an important implication.

Data and methodology

Overall, the approach used is valid and important in its consideration of the impacts of different threats upon avian functional trait space. The main text acknowledges the optimistic assumptions of a worldwide complete abatement of threats scenario. The additional stepped scenarios (partial and minimal abatement) are obviously simplified in being globally uniformly implemented. Nevertheless, such an approach seems valid given what is already a complex and ambitious analysis to communicate in one article, namely considering multiple dimensions of avian niche space, and treating, in combination and separately, multiple threat drivers.

Figure 2. Currently the mismatches/differences within trait space between proportional decline (a and b) and averted proportional decline (c and d) are quite subtle and might be made more obvious by `difference maps` e.g. (a minus c), and (b minus d), hence effective in showing the shortfall in abatement over mismatch in different parts of the trait space.

Figure 2, and lines 368-370 of legend. It would surely be valuable and interesting to highlight areas where decrease in functional trait space occupation was not preventable by managing drivers of extinction, rather than maintaining these areas as grey and indistinguishable from areas where there was no decrease in functional trait space occupation?

Extended Data Figure 4. Y-axis would be better scaled as percentage variance attributable to each pPC - eigenvalues being only meaningful in a relative sense. Knowing the absolute variance attributable to pPC4 and onwards would be more meaningful than use of an elbow as the cutoff.

Supplementary Figure 1. This figure legend is not stand alone - unclear what relative density refers to here, or what this set of figure elements is showing us, without reference to the main text.

Analytical approach

The treatment of avian niche space variation using a functional trait space approach, combining phylogenetic PCA and functional diversity estimation using probabilistic hypervolumes (rather than convex hull volumes), are robust approaches, as is the use of models fitting random effects to account for spatial and phylogenetic relationships among species.

Suggested improvements

Other than sensitivity suggested above, I don't have further suggestions for additional analyses

Clarity and context

The clarity in reporting the key results could be improved. Specifically, in the abstract (lines 33-36), where it says "the immediate abatement of all threats across at least half of species ranges for ~10,000 bird species - will only prevent half of projected species extinctions and functional diversity loss attributable to current and future threats in the next 100 years." This is at odds with the subsequent main text where it says (lines 131-132): "Under the complete abatement scenario, half of biodiversity loss predicted under the baseline scenario could be prevented" where complete abatement is previously reported (lines 120-121) as "removal of all direct drivers of extinction across the entirety of all species ranges". Question for clarification: is prevention of loss of half of biodiversity associated with removal of all threats across the entirety of species ranges or at least half of species ranges?

Lines 142-154. This section highlights some species for which large-scale threat abatement will have little or no benefit. However, it does not make clear which remaining mechanisms (given complete abatement) inherent in the model are driving them to likely extinction.

References

The manuscript appropriately references previous literature.

Version 1:

Decision Letter:

5th February 2025

Dear Dr Stewart,

Your manuscript entitled "Threat reduction must be coupled with targeted recovery programmes to conserve global bird diversity" has now been seen by our original four reviewers, whose comments are attached. As you will see, the reviewers

commend the revisions, but some of the reviewers have remaining concerns which will need to be addressed before we can offer publication in Nature Ecology & Evolution. We will need to see your responses to the reviewers' comments, along with a revised manuscript, before we can reach a final decision regarding publication.

We therefore invite you to revise your manuscript taking into account all reviewer and editor comments. Please highlight all changes in the manuscript text file.

* If you have not done so already please begin to revise your manuscript so that it conforms to our Article format instructions at <http://www.nature.com/natecolevol/info/final-submission>. Refer also to any guidelines provided in this letter.

* Extended Data Figures - please ensure that any supplementary figures and tables that are crucial to the manuscript's conclusions are converted into Extended Data figures and tables to increase visibility of these data. Extended Data figures and tables are online-only (present in the online PDF and full-text HTML versions of the paper), peer-reviewed display items that provide essential background to the article but are not included in the main article due to space constraints. A maximum of ten Extended Data display items (figures and tables) is permitted.

Link Redacted

Nature Ecology & Evolution is committed to improving transparency in authorship. As part of our efforts in this direction, we are now requesting that all authors identified as 'corresponding author' on published papers create and link their Open Researcher and Contributor Identifier (ORCID) with their account on the Manuscript Tracking System (MTS), prior to acceptance. ORCID helps the scientific community achieve unambiguous attribution of all scholarly contributions. You can create and link your ORCID from the home page of the MTS by clicking on 'Modify my Springer Nature account'. For more information please visit www.springernature.com/orcid.

[redacted]

Reviewer comments:

Reviewer #1 (Remarks to the Author):

The revisions to the manuscript are great and have improved the clarity of the research, addressing most of my concerns – particularly the reframing of the results to make it clear that it is not that threat abatement won't work, it is that it is potentially too late for a small set of marginal species where threats have left them irreparably vulnerable to extinction without targeted intervention. The authors made it much clearer why they built their models the way they did and, whilst they still only weakly reflect the real-world application of red list information that underpin assessments, they do fit the purpose of this theoretical piece.

My only major concern remains around the interpretation of the results, which initially made it hard to grasp the rationale for the study design used. The authors present work which suggest that abatement is 'only' going to stop 50% of projected extinctions. However, there still seems to be a bit of context missing. It seems implicit that, following total abatement, there are 97.5% of species who do not become extinct because total abatement of threat is sufficient to prevent their extinction.

What the authors do not seem to take into account is the future situation where, yes, we still lose 50% of projected extinctions under the total abatement scenario, but the lack of threats to continue to act on species will ensure continuously lower extinction rates beyond the authors' time horizon, compared with the no—or partial—abatement approach. I worry that the somewhat sensationalist approach to the interpretation of the findings have the potential to undermine real-world efforts to abate threat sufficiently to deliver conservation success. Hopefully the authors can consider this in any subsequent modifications.

Reviewer #2 (Remarks to the Author):

I thank the authors for their detailed responses and amendments to the manuscript. Nothing further from me.

Reviewer #2 (Remarks on code availability):

All data and code seem to be open and accessible for reproducibility.

Reviewer #3 (Remarks to the Author):

The authors have done a good job of responding to the many comments from reviewers. They have defended their approach, provided explanations, conducted additional analyses, and made many of the suggested changes. Altogether the manuscript is much improved.

I have two remaining queries, and then a few minor comments.

Query 1

In response to my comments about whether the model captures extinction risk or recovery potential, the authors provided a nice supporting analysis using Green Status data, alongside a clear explanation, which is much appreciated. I did, however, find that their explanations of what the random phylogenetic and spatial effects capture varied somewhat among their explanations to different reviewers. In another response to review comment, they state that “This means that if higher/lower threat impact scores are given for particular species groups, for particular parts of the world or for particular threats, due to subjective application of the system for measuring impact, this will be estimated by random and fixed effects in our extinction risk model to provide the best description of extinction risk given the data”. This is not the same as their explanation that “In our model the spatial and phylogenetic random effects can account for higher or lower than expected extinction risk given threats due to non-random differences in baseline extinction rates across space and the avian tree of life”. Nor as their text in the manuscript at L516-518 saying “Threats varied across the tree of life and spatially so we included phylogeny and spatial variables as random effects”.

Clarification would be appreciated, and I think the explanation of what the random phylogenetic and spatial effects capture could be better integrated into the manuscript. Specifically, at L143-153, I think the explanation provided to reviewers was clearer than the text here in the manuscript. In particular, inclusion of supporting references (as was in their response to reviewers) would help. Additionally, considering L154-162, these examples could be better linked to the explanation of random effects, if explanation and examples were discussed together. I also think the petrel example could be improved from a vague and unreferenced statement of unnamed ‘factors associated’.

Finally on this point, I find the statements at L172-176 weak, because the model does not explicitly capture these effects. This is indeed an explanation, but these mechanisms are not captured in the model, and hence this conclusion does not arise from the work presented here.

Query 2

Supplementary Dataset 1. Why do two threats (2.2 and 10) have a negative posterior mean? This seems counterintuitive, and makes me wonder about the behaviour of the model – is it over-fitted?

Minor comments

L95 should be “predicted” past tense? In general, the text describing methods moves between present and past tense, and I think it would be easier to read if it was all in past tense.

L133-142. The explanation of this analysis was only clear from the response to reviewers. The reason there might be underestimates is because some species might recover to LC, which is a subtle but important difference to not including LC species in the model. Taking the explanation in the response to reviewers and incorporating into the main text would help here.

L510-514 The authors quote numbers for all species (i.e. including LC), but only include NT and threatened species in the extinction risk model. Surely it is only the species included in the model that are of relevance here?

L522-523. "Threats have been comprehensively described for 99% of species in these categories". This is still an incorrect statement – threats have been described, but that does not mean that these descriptions are comprehensive. And indeed the authors contradict this statement at L549 where they say 11% of threat data were missing scope or severity information.

L534-548. The authors responded to my comments on the treatment of threat data, but misunderstood my point, so my apologies. I appreciate that threats are not always documented to level 3, and so their treatment of level 3 threats is understandable. My issue was with multiple threats listed under the same level 2 threat, i.e. where they explain that a species may be impacted by two different invasives, which are documented as separate threats, the authors treat these as only one threat. My argument is that they are not one threat, they are two, which may operate in very different ways, impact the species in different and cumulative and/or interactive ways, and require different interventions. I do not think they should be treated as one threat; their impact should instead be summed.

L627-628. The treatment of DD species is likely to have only a small effect on overall predictions given their small number, but classifying them all as LC is likely an underestimate of extinction risk.

Reviewer #4 (Remarks to the Author):

The authors have addressed the comments I made including a sensitivity analysis varying the number of unique threatened species targeted from 40 to 200 and I think Extended Data Figure 4 is a very helpful addition.

The authors have also attempted to plot the difference maps suggested for Figure 2. Although they suggest possibly using their new panel b as a supplementary figure, it is new panel d (proportion of density lost under baseline extinction scenario, not saved) that I find more revealing and complementary to the existing figure in the main paper. However, I wouldn't be prescriptive on using either of these. It is now made clear there are very few areas of trait space (three blue pixels in revised figure) where complete abatement made no contribution to reducing the loss in density of occupation, meaning these are effectively not visible hence not worth showing.

The explanation of factors in the model driving species to extinction even under complete abatement is now somewhat clearer, although this model outcome is clearly an area of some debate.

*****END*****

Version 2:

Decision Letter:

31st March 2025

Dear Dr. Stewart,

Thank you for submitting your revised manuscript "Threat reduction must be coupled with targeted recovery programmes to conserve global bird diversity" (NATECOLEVOL-24061657B). It has now been seen again by two of the original reviewers and their comments are below. The reviewers find that the paper has improved in revision, and therefore we'll be happy in principle to publish it in Nature Ecology & Evolution, pending minor revisions to comply with our editorial and formatting guidelines.

If you have not done so already, please ensure that you also email us completed copies of the Reporting summary and Editorial policy checklists:

Reporting summary: https://www.nature.com/documents/nr-reporting-summary.pdf

Editorial policy checklist: https://www.nature.com/documents/nr-editorial-policy-checklist.pdf

[redacted]

Reviewer #1 (Remarks to the Author):

The changes and additions made by the authors across the revised versions have greatly improved the clarity of the manuscript and the interpretation of the results. I have no further comments.

Reviewer #3 (Remarks to the Author):

The authors have addressed the latest round of reviews in detail, and I appreciate the changes they have made. I have no further comments.

Summary of major changes

In response to the reviewer's main comments, we have made substantive changes to the manuscript, listed below. Detailed responses are in the subsequent section.

1. We have added further discussion on what factors could contribute to species extinctions under complete abatement of all drivers of extinction. We stand by our conclusion that there is no evidence that all extinctions can be avoided with complete abatement of threats, so therefore additional conservation measures such as species management actions and habitat restoration will be needed. We provide detailed discussion in response to reviewer's 1 and 3's queries about this below. We also highlight that under partial abatement only half of diversity loss that was attributable to threats (diversity loss under complete abatement) was preventable and it is this result we refer to in the abstract and conclusion of the manuscript.
2. We conducted a series of sensitivity analyses to test the impact of:
 - a. The number of dimensions and ordination method
 - b. Random effect prior specification
 - c. Assumption of whether past threats that are unlikely to return continue to contribute to extinction risk or not

In all sensitivity analyses our conclusions were unchanged. We have included descriptions of these analyses in the supplementary information.

3. We simulated the impact of protecting between 40 and 200 unique threatened species (in 20 species increments). We have added a new extended data figure (extended data figure 4) to show these results, and have edited the methods and the results accordingly.
4. We have reworded the methods and part of the results for improved readability and clarity.
5. We have changed figure 1, and supplementary figures to better show uncertainty. We made small changes to figure 2 and 3 that were suggested by the reviewers.

We have made numerous smaller changes in response to reviewers' suggestions which are detailed below.

Detailed response

Referees and editor's comments are in blue.

Reviewer #1 (Remarks to the Author)

COMMENT: This is a thought-provoking and interesting piece of research that highlights the different impacts of threats and potential threat abatement scenarios on the conservation of two facets of biodiversity: species richness and 'functional diversity'. The work is very well

composed, with clear English and visualisations, and there is a huge number of results and outputs to consume (perhaps bordering too much for a single paper!). There are, however, some fundamental flaws in this manuscript--related to the omission of key trait data and the incorrect estimation of Red List categories--that I believe are currently prohibitive to its publication if left unaddressed, which I outline below.

RESPONSE: We thank the referee for their efforts in reviewing this manuscript. We outline the changes we have made and how they address the referees' comments, below.

COMMENT: **Functional diversity**

My first issue centres on the use of the term 'functional diversity' (FD) in this manuscript. While the analyses using the morphospace are interesting and appropriate, it feels like a stretch to call the morphospace 'FD' whilst it lacks any information on other key 'functional' traits, particularly related to fecundity and longevity. This is a pronounced omission given that the work focuses on short term extinction scenarios—scenarios that include criteria related to generation length—where one would expect traits such as fecundity, age at maturity, and lifespan to play an important role in the fate of species following threat abatement.

For example, when looking at the extinction risk to the FD of birds using traits related to fecundity and longevity, Carmona et al. (2021, *Science Advances*: <https://doi.org/10.1126/sciadv.abf2675>) found that "extinction risk is not randomly distributed but localized in certain areas of the functional space occupied by species with large size, slow pace of life, or low fecundity", and "species with later fledging ages, longer incubation times, and larger sizes having up to six times higher threat risk than smaller species with faster breeding times". However, searching through the manuscript there is zero mention of 'fecundity', 'reproduction', or 'longevity' despite their well-established relevance to extinction risk. I think it is important to ideally include additional analyses including these data for birds to show the effects on the results, or at a minimum address their absence in the text, by exchanging the term 'FD' for 'morphospace' or including discussion around why these traits were not deemed important to characterise conservation-relevant functional indices for birds under extinction scenarios.

RESPONSE: We understand the reviewer's point but we think that the recommendation does not take into account either the goals of our analyses or recent progress in the field of functional morphology (see e.g. Schleuning, et al. 2023). Variation in species *response* to human-induced pressures have been widely studied using life-history traits whereas the aim of this manuscript is to analyse the *effect* that predicted species extinctions might have on ecosystems. Therefore, contrary to the suggestion of the reviewer, we are not interested in the traits associated with extinction risk, but the effects of extinction on the diversity of roles or functions that species play in an ecosystem. To achieve this, we apply the now-standard method of inferring these functions using morphological traits; see, for example, the recent attempt to understand ecological impacts of Anthropogenic extinctions (Matthews et al., 2024 *Science*).

The reviewer is correct that life-history traits have been used to study functional diversity change on a large-scale. For many tetrapod groups, life-history traits have provided an opportunity to study changes in trait composition where availability of morphological data is more limited than in birds. With the release of morphological data for nearly all bird species (the Avonet dataset, Tobias et al., 2022), we are presented with a new opportunity to study how extinction will affect ecosystem processes (Bregman et al., 2016) and trophic interaction networks (Dehling et al., 2016) through the lens of ecologically relevant morphological traits. Importantly, morphological proxies for diet and foraging behaviour can provide more in-depth information about an organism's precise ecological role than standard life history or ecology data alone (Pigot et al., 2020). Morphological traits therefore provide a more direct link to the effects that functional diversity loss can have on ecosystem function, whereas life history traits are primarily informative about species responses to human activity (Hordley et al., 2021).

Therefore, we disagree with the reviewer's point and believe that the focus of this analysis should remain firmly on morphological traits that have been linked to ecological roles (Pigot et al., 2020). In addition, we retain the term 'functional diversity' because this is widely used to describe morphospace (Stewart et al., 2022; Ali et al., 2022; Matthews et al., 2024) and is more readily understood by a wide readership than terms such as morphological diversity. Use of the term functional diversity is appropriate because the impact of morphological diversity change is not limited to morphology due to the association between avian morphology and ecology (Pigot et al., 2020), in line with our definition of functional diversity in the manuscript (line 50-51, and line 468-470).

Finally, we agree with the reviewer that we need to clarify some of these points, so we have added justification for the decision to exclude life-history traits from functional diversity estimations in the manuscript (line 476-478).

COMMENT: Threat data and extinction scenarios

The threat work is very thorough and involved, but there are some issues that require at a minimum clarification and at a maximum rethinking entirely. First, it feels intuitively circular to use threat data that was used to inform the assessment of a species on the Red List to then predict its Red List category – species with high threat severity with broad scope will be experiencing the greatest impacts and be assessed in the highest categories. It reminds me of previous work using range size to predict Red List categories – despite Red List categories typically being designated based on range size in the first place.

RESPONSE:

We understand the concern and agree that using range size to predict Red List status is circular. However, threat data are not directly used to assign species to a Red List category during IUCN assessments. Rather, threat listing is a separate step of the assessment process which involves several classifications, including habitats, conservation actions and other information (IUCN, 2001; IUCN Standards and Petitions Committee, 2024). To determine Red List status, the IUCN assessors use five criteria that refer to population size, population size change, range size, and range size change (and a rarely used criteria that focuses on

predicted extinction risk based on population viability analyses). For each criterion there are thresholds that when met trigger listing for a given Red List status. Assessors can use empirical data (e.g. estimates of population size) or predictions. Predicted or observed changes may certainly be influenced by existing threats, as they are influenced by ecology and all other factors that influence species distribution and population dynamics, but as they are not directly used in the criteria it is not a circularity issue.

Moreover, our model predicts how extinction risk would change if threats were reduced in scope and is not used to make inferences about how threat scope and severity are related to extinction risk. We are therefore not trying to disentangle mechanisms leading to extinction risk.

COMMENT: This concern is exacerbated by the fact that the Red List are in the process of revising this system for measuring impact due in part to it being subjectively applied.

RESPONSE: The IUCN are revising the impact scores, which are based on scope, severity and timing (<https://www.iucnredlist.org/resources/threat-classification-scheme>). We do not use the impact scores, but use the individual metrics of scope, severity and timing to assign expected population decline over 10 years or 3 generations. We therefore follow IUCN recommendations. For example, the Species Threat Abatement and Restoration metric (STAR, Mair et al., 2021), developed by the IUCN Species Survival Commission Post-2020 Taskforce (hosted by the University of Newcastle), is entirely reliant on threat scope and severity scores.

We follow the STAR method when assigning an expected population decline based on threat scope and severity (as done by Garnett et al., 2019). We go one step further by modelling the association between threat impact and extinction risk for all threats. This means that if higher/lower threat impact scores are given for particular species groups, for particular parts of the world or for particular threats, due to subjective application of the system for measuring impact, this will be estimated by random and fixed effects in our extinction risk model to provide the best description of extinction risk given the data. This approach means we can quantify the extent to which threat impact contributes to extinction risk.

COMMENT: There are spatially explicit data, derived from Red List threat data, available to estimate the pressure of different drivers across the range of a species that could help parameterise or validate the predictions made here (Harfoot et al. 2021, *Nature Ecology & Evolution*).

RESPONSE: Harfoot et al. (2021) map the probability that a random species is affected by a range of threats, meaning that the differences between species for a given location are not described, something we are explicitly interested in. The maps also do not capture the severity of the impact, or the biological response to that impact, such as the contribution that threats make to extinction risk. As such it is not possible to validate our outputs with the Harfoot maps as they measure different things.

We did test the performance of our model by quantifying the accuracy of extinction risk prediction. We found that for 86.8% species we were able to correctly predict their extinction category (see line 88, and Supplementary Information: Extinction risk model description).

COMMENT: A potentially bigger issue is what appears to be a fundamental misunderstanding of the IUCN Red List criteria in the formulation of extinction scenarios, though perhaps I have just misunderstood the details in the methods. As I understand it from reading the manuscript, the 'complete abatement' scenario sets all future population declines to zero, and the models are used to predict the Red List category using this new data? However, given that this means species % population declines are 0 for 10 years or three generations (as you state on lines 434-435), and by definition the threats driving the declines are removed, that after this 0% decline for 10 years / 3 generations, all species would qualify as Least Concern unless they qualified for a threatened category under criteria B1ac, B2ac, or D1 (see the IUCN Red List criteria here: <https://www.iucnredlist.org/resources/categories-and-criteria>). Further, by applying the expected population declines from Garnett et al., only two situations represent where, following partial abatement, a species would be still eligible to be assessed as VU, EN, or CR on the Red List based on % population declines alone: scope of 'Majority' or 'whole' with severity of 'very rapid declines' (see supplementary table 2 in Mair et al. 2021).

I think this has the potential to explain the authors' surprising findings that extinction risk does not decrease as much as expected following the abatement of threat: they are artificially inflating extinction risk—as derived from Red List categories—in their models by incorrectly assigning species to more severe Red List categories.

RESPONSE: We thank the reviewer for taking the time to spell out this potential problem. After reflecting on the issue, we are sure that we are not artificially inflating extinction risk. Our model does not overestimate the number of extinctions compared to the number of extinctions predicted from IUCN Red List Index extinction risk categories (line 57-63 of the Supplementary Information), and the number of extinctions we project falls within other estimates of future avian extinctions (Monroe et al., 2019; Andermann et al., 2021).

The complete abatement scenario sets all future declines *resulting from known threats* to zero but does not set all future population declines to zero. It is unrealistic to assume that all population declines are reversible (the IUCN Red List criteria A2 and A4 explicitly reference population declines which are not reversible; IUCN, 2001; IUCN Standards and Petitions Committee, 2024), and it is possible that when all known threats are removed, species populations continue to decline. This is particularly true for species which have a small number of remaining individuals, severely fragmented ranges or small range size.

This is supported by the fact that many species already rely on targeted recovery programmes for their survival, and many more are expected to in future (Bolam et al., 2023). Therefore, it is not an error in the modelling but rather a realistic output that under a complete-abatement scenario some species will not return to Least Concern. Our findings therefore add to emerging evidence that threat reduction alone will no longer be enough to preserve biodiversity in its entirety (Bolam et al., 2023; Pereira et al., 2024). We have added an explanation to the manuscript regarding how species can continue to be threatened with extinction once their threats are removed (line 172-176).

COMMENT: Indeed, on lines 155-161, the authors surmise that their findings challenge “some of the key assumptions of global conservation strategies”, referencing the STAR metric and its potentially incorrect assumption that threat abatement would facilitate species being downlisted to Least Concern. However, this assumption is correct in terms of the application of the Red List criteria, as explained above. Species with no declines for 10 years / 3 generations and no current or plausible future threats would indeed be assessed as LC in the vast majority of cases, and I would expect to see a greater decline in extinction risk with the criteria correctly applied.

RESPONSE: As mentioned above, there are different reasons why species might continue to decline following threat abatement and these are captured in the Red List assessment protocol. Species with naturally small population or range sizes (e.g. those endemic to small islands) are vulnerable to extinction even without ongoing anthropogenic threats, and species could continue to experience population decline following threat abatement if populations were no longer self-sustaining.

COMMENT: In fact, the authors’ suggestion that abatement alone is not enough, but restoration is also needed, is acknowledged by Mair et al. 2021 in the paper outlining the STAR metric, where they say:

“Restoration may be particularly important for some species, including those assessed under Red List sub-criteria D/D1 (with a very small population), or Bc (with a small range with severe fragmentation, plus extreme fluctuations). For species uniquely assessed under these criteria (2.8% of those included in this study), threat abatement alone is unlikely to eliminate extinction risk, and so might need to be complemented by restoration in order to achieve Least Concern status (see Supplementary Discussion). Moreover, depending on habitat loss and threat type, restoration of habitat may be beneficial for a larger proportion of threatened species.”

RESPONSE: We agree and this supports our conclusions. We have added a description that restoration may help species for which threat reduction is not enough to alleviate extinction risk and highlight that this point was also raised by Mair et al. (2021) (line 174-181).

COMMENT: **Minor comments**

Line 431: when a group is described as ‘comprehensively’ assessed on the Red List, that means that >80% of the species in the group have been assessed on the Red List, not that the threat data is comprehensively described for each species. A group of species could be comprehensively assessed but have poor threat data (e.g. turtles).

RESPONSE: We have reworded this sentence to avoid confusion (line 522-523).

COMMENT: Line 686: Top 100 unique threatened species seems arbitrary and it’s unclear why NT species were omitted despite being included in the extinction scenarios.

RESPONSE: Near-Threatened species were not included despite being included in the extinction scenarios, because targeted recovery programmes are costly and time-consuming so often reserved for the most threatened species. For Near-Threatened species threat

abatement is expected to be more appropriate until their decline worsens, or population/range size shrinks such that they are listed as threatened.

We agree that the choice of 100 species is arbitrary, so we have run the analysis a number of times varying the number of unique threatened species which were prevented from going extinct. Between 40 (~3% of threatened species studied) and 200 (~16% of threatened species studied) species were prevented from going extinct, at increments of 20 species. We have updated the methods (line 765-771), and the main text (line 309-311) to reflect this, and plotted the results in extended data figure 4.

COMMENT: Figure 1 and supplementary figures don't adequately display underlying data and this will need to be addressed (e.g. box plots with points rather than bar charts, no raw points or measures of uncertainty are shown in figure 1).

RESPONSE: We have changed figure 1 to show uncertainty and raw data points, and supplementary figures 2, 4 and 6 to boxplots with raw data points.

Reviewer #2 (Remarks to the Author)

COMMENT: This is a fabulous paper that investigates the effectiveness of different conservation strategies in preserving both species richness and functional diversity among birds over the next 100 years. I have a few major concerns and a few minor concerns listed below.

RESPONSE: We thank the referee for their efforts in reviewing this manuscript and their positive feedback. We outline the changes we have made and how they address the referees' comments, below.

COMMENT: **Major**

The methodology assumes that all threats can be reduced by 100%, 50%, or 10%. This is a significant assumption that may not be realistic in many cases. For example, it would be nearly impossible to eliminate livestock farming and ranching entirely across the entire range of the Red Goshawk, which covers 1,600,000 km². Similarly, it's unlikely that invasive cats could be reduced by 10% across all Australasian bittern habitats. Threats like disease and climate change also present challenges that are difficult to address fully. It would be useful to consider the feasibility of these reductions in the modelling, taking into account the extent of the area affected and the practical challenges involved in managing each threat. The research assumes that all threats can be halted immediately. Again, this is just not feasible for most threats. Is there a way to step wise the threat reduction over several years to capture what is more realistic?

RESPONSE: We fully agree and have made sure it is clearly written in the manuscript that we apply optimistic scenarios of threat reduction where the impact of all threats can be halted immediately across at least 10%, 50% or 100% of species ranges. We explicitly discuss that the scenarios are optimistic and unrealistic (line 206-213). The goal of our study is to not

define realistic abatement scenarios (which would need to be more species and region specific) but rather to highlight that even extreme measures would fall short. Our message is that even if we were somehow able to apply such extreme measures, threat reduction will be insufficient to avoid all biodiversity loss. More limited reductions of threats can only result in more loss.

It is difficult to define realistic constraints on the extent and intensity of threat reduction possible, as the reviewer suggests. For example, agricultural lands cover around 38% of the Earth's terrestrial surface (Kehoe et al., 2017), but in many areas, it may be possible to avert species population declines due to agriculture, while maintaining food production (Fox, 2004). Addressing the land sharing-sparing debate is beyond the scope of our study. Identifying the maximum area across which the impact of threats can feasibly be removed becomes increasingly complicated, as there are nineteen threats included in the final extinction risk model, many of which are far less studied than agricultural threats such as livestock farming and ranching. We have therefore kept the optimistic and simplifying assumption that all threats can be completely abated across all species ranges, as the feasibility of large-scale threat reduction is a large, complex, and separate topic.

COMMENT: Line 488: more detail is required on the missForest imputation. Imputation of missing scope, severity, and timing data may introduce biases if the imputation model doesn't fully capture the variability of the original data, but hard to tell if this is a problem with so little detail.

RESPONSE: Given that the proportion of imputed data was low (<12% of threat-species combinations were missing data on scope, severity or timing), and the imputation accuracy was high ($82.5 \pm 0.8\%$) when tested on simulated removal of scope and severity data from the complete dataset, it is highly unlikely that imputation is biasing our results. A full description of the imputation procedure and accuracy tests can be found in the supplementary information (see section on "*Scope and severity imputation accuracy*", lines 328-354 in the Supplementary information).

It is possible that imputation more often predicts common categories of scope, severity or timing, than rare categories, however this was not the case (Table 1, 2 and 3 below).

Table 1 Percentage of threat data with given scope values in the full dataset (threat data where rows with missing scope, severity or timing values were removed), and for imputed values in the imputed dataset (where data was removed from the full dataset and imputed to test imputation performance, see Supplementary Information section on *Scope and severity imputation accuracy*). Values for imputed datasets show mean \pm standard deviation across 100 imputed datasets.

Scope	Minority (<50%)	Majority (50-90%)	Whole (>90%)
Full dataset	46.2%	47.8%	6%
Imputed datasets	46.9 \pm 2.4%	45.8 \pm 2.4%	7.3 \pm 1.3%

Table 2 Percentage of threat data with given severity values in the full dataset (threat data where rows with missing scope, severity or timing values were removed), and for imputed values in imputed datasets (where data was removed from the full dataset and imputed to test imputation performance, see Supplementary Information section on *Scope and severity imputation accuracy*). Values for imputed datasets show mean \pm standard deviation across 100 imputed datasets.

Severity	No decline	Causing/ Could cause fluctuations	Negligible declines	Slow, Significant Declines	Rapid Declines	Very Rapid Declines
Full dataset	2.9%	1.6%	15.2%	65.1%	13.1%	2.2%
Imputed datasets	3.6 \pm 0.4%	1.9 \pm 0.3%	17.1 \pm 0.8%	59.5 \pm 1.1%	15.1 \pm 0.9%	2.9 \pm 0.4%

Table 3 Percentage of threat data with given timing values in the full dataset (threat data where rows with missing scope, severity or timing values were removed), and for imputed values in the imputed dataset (where data was removed from the full dataset and imputed to test imputation performance, see Supplementary Information section on *Scope and severity imputation accuracy*). Values for imputed datasets show mean \pm standard deviation across 100 imputed datasets.

Timing	Past, Unlikely to Return	Past, Likely to Return	Ongoing	Future
Full dataset	3.1%	2.6%	89.7%	4.6%
Imputed datasets	3.2 \pm 1.1%	2.2 \pm 0.9%	90.2 \pm 1.8%	4.3 \pm 1.2%

COMMENT: The use of a Markov Chain Monte Carlo multivariate generalized linear mixed model with a fixed residual variance of 166 might oversimplify the model, particularly for ordinal response variables like extinction risk. This simplification could lead to inaccurate estimates of extinction risk and functional diversity loss. Perhaps try to evaluate different model structures or priors to ensure robustness, and consider using alternative methods to estimate residual variance for ordinal data.

RESPONSE: We apologise and realise that we may not have explained this clearly in the original submission. The residual variance was set to 1 (not 166) as recommended by Hadfield (2017) when fitting ordinal models (see discussion on p.48 of Course Notes of the MCMCglmm package: <http://cran.nexr.com/web/packages/MCMCglmm/vignettes/CourseNotes.pdf>). Capellini et al. (2015) and de Villemereuil et al. (2013) also fix the residual variance to 1 when fitting MCMCglmm models on binary data (in the manuscript we cited Capellini et al. [2015] with the numbered reference 66, which could look like 166 when the 1 was next to the 66 for the reference). We have reworded line 583-584 to prevent this confusion.

Nevertheless, we tested the impact of estimating (degree of belief of 0 and expected covariance of 1) rather than fixing residual variance. The model took longer to converge (posterior estimates were sampled between 100 000 and 500 000 iterations rather than between 3000 and 103 000 iterations) and provided a poorer description of the data than when residual variance was fixed as self-prediction accuracy was lower (75.8% vs 86.8% when residual variance was fixed). In addition, the number of extinctions projected from species included in the model (2087 species) was underestimated relative to that projected from the real IUCN categories (IUCN category predicted extinction: 500 ± 12 species, original model with fixed residual variance: 502 ± 19 species, new model with estimated residual variance: 410 ± 26 species).

We also tested the sensitivity of the model to random effect prior specification which did not affect our results (see new section of Supplementary Information section on *Prior specification*, lines 355-391). We used weak priors for fixed effects, so we do not expect that the priors unduly influenced posterior estimates (Supplementary figure 14).

COMMENT: Disentangling the independent effects of multiple threats is complex, and the assumption that threats are not randomly distributed with respect to phylogeny and spatial variables may oversimplify real-world conditions. This could lead to inaccurate assessments of threat impacts and biased extinction projections. Maybe explore alternative modelling approaches that account for potential interactions between threats and spatial or phylogenetic variables more explicitly.

RESPONSE: We ran a series of sensitivity analyses to test for interactions between threats and spatial variables (categorising species by latitudinal zone), and threats and phylogenetic variables (order). In summary, interactions between the latitudinal zone of species breeding and resident ranges and threats were not useful for explaining variation in extinction risk.

While singularities prevented us from including all interactions between threats and orders, a reduced model with five threats, and three orders had no significant interactions between threats and order. As a result, we conclude that the extinction risk model with spatial and phylogenetic random variables and no interactions between threats and spatial or phylogenetic variables is most appropriate for producing accurate extinction risk projections. We provide further detail on sensitivity analyses conducted below.

First, we tested whether interactions between spatial variables and threats were important for explaining variation in extinction risk. To do this we classified species into three categories, "*Tropical only*", "*Temperate*" and "*Polar*" based on the minimum and maximum latitude of species breeding and resident ranges in areas where a species was extant (native or reintroduced) as provided in Avonet (Tobias et al., 2022). "*Tropical only*" included species only found in tropical latitudes with a minimum latitude of more than or equal to -23.44° and a maximum latitude of less than or equal to 23.44° . "*Temperate*" included species which had at least part of their range in temperate latitudes with a minimum latitude of less than -23.44° or a maximum latitude of more than 23.44° , and a minimum latitude that was more than -66.56° and maximum latitude that was less than 66.56° . "*Polar*" included species which had at least part of their range in polar latitudes, with a minimum latitude of less than or equal to -66.56° or a maximum latitude of more than or equal to 66.56° . We included these categories as a categorical fixed effect, hereafter referred to as zone.

We ran the extinction risk model with phylogenetic and spatial random effects (minimum latitude, maximum latitude and centroid longitude), and with 19 threats and zone as fixed effects, as well as interactions between zone and all threats. We used Cauchy-scaled Gelman priors for all fixed effects, a Chi-squared prior for the phylogenetic random effect, and parameter expanded priors for the spatial random effects. The model was run for 103 000 iterations with a 3000 iteration burn in period and a thinning interval of 100, giving 1000 posterior values for each fixed effect. No interactions between zone and threats were significant when all interactions were included ($n=2087$ sp., $pMCMC > 0.1$ for all interactions). We removed interactions with a $pMCMC < 0.5$ and reran the model but none of the remaining interactions were significant ($n=2087$ sp., $pMCMC > 0.1$ for all interactions). We repeated this once more, removing interactions with a $pMCMC < 0.2$, after which no interactions were significant ($n=2087$ sp., $pMCMC > 0.1$ for all interactions). These findings suggest that including more explicit interactions between threats and spatial variables was not useful for explaining additional variation in extinction risk in this case.

Next, we aimed to assess whether interactions between threats and order (according to the taxonomy used by Jetz et al. [2012]) were useful for explaining variation in extinction risk. It was not possible to estimate the impact of the interaction between all threats and orders on extinction risk, as many orders were not affected by all threats, and many threats did not affect all orders. As such we estimated interactions between orders and threats using a reduced sample including 1248 species (as opposed to 2087 included in the original extinction risk model). We selected three orders with the greatest number of threatened or Near Threatened species (*Passeriformes*, *Psittaciformes* and *Galliformes*) and the five threats which affected the most species amongst these orders ("*2.3 Livestock farming & ranching*",

"5.1 Hunting & collecting terrestrial animals" and "5.3 Logging & wood harvesting"). We used Cauchy-scaled Gelman priors for all fixed effects, a Chi-squared prior for the phylogenetic random effect, and parameter expanded priors for the spatial random effects. The model was run for 103 000 iterations with a 3000 iteration burn in period and a thinning interval of 100, giving 1000 posterior values for each fixed effect. Order was not significant in explaining variation in extinction risk (n=2087 sp., pMCMC>0.1 for all orders). No interactions between order and threats were significant, and order was not significant in explaining variation in extinction risk (n=1248 sp., pMCMC>0.1 for interactions and all orders). Therefore, we conclude that explicitly including order as a fixed effect would not improve predictions of extinction risk, and of the interactions between orders and threats for which the impact on extinction risk could be estimated, none improve the model's ability to explain variation in extinction risk.

COMMENT: The methodology uses phylogenetic principal component analysis (pPCA) to reduce dimensionality and assess functional trait space, but the choice of three pPCs might not fully capture all relevant variation. The selected number of pPCs could affect the accuracy of functional diversity estimates and the identification of critical traits. I suggest performing robustness checks with different numbers of pPCs and validate the results using alternative dimensionality reduction techniques.

RESPONSE: We chose the number of phylogenetic principal components using a scree plot which suggests three principal components provides optimal description of variance in morphological traits among species. We test the impact of using a different number of phylogenetic principal components (two and four) and alternative dimensionality reduction techniques (principal coordinates analysis and non-metric multidimensional scaling). These analyses and results are described in the new Supplementary information section (*Dimensionality and ordination method used in functional diversity estimations*, lines 235 to 328).

In summary, we found that while the number of principal components used (dimensions) did affect projections of functional diversity loss, the proportion of variance in diversity loss described by dimensionality was low compared to the proportion of variance described by the extinction scenario. Using a different number of principal components did not affect our conclusions that partial abatement will only be partially effective at averting projected diversity loss, and that abatement of habitat loss and degradation and hunting and collection will avoid the greatest proportion of projected diversity loss relative to other drivers. The dimensionality reduction technique (ordination method) had little effect on functional diversity estimations, and did not affect our conclusions.

COMMENT: **Minor**

Title: the title confusing. Usually, I think of targeted recovery programs to include threat reduction. Maybe; 'Threat reduction must be coupled with captive breeding programs to conserve global bird diversity'

RESPONSE: We use the term “recovery program” to refer to ex-situ conservation actions such as captive-breeding efforts, re-introductions into restored habitats, and measures which aim to boost species survival and reproductive success in-situ that don’t involve threat-abatement, such as nest boxes and supplementary feeding. The term “ex-situ actions” is therefore too narrow to capture all conservation actions that could fit within our definition of targeted recovery programs. Our use of the term recovery programme is broadly consistent with the use of this term by Bolam et al. (2023) and with the conservation actions classification scheme used by the IUCN (Salafsky et al., 2008), which includes actions which boost survival and recovery with ex-situ conservation and reintroductions in the “Species management” category. The Global Biodiversity Framework does not include any measures of threat abatement in their description of species recovery action.

“...species recovery actions (such as vaccinations, supplementary feeding, provision of breeding sites, planting and protection of seedlings)...” (Convention on Biological Diversity, 2024).

Nevertheless, we acknowledge that there is some ambiguity as recovery programmes can include measures to tackle specific threats such as the culling of invasive species. To address this, we have added a definition of targeted recovery programmes in the main text to avoid confusion (lines 302-304 and lines 66-69).

COMMENT: Line 61: requires references as this is not the case in most literature I have read

RESPONSE: There are few studies on the projected impact of threat reduction on functional diversity, and especially few that analyse multiple threats. We have given examples in line 63 (end of sentence beginning in line 61) of papers which either assess coverage of conservation actions (Bolam et al., 2023) or assess change in functional diversity with a single threat (Hughes et al., 2023).

COMMENT: Line 65: the term ‘targeted recovery programmes’ is confusing, are you talking about captive breeding (ex-situ actions), or recovery plans which capture all that is needed (invasive species management, appropriate fire regimes, etc)?

RESPONSE: We have now added a definition of the term targeted recovery programme (line 302-304 and lines 66-69).

See response above for more information.

COMMENT: Line 70: Unclear as to what ‘abatement can reduce the need for direct action remains unclear.’ Usually we talk about threat abatement (i.e., reducing invasive cat populations) as direct action, but that doesn’t seem to be the case here?

RESPONSE: We have reworded this sentence (line 69-73).

COMMENT: Maybe add some caveats in the manuscript about some species have already gone extinct but have not yet been updated in the database (e.g., *Neochmia ruficauda* *ruficauda*).

RESPONSE: We have added a sentence to describe this caveat (line 444-446).

COMMENT: The study reconciles taxonomic discrepancies between AVONET and IUCN using a crosswalk but does not mention the extent to which taxonomic changes or updates might impact the results.

RESPONSE: Taxonomic discrepancies were reconciled between AVONET and IUCN using *rl_synonym* from the *rredlist* package. We have added a line to describe the extent of these changes (line 446-448).

There were a lot more taxonomic discrepancies between Birdlife (taxonomy used by AVONET and IUCN) and BirdTree (species nomenclature used by Jetz et al. [2012]), which were resolved with the AVONET crosswalk. We repeated the analyses with all synonyms as they appear in BirdLife (but with principal component analysis rather than phylogenetic principal component analysis), and the results of these analyses were described in the Supplementary information (in the section "*Reconciling BirdLife and BirdTree taxonomies and including all BirdLife synonyms*", line 157-235) and summarised in line 458-462 of the main manuscript.

We have added clarification to line 446 to 448 in the main text, to highlight that taxonomic reconciliation did not affect our conclusions.

Reviewer #3 (Remarks to the Author)

COMMENT: This manuscript models the relationship between threats to birds and their extinction risk, and then predicts extinctions under different scenarios of threat abatement. The authors use species traits to consider how functional diversity will be affected by predicted extinctions under those threat abatement scenarios. They conclude that threat abatement alone is not sufficient to prevent extinctions and the loss of functional diversity among birds.

The study is interesting and timely, and conclusions support previous work. However, it is quite dense and the methods are difficult to get to grips with. My overarching comment is conceptual, and I have two major queries on the methods, which require clarification.

RESPONSE: We thank the referee for their efforts in reviewing this manuscript. We outline the changes we have made and how they address the referees' comments, below.

Main comment on the concept:

COMMENT: I am not convinced that the model structure forms an ecological basis for the conclusion that species would continue to go extinct even under complete threat abatement. If I understand correctly, then under the complete threat abatement scenario, all threats affecting the species are set to zero, and hence the drivers of predicted extinction risk are the random model effects, which are phylogeny and longitude/latitude. I don't believe that from an ecological perspective, these variables are sufficient to predict the ability of species to

recover, or not, if their threats were abated. The ability of a species to recover without targeted intervention is likely to be reliant on the history of impacts on the species (resulting in e.g. very small populations or genetic bottlenecks) as well as environmental factors such as the availability of habitat, see e.g. the justifications provided by Bolam et al. 2022.

Phylogeny and location have been shown to explain variation in extinction risk, but to what extent do they explain recovery? I think this is a model of extinction risk, not of recovery, hence while I think that the model reaches reasonable conclusions, I'm not sure that it does so in an ecologically sound way.

RESPONSE: We agree with the reviewer that fitting a more mechanistic model of recovery would be interesting. However, to our knowledge, understanding of the drivers of recovery is limited and it is unclear which ecological factors widely facilitate (or prevent) recovery. In the future, the IUCN Green Status of Species may provide a means to model species recovery at a large scale. Currently it is in its preliminary stages and Green Status assessments are available for 181 species (only 35 birds) as a proof of concept (Grace et al., 2021). As a small test, for the 24 bird species included in our model and this preliminary assessment (Grace et al. [2021] also assessed eight Least Concern bird species, one subspecies, and two species which did not have a synonym match under BirdTree so were not included our the model) we found that our prediction of extinction risk under complete abatement was significantly lower for species with higher recovery scores ($n=24$, $\beta=-0.01$, $p<0.001$ in a linear model of species recovery score against the log₁₀-transformed mean probability of extinction under complete abatement across 1000 model iterations). Of the five species with the highest recovery scores that were viable and functional across 60-90% of their indigenous range, all had a mean extinction probability of less 0.1 under complete abatement, and four out of five had a mean extinction probably of less than 0.05. Conversely, the five species with the lowest recovery scores that were viable and functional across less than a quarter of their indigenous range (species recovery scores ranged between 8 and 25%) had higher mean extinction probabilities under complete abatement ranging from 0.19 to 0.7. While this is a very small test, it means our model can capture some of our understanding of factors affecting recovery potential.

In our model the spatial and phylogenetic random effects can account for higher or lower than expected extinction risk given threats due to non-random differences in baseline extinction rates across space and the avian tree of life. For example, island endemics are particularly sensitive to extinction, due to their small and isolated ranges (Matthews et al., 2024) which may be captured through spatial random effects, and evolution of traits associated with increased extinction risk such as flightlessness, which may be captured by phylogenetic random effects (Matthews et al., 2024). Deviations from expected extinction risk, given the threats which a species are affected by, can reflect fast or slow life history (Purvis et al., 2000), variation in overlap with areas of high human influence (Di Marco et al., 2018), and isolation and connectivity (e.g. higher risk for island endemics) (Matthews et al., 2024). These factors are expected to be important for explaining variation in both extinction risk and species recovery (Oliver et al., 2013; Capdevila et al., 2022; Feng et al., 2022), and often exhibit high degrees of spatial or phylogenetic correlation (Böhning-Gaese and Oberrath, 1999; Evans and Gaston, 2005; Orme et al., 2006). The relevance of spatial and

phylogenetic variables is supported by the fact that background extinction rates, and extinctions due to stochastic events (rather than due to human activity) vary by taxonomic group (Ceballos et al., 2015), and across space (Pimm et al., 2014). While not perfect, our model does indirectly account for some of the factors linked to recovery potential.

We agree it would be optimal to have information on what drives recovery more specifically. However, data on species recovery potential do not exist, and thus, describing potential recovery under threat abatement with a model of extinction risk is the best we can do. Previous studies have used this approach. For example, Di Marco et al. (2012), created the Extinction risk Reduction Opportunity metric by comparing mammalian extinction risk categories as listed to extinction risk categories predicted using life history traits. They identified species with greatest recovery potential as those which had the greatest extinction risk relative to that predicted from life history traits.

COMMENT: Taking the text at L142-154, I don't find this paragraph provides a convincing discussion of the model behaviour, and I'm struggling to understand the ecological explanations here. In what way, i.e. through which specific variables, does the model capture species that are vulnerable despite having few threats? The example of the Grenada dove seems to suggest that actually the modelling framework itself is not particularly appropriate for this kind of case. The manuscript states that the dove is predicted to remain at risk of extinction even under complete threat abatement, because of it has a small population in a tiny area affected by multiple threats. However, if there was complete threat abatement, then at a minimum, the dove would no longer be affected by those threats, and potentially more likely, the threats would no longer be operating in the area. So what does multiple threats have to do with it? I have inferred that the model predicts a high extinction risk for this species because of the phylogenetic and geographical random effects in the model – is that correct? If so, this really runs counter to the exercise, as the abatement of threats to the species would be place-specific, meaning Grenada would no longer be a high-threat environment for the species. The potential limitation, then, to the recovery of the species would be its small population size and the small area of Grenada combined with limited natural habitat – these would be the key attributes to identifying that the species requires more than threat abatement to recover. I don't think that these are captured in the model. Hence, I don't find this interpretation convincing because it is not an accurate description of how the model is behaving. I would be open to persuasion on this point, but I do not currently see evidence in the manuscript for the suitability of this model to predict species response to threat abatement.

RESPONSE: First, we thank the reviewer for identifying an error. When looking for examples we mistakenly wrongly reported *Leptotila wellsi* as a species predicted to have high extinction risk under complete abatement. In fact, our model predicts low extinction risk under complete abatement *Leptotila wellsi*. We have corrected the example now discussing *Dicaeum quadricolor* (line 154-159).

More generally, as described in the previous comment, our model captures variation in species which are vulnerable despite being affected by few threats (or vice versa) through phylogenetic and spatial random effects. Species may vary in their extinction risk following

threat abatement due to the availability of remaining intact habitat or due to broader patterns in extinction rates linked to latitudinal gradients in climate variability (Mittelbach et al., 2007) and range size (Pimm et al., 2014). Defining a model that disentangles and identifies the mechanisms beyond threats that drive extinction and recovery is beyond the scope of this work. We argue that using phylogeny and spatial variables to capture attributes not reflected in the threats is a suitable approach to evaluate how changes in threats could influence predicted extinction risk, when recovery is understood as a reduction in predicted extinction risk. We also note that the potential to model other attributes is limited by data availability, as life history data are not available for many species.

While a full analysis of the factors contributing to vulnerability was beyond the scope of this study, we tested whether factors mentioned by the reviewer: population size and remaining intact habitat (using the human modification index [BirdLife International and Handbook of the Birds of the World, 2023]) were associated with extinction risk under complete abatement (model details below). We found that species with smaller population sizes did have higher predicted extinction risk with threat abatement ($\beta = -0.34$, $n = 1649$, $p < 0.002$) suggesting our model can capture factors that can limit recovery. On the other hand, we did not find that more intact areas had lower risk ($\beta = 0.28$, $n = 1649$, $p > 0.05$), although it is possible this reflects the fact that the human modification index uses information on the density of human infrastructure (settlement, transport lines, energy and mining production) which reflect threats we modelled directly. We have added discussion of the factors in the model that can result in high predicted extinction risk even under complete abatement (line 143-153).

We have reworded the paragraphs in line 143-162 to improve the description of the model. It is true that having multiple threats is not relevant given that this is complete abatement, so we have removed mention of this. In addition, we have added an explanation of the role of random effects which predict species extinction under complete abatement.

Model details: the model was fitted for 1649 species as those with no synonym matches in Callaghan et al. (2021), and with restricted range data were not included. We used a gaussian MCMCglmm model of mean sum of spatial and phylogenetic random effects for each species across iterations (representing extinction risk under complete abatement and no past threats), against log₁₀-transformed population size estimates, obtained from Callaghan et al. (2021), and mean human footprint index across a species range calculated from the human modification index (Kennedy et al., 2019) using BirdLife data on species distributions (BirdLife International and Handbook of the Birds of the World, 2023). One phylogenetic random effect was used with a chi-squared prior. We did not include spatial random effects as the posterior random effect variances of spatial variances clustered close to zero. Normal priors were used for fixed effects with an expected value (μ) of 0 and strength of belief (V) of 1×10^8 (Hadfield, 2017). MCMC chains were run for 53 000 iterations with a 3000 iteration burn-in period and a thinning interval of 100 iterations.

COMMENT: Main comments on methods 1

The manuscript does not include the final model used to predict species extinction risk based on threats as fixed effects and phylogeny/geography as random effects. It would help

the reader understand the modelling, and hence how conclusions are reached, if the structure of the final model was presented. This is a fairly major omission in terms of the transparency of the methods.

RESPONSE: We have added the final model structure (line 597-599). We also highlight that the parameter estimates of fixed effects (and whether they were included in the final reduced model) can be found in supplementary dataset 1.

COMMENT: Main comments on methods 2

It is unclear to me how Least Concern species have been treated. As far as I understand, the model of extinction risk against threats does not include LC species, yet the model predicting numbers of extinctions does seem to include LC species, despite the fact that their threat profile cannot be varied among scenarios. I feel like this exaggerates the sample size, given LC species do not vary among scenarios. If the authors wish to argue against this, then I would suggest that there needs to be a much more explicit statement on the fact that LC species remain LC in all scenarios and are not considered to be impacted by threats. Related to this, at L358-361, I don't understand what is meant by not including threats affecting LC species – does this not mean rather that LC species as a whole are not included?

RESPONSE: Our extinction risk model does not include Least Concern species because threat classification for these species is inconsistent (not comprehensively conducted during assessments); therefore, threats may not be listed even if occurring. Nevertheless, it is meaningful to report projected diversity loss as a percentage of the complete assemblage, rather than as the percentage of threatened species so the reader can understand how threats and threat abatement are expected to affect avian diversity as a whole.

We predicted extinctions using a two-step process. Firstly, we predicted the probability that species were assigned to each extinction risk category using the extinction risk model. Species not included in the extinction risk model (Least Concern species or species with missing range or threat data) were given a probability of 1 of being assigned to their extinction risk category as currently listed by the IUCN Red List and a probability of 0 of being assigned to other extinction risk categories. Secondly, we stochastically projected extinctions based on expected extinction probabilities for each extinction risk category. Therefore, we accounted for the low chance that Least Concern species could go to extinct due to stochastic events (probability of extinction in the next 100 years of 0.0001), and included Least Concern, Near Threatened and threatened species in extinction scenarios. As the reviewer highlights, because Least Concern species were not included in the extinction risk model, they could not show a response to abatement scenarios, but as that is lowest ("background") risk of extinction, abatement has by definition no effect on LC species. We have made this explicit in lines 136-137, lines 390-391, lines 408-409, lines 421-424, 429-431 and lines 670-672. We have also reworded the methods (particularly lines 615-628) to make our approach clearer.

COMMENT: Specific comments

Fig 1a. I don't understand why circles are used here – are the extinctions in complete abatement a perfect subset of the extinctions in partial abatement, i.e. the same species? Why is the scale on the left-hand side non-linear?

RESPONSE: We have changed figure 1 to make it clearer, and to show raw data points and more obvious indications of uncertainty (distribution of data in figure 1a shown through violin plots, and standard deviation in figure 1b through error bars).

COMMENT: L137-141. Re-running the analysis using LC weightings for NT species doesn't particularly make sense to me, as I don't think we can expect NT species to have comparable threat profiles to LC species.

RESPONSE: We did not rerun the model, but rather used the output of the model (predicted probability of being classified into each of Red List categories from NT to CR) differently to predict extinctions. We did this because our model does not include Least Concern species, and so cannot predict assignment to the category Least Concern. However, in reality threat abatement could take a currently threatened species to a LC category (full recovery). To evaluate whether predicted species and functional loss would be much lower if species could recover to Least Concern, we tested the effect of assuming a low (equal to LC) extinction probability of 0.0001 for the Near Threatened category. Therefore, species predicted to be NT would in fact recover all the way to a LC status (that means their probability of extinction would be 0.0001 rather than 0.01).

COMMENT: L155-161. Firstly, which global conservation strategies make this assumption? STAR is not a conservation strategy and, for example, the Kunming-Montreal Global Biodiversity Framework includes Target 4, which was developed specifically to address the fact that some species will need targeted conservation. Secondly, I think there needs to be an explanation as to exactly which model parameters are predicting that there will still be species extinctions when all threats are alleviated. Taking the example of the STAR metric – Mair et al. 2021 state that their major assumption does not hold for species listed under specific Red List criteria. Whereas it is not at all clear here what is driving those predicted extinctions in the model.

RESPONSE: We have edited line 163-166 (was line 155-161).

Target 4 was added to the Global Biodiversity Framework in a late draft (Bolam et al., 2023), and while seven of the targets are about threat abatement only one is about targeted recovery. Increased ambition and action on threat reduction is needed, but our results suggest that targeted recovery programmes are also essential and should receive better attention in conservation strategies.

To further clarify, we have added a description of the factors in the model resulting in predicted extinctions even under complete abatement (line 143-162). For a full discussion see page 15-17 of this document.

COMMENT: Fig 2: can the pPC axes be described? It's not clear what functional space is

represented here, and so it's not possible to understand what is lost or not. The use of silhouettes makes for an attractive figure, but they don't provide enough information to understand the patterns.

RESPONSE: We have added a description of the phylogenetic principal components into the figure caption of figure 2 (line 395-399).

COMMENT: Fig 3: Similarly, the reliance on silhouettes means I'm not sure what variation is related to the drivers of extinction. Part b is a nice figure but it doesn't allow me to see the data supporting the conclusion that drivers of extinction are unevenly distributed across the avian tree of life.

RESPONSE: We have added a description of the phylogenetic principal components into the figure caption of figure 3 (line 417-421). We have reworded the caption of figure 3 to reflect the fact that the figure shows that the drivers of extinction vary across the tree of life, as we provide no quantitative evidence in figure 3 that the drivers of extinction are unevenly distributed with respect to phylogeny.

COMMENT: L247-280. This section could be re-written to be more accessible. The text describes model results in relation to pPC axes, and then includes the relevant morphology description in brackets, which makes it hard work for the reader. L273-275 presents results the other way around and is much easier to understand.

RESPONSE: We have reworded lines 264-298 (was lines 247-280) to make this section easier to read.

COMMENT: L474-481. I'm not convinced by the treatment of multiple threats under the same level 2 category - this is effectively ignoring multiple threats. The threat classification system is hierarchical, which means that each of the most detailed levels documents a different threat, even if those different threats are e.g. three different invasive species.

RESPONSE: Unfortunately, third level threats are not always described (e.g. for some species listed as affected by a level 2 threat category for which level 3 subcategories do exist, there are no listed level 3 subcategories for that threat). As such there is inconsistency in the use of the level 3 classification that makes it impossible to incorporate that level.

We have clarified this in the text (line 545-548).

COMMENT: L503-511. The purpose of grouping threats into broad categories of drivers is not explained here, and hence this paragraph doesn't make sense to the reader. I think in general that some careful revision of methods structure to make the flow more logical could help the reader understand what has been done.

RESPONSE: We have restructured and rewritten the methods to make them easier to understand.

COMMENT: L534-537. How can a DD species have a probability of extinction? It is not explained in the text what probability of extinction is actually assigned to DD species.

RESPONSE: We have now included the probability of extinction assigned to data-deficient species (line 627-628).

COMMENT: L554-556. This is a confusing assumption, surely under this logic, in the threat abatement scenarios every threat that has been abated should become a past threat, and so should be maintained rather than given a zero?

RESPONSE: Past threats that are unlikely to return present a challenge because if we assume that all threats in the past make an analogous contribution to extinction risk as current and future threats, then the impact of past threats may be overestimated, but if we remove the impact of past threats altogether, we assume that species extinction risk is completely independent of threats which have affected them in the past, and the impact of past threats may be underestimated. As the reviewer comments above, it is unlikely that all past effects will have no impact on species extinction risk, especially when past threats have had a severe impact on a species, but it is equally unrealistic to assume all past threats will have permanent effects.

In the analyses described in the main text, we assume that the impact of threats listed as "*Past, Unlikely to Return*" cannot be abated. As such, the impact of extinction lag from listed past threats is not underestimated because they are included when projecting extinction risk under complete abatement. Our approach optimistically assumes that current and future threats cannot only be abated but also that any extinction lag resulting from these threats can be erased. So, in our approach, under complete abatement the model will predict the Red List category of a species with listed past and current threats as a function of its listed past threats (and random effects) whereas for a species with only current threats listed, the Red List category will be predicted using only random effects (abatement remove the current threat with no legacy effects).

We initially wanted to explicitly estimate the effect of past threats to better capture legacy effects by modelling past threats separately from current threats. For each threat category we would then have two predictors: one capturing the effect of current threats (species listed as currently affected) and another predictor for the effect of past threats (species listed as affected in the past). Unfortunately, information on past threats is very limited, for 15 out the 38 threats categories there were no species listed as affected under the timing "*Past, Unlikely to Return*", and only 6 categories had data for more than 10 species. Therefore, we could not include past effects separately to current and future threats. If we could include past threats separately, however, it would not change our message, as the impact of extinction lag resulting from current and future threats could only increase extinction risk, meaning complete abatement would still be insufficient to prevent biodiversity loss.

Arguably there could be an even more optimistic scenario in which extinction lag from all threats including past threats could be removed, such that abated and past threats would have no impact on species extinction risk. Under this assumption past threats that are unlikely to return are not used when fitting the model and do not contribute to species extinction risk in either the baseline or the complete abatement scenario. This more optimistic scenario did not quantitatively change the result, with nearly identical projected

species extinctions and functional richness loss in both the baseline scenario and complete abatement scenario (table 4). This model was fitted using the same priors, number of iterations, thinning interval, and burn in period as described in the main manuscript.

Table 4. The effect of assuming that the impacts of past threats are retained or removed. Species extinctions and functional richness loss under the baseline scenario and the complete abatement scenario, when past threats are included in the extinction risk model (with past threats) and contribute to extinction risk in extinction scenarios, and when past threats are not included in the extinction risk model and do not contribute to extinction risk in extinction scenarios (without past threats). Complete abatement refers to biodiversity loss remaining after complete abatement (rather than the reduction in biodiversity loss).

	Species extinctions		Functional richness loss	
	With past	Without past	With past	Without past
Baseline	5.2 ± 0.2% (517 ± 19 sp.)	5.2% ± 0.2% (518 ± 17 sp.)	3.2 ± 0.4%	3.2 ± 0.4%
Complete abatement	2.6 ± 0.2% (254 ± 19 sp.)	2.7% ± 0.2% (254 ± 17 sp.)	1.5 ± 0.3%	1.5 ± 0.3%

Reviewer #4 (Remarks to the Author)

COMMENT: **Key results**

The stand-out finding of this study is that even a scenario of worldwide large-scale threat abatement is predicted only to mitigate 50% of the biodiversity loss (both taxonomic and functional) expected from ongoing, and projected to continue, threat levels. This is notably at odds with an assumed outcome that complete threat abatement would allow the great majority of species to be downgraded to Least Concern. Importantly, relative levels of mitigation between taxonomic and functional diversity loss are found to vary according to the specific threat driver being abated. Finally, the findings of this study point to the need for a combination of large-scale threat abatement and targeted species recovery programmes to conserve maximum levels of biodiversity.

RESPONSE: We thank the referee for their efforts in reviewing this manuscript and their positive feedback. We outline the changes we have made and how they address the referees' comments, below.

COMMENT: **Validity**

The data interpretation and robustness of the conclusions are generally very good. The focus on the statistic of avoidance of 68% of projected functional diversity loss by targeted protection of 100 most functionally unique threatened birds is a little crude. In line with the use of several abatement scenarios, I would prefer to understand the sensitivity of these statistics to several different multiples of the number of functionally unique species targeted.

RESPONSE: The reviewer makes a good point, and we have now run the analysis several times varying the number of unique threatened species targeted from 40 (~3% of threatened

species studied) to 200 (~16% of threatened species studied) at increments of 20 species. We have updated the methods (line 767-773), and the main text (line 309-311) to reflect this, and plotted the results in extended data figure 4. We found that preventing at least 50% of functional diversity loss requires targeting 100-120 species, while protecting more than $\frac{3}{4}$ may require targeting 180-200 species.

COMMENT: Also, emphasis on the 68% figure raises a couple of points. On the one hand 68% of the 3.2% of projected FD decline under baseline conditions, is equivalent to circa 2% of global avian FD loss avoided. While 2% sounds less significant, it would be illustrative to explicitly state that this means targeting of circa 1% of avian species saves around 2% of avian FD, if that 1% is the most functionally unique threatened species.

RESPONSE: Thank you for this suggestion, we have added a line to describe this (line 314-316).

COMMENT: Significance

This is a significant step forward in understanding the influence of threat drivers on taxonomic and functional diversity. This study provides further clear evidence of the critical need for both large-scale threat abatement as well as targeted species interventions and will be of wide interest to conservation ecologists, conservation managers and policy makers. This further underscores the radical steps and commitments needed to reverse biodiversity declines. The finding that 'hunting and collection' and 'disturbance and accidental mortality' disproportionately impacts functional diversity, hence ecosystem functioning and resilience, is an important message of this work. While halting wholesale habitat loss is a top priority, the steady erosion of ecosystem functional elements via various forms of disturbance and exploitation is an important implication.

Data and methodology

Overall, the approach used is valid and important in its consideration of the impacts of different threats upon avian functional trait space. The main text acknowledges the optimistic assumptions of a worldwide complete abatement of threats scenario. The additional stepped scenarios (partial and minimal abatement) are obviously simplified in being globally uniformly implemented. Nevertheless, such an approach seems valid given what is already a complex and ambitious analysis to communicate in one article, namely considering multiple dimensions of avian niche space, and treating, in combination and separately, multiple threat drivers.

RESPONSE: We thank the reviewer for the positive comments and are happy they see the value and novelty of our study.

COMMENT: Figure 2. Currently the mismatches/differences within trait space between proportional decline (a and b) and averted proportional decline (c and d) are quite subtle and might be made more obvious by 'difference maps' e.g. (a minus c), and (b minus d), hence effective in showing the shortfall in abatement over mismatch in different parts of the trait space.

Figure 2, and lines 368-370 of legend. It would surely be valuable and interesting to highlight

areas where decrease in functional trait space occupation was not preventable by managing drivers of extinction, rather than maintaining these areas as grey and indistinguishable from areas where there was no decrease in functional trait space occupation?

RESPONSE: Plotting difference maps between proportional decline and averted proportional decline (proportion saved in figures below) shows proportional decline not averted (proportion not saved in figures below). We thank the reviewer for this suggestion, and we have tried plotting it (examples shown for pPC1 and pPC2 below), but we are not sure that it adds much to the interpretation of the plot. If the reviewer feels adding a difference plot (like panel b in figure 1 below) would be useful, we could add it as a supplementary figure.

There were very few areas of trait space (there are three blue pixels in each panel below) where complete abatement made no contribution to reducing the loss in density of occupation. There were however many pixels where complete abatement resulted in only a very small reduction in the density lost (light yellow in examples below). Nevertheless, we have updated figure 2 in the manuscript so that areas where complete abatement made no contribution to reducing the loss in density of occupation are shown in blue (to match panel a in figure 1 below).

As a proportion of density in full assemblage

As a proportion of density lost under baseline extinction scenario

Figure 1 Averted decline (a and c, “*proportion saved*”) and decline not averted (b and d, “*proportion not saved*”) under the complete abatement scenario in which all current and future drivers of extinction, including those that occurred in the past and are likely to return, are removed entirely, for pPC1 and pPC2. Averted decline and decline not averted are expressed as a proportion of the density of occupation in the full assemblage (a and b) and as a proportion of density lost under the baseline extinction scenario. Grey indicates areas where no density was lost under the baseline extinction scenario and blue indicates where no density loss was averted under the complete abatement scenario.

COMMENT: Extended Data Figure 4. Y-axis would be better scaled as percentage variance attributable to each pPC - eigenvalues being only meaningful in a relative sense. Knowing the absolute variance attributable to pPC4 and onwards would be more meaningful than use of an elbow as the cutoff.

RESPONSE: We have rescaled the y axis to the percentage variance attributable to each pPC (now extended data fig. 5). We used the elbow to identify the number of pPCs to use, because variance thresholds can be arbitrary. However, we have also tested the sensitivity of

our results to this decision by estimating functional diversity loss under extinction scenarios with two or four pPCs. While using a different number of pPCs did affect estimations of functional diversity loss it did not change our conclusions (*Dimensionality and ordination method used in functional diversity estimations*, lines 235 to 328).

COMMENT: Supplementary Figure 1. This figure legend is not stand alone - unclear what relative density refers to here, or what this set of figure elements is showing us, without reference to the main text.

RESPONSE: We have edited the figure legend of Supplementary figure 1 (line 452-465 of Supplementary information).

COMMENT: Analytical approach

The treatment of avian niche space variation using a functional trait space approach, combining phylogenetic PCA and functional diversity estimation using probabilistic hypervolumes (rather than convex hull volumes), are robust approaches, as is the use of models fitting random effects to account for spatial and phylogenetic relationships among species.

Suggested improvements

Other than sensitivity suggested above, I don't have further suggestions for additional analyses.

Clarity and context

The clarity in reporting the key results could be improved. Specifically, in the abstract (lines 33-36), where it says "the immediate abatement of all threats across at least half of species ranges for ~10,000 bird species - will only prevent half of projected species extinctions and functional diversity loss attributable to current and future threats in the next 100 years." This is at odds with the subsequent main text where it says (lines 131-132): "Under the complete abatement scenario, half of biodiversity loss predicted under the baseline scenario could be prevented" where complete abatement is previously reported (lines 120-121) as "removal of all direct drivers of extinction across the entirety of all species ranges". Question for clarification: is prevention of loss of half of biodiversity associated with removal of all threats across the entirety of species ranges or at least half of species ranges?

RESPONSE: We have rewritten these sections for clarity. In summary, in the complete abatement scenario $\frac{1}{2}$ of projected biodiversity loss under the baseline scenario can be prevented, while only $\frac{1}{4}$ is prevented in the partial abatement scenario. That means only half of the "preventable loss" (that avoided under complete abatement) is prevented in the partial abatement scenario.

We have clarified this in line 184-187. We have also edited line 264-298 to improve readability.

COMMENT: Lines 142-154. This section highlights some species for which large-scale threat abatement will have little or no benefit. However, it does not make clear which remaining mechanisms (given complete abatement) inherent in the model are driving them to likely extinction.

RESPONSE: We have edited what was lines 142-154 to make clear which remaining mechanisms in the model are driving species to extinction under complete abatement (line 143-162).

COMMENT: References

The manuscript appropriately references previous literature.

References

- Ali, J. R., Blonder, B. W., Pigot, A. L., & Tobias, J. A. (2023). Bird extinctions threaten to cause disproportionate reductions of functional diversity and uniqueness. *Functional Ecology*, 37(1), 162-175.
- Andermann, T., Faurby, S., Cooke, R., Silvestro, D. and Antonelli, A. (2021), iucn_sim: a new program to simulate future extinctions based on IUCN threat status. *Ecography*, 44, 162-176.
- BirdLife International and Handbook of the Birds of the World. (2023). Bird species distribution maps of the world. Version 2023.1. Available at: <http://datazone.birdlife.org/species/requestdis>.
- Böhning-Gaese, K., & Oberrath, R. (1999). Phylogenetic effects on morphological, life-history, behavioural and ecological traits of birds. *Evolutionary Ecology Research*, 1(3), 347-364.
- Bolam, F., C., Ahumada, J., Akçakaya, H., R., Brooks, T., M., Elliott, W., Hoban, S., ... & Butchart, S., H. (2023). Over half of threatened species require targeted recovery actions to avert human-induced extinction. *Frontiers in Ecology and the Environment*, 21(2), 64-70. <https://doi.org/10.1002/fee.2537>.
- Bregman, T., P., Lees, A., C., MacGregor, H., E., Darski, B., de Moura, N., G., Aleixo, A., ... & Tobias, J., A. (2016). Using avian functional traits to assess the impact of land-cover change on ecosystem processes linked to resilience in tropical forests. *Proceedings of the Royal Society B: Biological Sciences*, 283(1844), 20161289.
- Callaghan, C., T., Nakagawa, S., & Cornwell, W., K. (2021). Global abundance estimates for 9,700 bird species. *PNAS*, 118 (21), e2023170118. <https://doi.org/10.1073/pnas.2023170118>.
- Capdevila, P., Stott, I., Cant, J., Beger, M., Rowlands, G., Grace, M., & Salguero-Gómez, R. (2022). Life history mediates the trade-offs among different components of demographic resilience. *Ecology letters*, 25(6), 1566–1579. <https://doi.org/10.1111/ele.14004>.
- Capellini, I., Baker, J., Allen, W., L., Street, S., E., & Venditti, C. (2015). The role of life history traits in mammalian invasion success. *Ecology Letters*, 18, 1099-1107. <https://doi.org/10.1111/ele.12493>.
- Carmona, C., P., Tamme, R., Pärtel, M., De Bello, F., Brosse, S., Capdevila, P., González, R. M., González-Suárez, M., Salguero-Gómez, R., Vásquez-Valderrama, M., & Toussaint, A. (2021). Erosion of global functional diversity across the tree of life. *Science Advances*, 7(13), eabf2675.

Ceballos, G., Ehrlich, P. R., Barnosky, A. D., García, A., Pringle, R. M., & Palmer, T. M. (2015). Accelerated modern human-induced species losses: Entering the sixth mass extinction. *Science advances*, 1(5), e1400253.

Convention on Biological Diversity (2024). Consolidated guidance notes for the targets of the Kunming-Montreal Biodiversity Framework. Available at: <https://www.cbd.int/gbf/targets/notes.shtml>.

de Villemereuil, P., Gimenez, O., & Doligez, B. (2013). Comparing parent-offspring regression with frequentist and Bayesian animal models to estimate heritability in wild populations: a simulation study for Gaussian and binary traits. *Methods in Ecology and Evolution*, 4(3), 260-275. <https://doi.org/10.1111/2041-210X.12011>.

Dehling, D., M., Jordano, P., Schaefer, H., M., Böhning-Gaese, K., & Schleuning, M. (2016). Morphology predicts species' functional roles and their degree of specialization in plant-frugivore interactions. *Proceedings of the Royal Society B: Biological sciences*, 283(1823), 20152444.

Di Marco, M., Cardillo, M., Possingham, H., P., Wilson, K., A., Blomberg, S., P., Boitani, L. & Rondinini, C. (2012). A novel approach for global mammal extinction risk reduction. *Conservation Letters*, 5, 134-141. <https://doi.org/10.1111/j.1755-263X.2011.00219.x>.

Di Marco, M., Venter, O., Possingham, H., P., & Watson, J., E., M. (2018). Changes in human footprint drive changes in species extinction risk. *Nature Communications*, 9, 4621. <https://doi.org/10.1038/s41467-018-07049-5>.

Evans, K., L., & Gaston, K., J. (2005). People, energy and avian species richness. *Global Ecology and Biogeography*, 14, 187-196. <https://doi.org/10.1111/j.1466-822X.2004.00139.x>.

Feng, C., T., Cao, M., Liu, F., Z., Zhou, Y., Du, J., H., Zhang, L., B., Huang, W., J., Luo, J., W., Li, J., S., & Wang, W. (2022). Improving protected area effectiveness through consideration of different human-pressure baselines. *Conservation Biology*, 36, e13887. <https://doi.org/10.1111/cobi.13887>.

Fox, A., D. (2004). Has Danish agriculture maintained farmland bird populations?. *Journal of Applied Ecology*, 41, 427-439. <https://doi.org/10.1111/j.0021-8901.2004.00917.x>.

Garnett, S. T., Butchart, S. H. M., Baker, G. B., et al. (2019). Metrics of progress in the understanding and management of threats to Australian birds. *Conservation Biology*, 33(2), 456-468.

Grace M., K., Akçakaya H., R., Bennett E., L., Brooks, T., M., Heath, A., Hedges, S., Hilton-Taylor, C., Hoffman, M., Hochkirk, A., Jenkins, R., ... & Young, S. (2021). Testing a global standard for quantifying species recovery and assessing conservation impact. *Conservation Biology*, 35, 1833-1849. <https://doi.org/10.1111/cobi.13756>.

Hadfield, J., D. (2017). MCMCglmm Course Notes. <http://cran.nexr.com/web/packages/MCMCglmm/vignettes/CourseNotes.pdf>.

- Harfoot, M. B., Johnston, A., Balmford, A., Burgess, N. D., Butchart, S. H., Dias, M. P., ... & Geldmann, J. (2021). Using the IUCN Red List to map threats to terrestrial vertebrates at global scale. *Nature Ecology & Evolution*, 5(11), 1510-1519.
- Hordley, L. A., Gillings, S., Petchey, O. L., Tobias, J. A., & Oliver, T. H. (2021). Diversity of response and effect traits provides complementary information about avian community dynamics linked to ecological function. *Functional Ecology*, 35(9), 1938-1950.
- Hutchings, J. A., Butchart, S. H., Collen, B., Schwartz, M. K., & Waples, R. S. (2012). Red flags: correlates of impaired species recovery. *Trends in Ecology & Evolution*, 27(10), 542-546. <https://doi.org/10.1016/j.tree.2012.06.005>.
- IUCN. (2001). IUCN Red List Categories and Criteria: Version 3.1. IUCN Species Survival Commission. IUCN, Gland, Switzerland and Cambridge, UK. Available at: <https://portals.iucn.org/library/sites/library/files/documents/RL-2001-001.pdf>.
- IUCN Standards and Petitions Committee. (2024). Guidelines for Using the IUCN Red List Categories and Criteria. Version 16. Prepared by the Standards and Petitions Committee. Available at: <https://www.iucnredlist.org/documents/RedListGuidelines.pdf>.
- Jetz, W., Thomas, G., Joy, J., Hartmann, K., & Mooers, A. O. (2012). The global diversity of birds in space and time. *Nature*, 491, 444-448.
- Kehoe, L., Romero-Muñoz, A., Polaina, E., Estes, L., Kreft, H., & Kuemmerle, T. (2017). Biodiversity at risk under future cropland expansion and intensification. *Nature Ecology & Evolution*, 1, 1129-1135. <https://doi.org/10.1038/s41559-017-0234-3>.
- Kennedy, C. M., Oakleaf, J. R., Theobald, D. M., Baruch-Mordo, S., & Kiesecker, J. Managing the middle: A shift in conservation priorities based on the global human modification gradient. *Glob Change Biol.* 2019; 25: 811-826. <https://doi.org/10.1111/gcb.14549>.
- Mair, L., Bennun, L. A., Brooks, T. M., et al. (2021). A metric for spatially explicit contributions to science-based species targets. *Nature Ecology & Evolution*, 5, 836-844.
- Maron, M., McAlpine, C. A., Watson, J. E. M., Maxwell, S. & Barnard, P. (2015), Climate-induced resource bottlenecks exacerbate species vulnerability: a review. *Diversity and Distributions*, 21, 731-743. <https://doi.org/10.1111/ddi.12339>.
- Matthews, T., Triantis, K., Wayman, J. P., Martin, T. E., Hume, J. P., Cardoso, P., ... & Sadler, J. (2024). The global loss of avian functional and phylogenetic diversity from anthropogenic extinctions. *Science*, 386(6717), 55-60. <https://doi.org/10.1126/science.adk7898>.
- Mittelbach, G. G., Schemske, D. W., Cornell, H. V., Allen, A. P., Brown, J. M., Bush, M. B., Harrison, S. P., Hurlbert, A. H., Knowlton, N., Lessios, H. A., McCain, C. M., ..., & Turelli, M. (2007). Evolution and the latitudinal diversity gradient: speciation, extinction and biogeography. *Ecology Letters*, 10, 315-331. <https://doi.org/10.1111/j.1461-0248.2007.01020.x>.

- Monroe, M., J., Butchart, S., H., M., Mooers., A., O., & Bokma F. (2019). The dynamics underlying avian extinction trajectories forecast a wave of extinctions. *Biology Letters*, 15, 20190633.
- Oliver, T., H., Brereton, T., & Roy, D., B. (2013), Population resilience to an extreme drought is influenced by habitat area and fragmentation in the local landscape. *Ecography*, 36: 579-586. <https://doi.org/10.1111/j.1600-0587.2012.07665.x>.
- Orme C., D., L., Davies R., G., Olson V., A., Thomas G., H., Ding T., S., Rasmussen, P., C., Ridgely, R., S., Stattersfield, A., J., Owsn, I., P., F., Blackburn, T., M. & Gaston, K., J. (2006) Global Patterns of Geographic Range Size in Birds. *PLOS Biology*, 4(7), e208. <https://doi.org/10.1371/journal.pbio.0040208>.
- Pereira, H., M., Martins, I., S., Rosa, I., M., D., Kim, H., Leadley, P., Popp, A., Van Vuuren, D., P., Hurtt, G., Quoss, L., Arneith, A., ... & Alkemade, R. (2024). Global trends and scenarios for terrestrial biodiversity and ecosystem services from 1900 to 2050. *Science*, 384, 458-465.
- Pigot, A., L., Sheard, C., Miller, E., T., Bregman, T., P., Freeman, B., G., Roll, U., Seddon, N., Trisos, C., H., Weeks, B., C., & Tobias, J., A. (2020). Macroevolutionary convergence connects morphological form to ecological function in birds. *Nature Ecology & Evolution*, 4, 230–239. <https://doi.org/10.1038/s41559-019-1070-4>.
- Pimm, S., L., Jenkins, C., N., Abell, R., Brooks, T., M., Gittleman, J., L., Joppa, L., N., ... & Sexton, J., O. (2014). The biodiversity of species and their rates of extinction, distribution, and protection. *science*, 344(6187), 1246752.
- Purvis, A., Gittleman, J. L., Cowlishaw, G., & Mace, G., M. (2000). Predicting extinction risk in declining species. *Proceedings. Biological sciences*, 267(1456), 1947–1952. <https://doi.org/10.1098/rspb.2000.1234>.
- Salafsky, N., Salzer, D., Stattersfield, A., J., Hilton-Taylor, C., Neugarten, R., Butchart, S., H., M., Collen, B., Cox, N., Master, L., L., O'Connor, S. & Wilkie, D. (2008). A Standard Lexicon for Biodiversity Conservation: Unified Classifications of Threats and Actions. *Conservation Biology*, 22, 897-911. <https://doi.org/10.1111/j.1523-1739.2008.00937.x>.
- Schleuning, M., García, D., Tobias, J.A. (2023) Animal functional traits: Towards a trait-based ecology for whole ecosystems. *Functional Ecology*, 37, 4–12.
- Stewart, P. S., Voskamp, A., Santini, L., Biber, M. F., Devenish, A. J., Hof, C., ... & Tobias, J. A. (2022). Global impacts of climate change on avian functional diversity. *Ecology Letters*, 25(3), 673-685.
- Tobias, J. A., Sheard, C., Pigot, A. L., Devenish, A. J. M., Yang, J., Sayol, F., Neate-Clegg, M. H. C., Alioravainen, N., Weeks, T. L., Barber, R. A., Walkden, P. A., MacGregor, H. E. A., Jones, S. E. I., Vincent, C., Phillips, A. G., Marples, N. M., Montaña-Centellas, F. A., Leandro-Silva, V., Claramunt, S., ... & Schleuning, M. (2022). AVONET: Morphological, ecological and geographical data for all birds. *Ecology Letters*, 25(3), 581–597.

Referees and editor's comments are in blue.

Reviewer #1 (Remarks to the Author)

COMMENT: The revisions to the manuscript are great and have improved the clarity of the research, addressing most of my concerns – particularly the reframing of the results to make it clear that it is not that threat abatement won't work, it is that it is potentially too late for a small set of marginal species where threats have left them irreparably vulnerable to extinction without targeted intervention. The authors made it much clearer why they built their models the way they did and, whilst they still only weakly reflect the real-world application of red list information that underpin assessments, they do fit the purpose of this theoretical piece.

RESPONSE: We thank the referee for their efforts in reviewing this manuscript, and their helpful comments in this and previous rounds of reviews.

My only major concern remains around the interpretation of the results, which initially made it hard to grasp the rationale for the study design used. The authors present work which suggest that abatement is 'only' going to stop 50% of projected extinctions. However, there still seems to be a bit of context missing. It seems implicit that, following total abatement, there are 97.5% of species who do not become extinct because total abatement of threat is sufficient to prevent their extinction. What the authors do not seem to take into account is the future situation where, yes, we still lose 50% of projected extinctions under the total abatement scenario, but the lack of threats to continue to act on species will ensure continuously lower extinction rates beyond the authors' time horizon, compared with the no—or partial—abatement approach. I worry that the somewhat sensationalist approach to the interpretation of the findings have the potential to undermine real-world efforts to abate threat sufficiently to deliver conservation success. Hopefully the authors can consider this in any subsequent modifications.

We agree that it is important to emphasise that threat abatement is essential to prevent Least Concern species from moving to higher threat categories in the long term. While we stated this in the conclusion (lines 355-357), we agree more could be done to emphasise the importance of threat abatement in reducing long term extinction risk throughout the manuscript. We have therefore reworded the abstract (lines 31-32, introduction (lines 60-63, lines 75-77) and results (line 124, lines 214-216) to reflect this.

We do not think our approach is sensationalist. The number of extinctions projected within only a few human generations, is three times the number of extinctions observed since 1500. We show that this will lead to a reduction in functional diversity of the avian assemblage, which could undermine the function and persistence of ecological communities (Cadotte et al., 2011; Wojcik et al., 2021). Reducing extinction risk in the near future is essential to avert this loss. Without focusing on near term extinctions, the importance of conservation measures that prevent imminent species extinctions could be overlooked. Threat reduction is key to reducing long-term extinction risk, but we show that in the near term (i.e. in the next

100 years), we will need complementary measures such as targeted recovery programmes, if we are to prevent all extinctions.

We now have more clearly emphasised that our results pertain to the near future. When describing diversity loss, we always state that extinctions are those projected for the next 100 years. We have edited the abstract, introduction and results (lines 31-32, lines 62-63, lines 75-77, line 124) to further highlight this.

We would like to clarify that the reviewer's statement that, "*there are 97.5% of species who do not become extinct because total abatement of threat is sufficient to prevent their extinction*" is not completely accurate. While 97.4% of species are projected to remain extant under complete abatement, 94.8% of species also remain extant under the baseline scenario, those are species that we do not project will go extinct in the next 100 years, so only 2.6% do not become extinct because of the total abatement of threats. As the reviewer states, threat abatement is still essential for the long-term persistence of Least Concern species and our additions to lines 60-61 and lines 214 to 216 highlight this.

We state biodiversity loss as a percentage of the current diversity (e.g. line 135-136 "*an average loss of $2.6 \pm 0.2\%$ of species richness (254 ± 19 species) and $1.5 \pm 0.3\%$ of functional richness remained*", which makes it clear that 97.4% of species are not projected to go extinct under complete abatement.

Reviewer #2 (Remarks to the Author)

COMMENT: I thank the authors for their detailed responses and amendments to the manuscript. Nothing further from me. All data and code seem to be open and accessible for reproducibility.

RESPONSE: We thank the referee for their efforts in reviewing this manuscript, and their helpful comments in previous rounds of reviews

Reviewer #3 (Remarks to the Author)

COMMENT: The authors have done a good job of responding to the many comments from reviewers. They have defended their approach, provided explanations, conducted additional analyses, and made many of the suggested changes. Altogether the manuscript is much improved.

RESPONSE: We thank the referee for their efforts in reviewing this manuscript, and their helpful comments in this round and previous rounds of reviews.

I have two remaining queries, and then a few minor comments.

Query 1

In response to my comments about whether the model captures extinction risk or recovery potential, the authors provided a nice supporting analysis using Green Status data, alongside

a clear explanation, which is much appreciated. I did, however, find that their explanations of what the random phylogenetic and spatial effects capture varied somewhat among their explanations to different reviewers. In another response to review comment, they state that "This means that if higher/lower threat impact scores are given for particular species groups, for particular parts of the world or for particular threats, due to subjective application of the system for measuring impact, this will be estimated by random and fixed effects in our extinction risk model to provide the best description of extinction risk given the data". This is not the same as their explanation that "In our model the spatial and phylogenetic random effects can account for higher or lower than expected extinction risk given threats due to non-random differences in baseline extinction rates across space and the avian tree of life". Nor as their text in the manuscript at L516-518 saying "Threats varied across the tree of life and spatially so we included phylogeny and spatial variables as random effects".

Clarification would be appreciated, and I think the explanation of what the random phylogenetic and spatial effects capture could be better integrated into the manuscript. Specifically, at L143-153, I think the explanation provided to reviewers was clearer than the text here in the manuscript. In particular, inclusion of supporting references (as was in their response to reviewers) would help.

Additionally, considering L154-162, these examples could be better linked to the explanation of random effects, if explanation and examples were discussed together. I also think the petrel example could be improved from a vague and unreferenced statement of unnamed 'factors associated'.

The phylogenetic and spatial random effects capture variation in extinction risk in species which are vulnerable despite being affected by few threats (or vice versa). Such variation could be due to non-random baseline extinction rates or due to other factors influencing extinction risk and recovery, such as small population size, that are not due to population decline resulting from listed past, ongoing or future threats. In theory, random effects could capture variation due to subjective application of threat listing, but as mentioned in the previous response to reviewers' comments, the concern around subjective application of threat listing applies to impact scores, not estimated scope and severity values, so this is not expected to affect the magnitude of random effects. While the statements around what random effects represent were not necessarily conflicting; we acknowledge that there was a lack of clarity in the description of what random effects represent in the manuscript.

We have edited the manuscript such that the explanation of random effects and examples are provided together, and to include explanations and references included in the response to reviewers' comments (lines 147 to 168). To prevent these paragraphs from becoming overly long we have retained only the Cebu flowerpecker example. We have also edited lines 556-563 (was lines 516-518) in the methods for consistency and clarity.

Finally on this point, I find the statements at L172-176 weak, because the model does not explicitly capture these effects. This is indeed an explanation, but these mechanisms are not captured in the model, and hence this conclusion does not arise from the work presented here.

We believe an explanation of the potential factors that could result in ongoing declines following threat abatement is needed here. We have rephrased lines 179-183 (was line 172-176) to make it clear that we did not explicitly test the reasons for ongoing declines, so these are potential explanations rather than conclusions drawn from our findings.

Query 2

COMMENT: Supplementary Dataset 1. Why do two threats (2.2 and 10) have a negative posterior mean? This seems counterintuitive, and makes me wonder about the behaviour of the model – is it over-fitted?

If the model were overfitted we would expect to see high correlation of posterior estimations between independent variables. We do not find this to be the case however (figure 1) as correlation between posterior estimates of threats was low. The maximum correlation between threats was 0.37 (between 2.1 'Annual and perennial non-timber crops' and 5.1 'Hunting and collecting terrestrial animals'). Therefore, while it is unexpected that 2.2 'Wood and pulp plantations' and 10 'Geological events' have a negative posterior mean, model outputs suggest it is accurately estimating the association between these threats and extinction risk. We added lines 656 to 663 to reflect this.

Threat scope and severity data are often measured, estimated and collated from populations which are affected by multiple threats. Our model estimates the independent contribution of each of these threats to extinction risk, so while it may appear that wood and pulp plantations and geological events are contributing to decline when combined with other threats, our model suggests that in general, species threatened by wood and pulp plantations and geological events have slightly lower extinction risk.

Figure 1 Correlation coefficients (r) between posterior estimates of independent variables (threats) in the extinction risk model. Correlation coefficients vary between -0.24 and 0.37.

Minor comments

COMMENT: L95 should be “predicted” past tense? In general, the text describing methods moves between present and past tense, and I think it would be easier to read if it was all in past tense.

RESPONSE: Thank you for bringing this to our attention. We have corrected the main text and the methods so that we use past tense throughout when referring to completed work in the methods and results.

COMMENT: L133-142. The explanation of this analysis was only clear from the response to reviewers. The reason there might be underestimates is because some species might recover to LC, which is a subtle but important difference to not including LC species in the model. Taking the explanation in the response to reviewers and incorporating into the main text would help here.

RESPONSE: We have reworded lines 137-146 to make this analysis clearer.

COMMENT: L510-514 The authors quote numbers for all species (i.e. including LC), but only include NT and threatened species in the extinction risk model. Surely it is only the species included in the model that are of relevance here?

RESPONSE: We have edited lines 551-553 to quote numbers for Near Threatened and threatened species which have threats listed. We have included 2104 species in these numbers rather than the 2087 included in the model (17 species that were missing spatial data that were excluded from the model were included in the counts) because we only

explain later in the method (lines 618-619) that we don't include species with missing spatial data in the model.

COMMENT: L522-523. "Threats have been comprehensively described for 99% of species in these categories". This is still an incorrect statement – threats have been described, but that does not mean that these descriptions are comprehensive. And indeed the authors contradict this statement at L549 where they say 11% of threat data were missing scope or severity information.

RESPONSE: We have removed the word comprehensively.

COMMENT: L534-548. The authors responded to my comments on the treatment of threat data, but misunderstood my point, so my apologies. I appreciate that threats are not always documented to level 3, and so their treatment of level 3 threats is understandable. My issue was with multiple threats listed under the same level 2 threat, i.e. where they explain that a species may be impacted by two different invasives, which are documented as separate threats, the authors treat these as only one threat. My argument is that they are not one threat, they are two, which may operate in very different ways, impact the species in different and cumulative and/or interactive ways, and require different interventions. I do not think they should be treated as one threat; their impact should instead be summed.

RESPONSE: Only 16% of species-threat combinations (second order threats) have more than one third order threat listed. Of the 3996 third order species-threat combinations 5% have an identical description and code as another third order threat listed for that species, with the only difference being that one or more of scope, severity and timing has complete data where it is absent in the other record, or that the invasive species is specified when it is unspecified in the other record. These records could give information on unique threats, or they could be duplications where threats are relisted when increased data on timing, scope or severity becomes available or the invasive species is identified. Due to this uncertainty, we assigned the maximum expected population decline from the second-order threat and any third-order threats for each second-order threat to avoid double counting of third-order threats.

We have tested the impact of the decision to take the maximum rather than the sum of expected population decline for third order threats on projected species extinctions. We calculated the sum of expected population decline for third order threats and reran the model. Taking the sum of expected population decline from third order threats rather than the maximum expected population decline had minimal impact on the projected extinctions in the baseline scenario (with maximum: 517 ± 19 extinctions, with sum: 518 ± 18 extinctions) or under complete abatement (with maximum: 263 ± 19 extinctions, with sum: 254 ± 18 extinctions). Avoidance of projected extinctions under driver-specific abatement scenarios was also similar whether the sum or maximum expected population decline from third order threats was used (Table 1), with a slight decrease in extinctions avoided under abatement of habitat loss and degradation and a slight increase in extinctions avoided under

abatement of invasive species and disease, when the sum rather than the maximum was used.

Table 1 Number of projected species extinctions avoided in the next 100 years under driver-specific abatement for six major drivers of extinction, when the maximum expected population decline of third order threats (or the second order threat) is used compared to when the sum of expected population decline from third order threats is calculated (or expected population decline from the second order threat is used when that is larger). The number of extinctions avoided is given as a percentage of total species richness (9873 species) in brackets. 1000 iterations were run for each driver-specific abatement scenario, with 9873 species, of which 2087 species were included in the model and could reduction in extinction risk in response to threat abatement.

Driver	Maximum third order threat	Sum of third order threats
Habitat loss and degradation	141 ± 24 species (1.4 ± 0.2%)	130 ± 24 species (1.3 ± 0.2%)
Hunting and collection	42 ± 23 species (0.4 ± 0.2%)	47 ± 22 species (0.5 ± 0.2%)
Climate change and severe weather	37 ± 22 species (0.4 ± 0.2%)	40 ± 22 species (0.4 ± 0.2%)
Disturbance and accidental mortality	15 ± 23 species (0.2 ± 0.2%)	16 ± 22 species (0.2 ± 0.2%)
Invasive species and disease	37 ± 22 species (0.4 ± 0.2%)	49 ± 23 species (0.5 ± 0.2%)
Pollution	8 ± 22 species (0.1 ± 0.2%)	11 ± 22 species (0.1 ± 0.2%)

We have added further justification of the decision to take the maximum expected population declines for threats listed under second order threats in the method (line 589-596),

"We took the maximum expected population decline for second order threats where multiple third order threats were listed as not all second order threats had information on third order threats, and without further information it is difficult to estimate the expected population decline from multiple third order threats. Only 16% of species-threat combinations had more than one third order threat listed, and when running the extinction risk model using the sum of expected population decline rather than the maximum, we found that this had a minimal impact on projected species extinctions."

COMMENT: L627-628. The treatment of DD species is likely to have only a small effect on overall predictions given their small number, but classifying them all as LC is likely an underestimate of extinction risk.

RESPONSE: We have added lines 684-690 to describe the potential impact of uncertainty resulting from data-deficiency on projected diversity loss.

"For many classes, Data Deficient species are likely to be at higher risk of extinction than data-sufficient species⁸², however Butchart and Bird found that this was less so the case for birds⁸³, and because Data Deficient species make up a very small percentage of total species (0.4%), uncertainty over the extinction risk of Data Deficient species was expected to have a negligible impact on projected diversity loss."

Reviewer #4 (Remarks to the Author)

COMMENT: The authors have addressed the comments I made including a sensitivity analysis varying the number of unique threatened species targeted from 40 to 200 and I think Extended Data Figure 4 is a very helpful addition.

RESPONSE: We thank the referee for their efforts in reviewing this manuscript, and their helpful comments in this round and previous rounds of reviews.

The authors have also attempted to plot the difference maps suggested for Figure 2. Although they suggest possibly using their new panel b as a supplementary figure, it is new panel d (proportion of density lost under baseline extinction scenario, not saved) that I find more revealing and complementary to the existing figure in the main paper. However, I wouldn't be prescriptive on using either of these. It is now made clear there are very few areas of trait space (three blue pixels in revised figure) where complete abatement made no contribution to reducing the loss in density of occupation, meaning these are effectively not visible hence not worth showing.

RESPONSE: We thank the reviewer for the initial suggestion of plotting areas of trait space not saved as a proportion of density lost under the baseline extinction scenario. We have now added this as an extended data figure (extended data figure 4 and described it in the methods on lines 760-761).

COMMENT: The explanation of factors in the model driving species to extinction even under complete abatement is now somewhat clearer, although this model outcome is clearly an area of some debate.

RESPONSE: We are glad that the explanation of the model is now clearer. Using the best available data, and an approach that did not require the limiting assumption that all species will return to minimal risk of extinction under complete abatement, we find that there is no evidence that complete threat reduction on its own will be enough to prevent all future extinctions. As this is an assumption that is frequently made in conservation, we welcome debate about our findings and hope that the ensuing discussion assists in the planning of global conservation strategies where both threat reduction and targeted recovery programmes will be needed.

Other changes

We added icon attributions to figure captions and updated figure 3 because one of the icons was partially obscured by the legend.

Literature cited

Cadotte, M., W., Carscadden, K., & Mirotchnick, N. (2011). Beyond species: functional diversity and the maintenance of ecological processes and services. *Journal of Applied Ecology*, 48(5), 1079–1087. <https://doi.org/10.1111/J.1365-2664.2011.02048.X>.

Wojcik, L. A., Ceulemans, R., & Gaedke, U. (2021). Functional diversity buffers the effects of a pulse perturbation on the dynamics of tritrophic food webs. *Ecology and Evolution*, 11(22), 15639–15663. <https://doi.org/10.1002/ece3.8214>.